# ANALYTIC DAG CONSTRAINTS FOR DIFFERENTIABLE DAG LEARNING

**Zhen Zhang**[1]    **Ignavier Ng**[2]    **Dong Gong**[3]    **Yuhang Liu**[1]    **Mingming Gong** [4,5]

**Biwei Huang**[6]    **Kun Zhang**[2,5]    **Anton van den Hengel**[1]    **Javen Qinfeng Shi**[1]

[1] Australian Institute for Machine Learning, The University of Adelaide
[2] Department of Philosophy, Carnegie Mellon University
[3] School of Computer Science and Engineering, The University of New South Wales
[4] School of Mathematics and Statistics, The University of Melbourne
[5] Department of Machine Learning, Mohamed bin Zayed University of Artificial Intelligence
[6] Halicioğlu Data Science Institute (HDSI), UC San Diego

## ABSTRACT

Recovering the underlying Directed Acyclic Graph (DAG) structures from observational data presents a formidable challenge, partly due to the combinatorial nature of the DAG-constrained optimization problem. Recently, researchers have identified gradient vanishing as one of the primary obstacles in differentiable DAG learning and have proposed several DAG constraints to mitigate this issue. By developing the necessary theory to establish a connection between analytic functions and DAG constraints, we demonstrate that analytic functions from the set $\{f(x) = c_0 + \sum_{i=1}^{\infty} c_i x^i | \forall i > 0, c_i > 0; r = \lim_{i \to \infty} c_i/c_{i+1} > 0\}$ can be employed to formulate effective DAG constraints. Furthermore, we establish that this set of functions is closed under several functional operators, including differentiation, summation, and multiplication. Consequently, these operators can be leveraged to create novel DAG constraints based on existing ones. Using these properties, we design a series of DAG constraints and develop an efficient algorithm to evaluate them. Experiments in various settings demonstrate that our DAG constraints outperform previous state-of-the-art comparators. Our implementation is available at `https://github.com/zzhang1987/AnalyticDAGLearning`.

## 1 INTRODUCTION

DAG learning aims to recover Directed Acyclic Graphs (DAGs) from observational data, which is a core problem in many fields, including bioinformatics (Sachs et al., 2005; Zhang et al., 2013), machine learning (Koller and Friedman, 2009), and causal inference (Spirtes et al., 2000). Under certain assumptions (Pearl, 2000; Spirtes et al., 2000), the recovered DAGs could be interpreted causally (Koller and Friedman, 2009).

There are two main categories of DAG learning approaches: constraint-based, and score-based methods. Most constraint-based approaches, *e.g.*, PC (Spirtes and Glymour, 1991), FCI (Spirtes et al., 1995; Colombo et al., 2012), rely on conditional independence tests, which typically necessitate a large sample size (Shah and Peters, 2020; Vowels et al., 2021). The score-based approaches, including exact methods based on dynamic programming (Koivisto and Sood, 2004; Singh and Moore, 2005; Silander and Myllymäki, 2006), A* search (Yuan et al., 2011; Yuan and Malone, 2013), and integer programming (Cussens, 2011), and greedy methods like GES (Chickering, 2002), model the validity of a graph according to some score function and are often formulated as discrete optimization problems. A key challenge for score-based methods is the super-exponential combinatorial search space of DAGs w.r.t number of nodes (Chickering, 1996; Chickering et al., 2004).

Recently, Zheng et al. (2018) developed a continuous DAG learning approach using Langrange Multiplier methods and a differentiable DAG constraint based on the trace of the matrix exponential

of the weighted adjacency matrix. The resulting method, named NOTEARS, demonstrated superior performance in estimating linear DAGs with equal noise variances. Very recently, Zhang et al. (2022) and Bello et al. (2022) suggest that one main issue for NOTEARS and its derivatives, such as Yu et al. (2019), is gradient vanishing for linear DAG models with equal variance. They have thus proposed new continuous DAG constraints by based on geometric series of matrices as well as log-determinant of matrices.

In fact, many of the proposed Directed Acyclic Graph (DAG) constraints can be unified, as demonstrated in Wei et al. (2020). Wei et al. (2020) reveals that, for a $d \times d$ adjacency matrix, an order-$d$ polynomial of matrices is necessary and sufficient to enforce the DAG property. However, from a computational standpoint, computing general matrix polynomials can be challenging. Considering the fact that infinite-order polynomials that converge, i.e., power series, can give rise to analytic functions that are often simpler to evaluate than general polynomials, it prompts the question of whether analytic functions could be utilized in constructing DAG constraints. Furthermore, it raises the possibility of employing techniques commonly used for analyzing analytic functions in the investigation of continuous DAG constraints.

The answer is yes. We demonstrate that any analytic function within the class of functions denoted as $\mathcal{F} = \{f | f(x) = c_0 + \sum_{i=0}^{\infty} c_i x^i; c_i > 0, \forall i > 0; \lim_{i \to \infty} c_i / c_{i+1} > 0\}$ can be utilized to formulate Directed Acyclic Graph (DAG) constraints. In fact, the DAG constraints introduced in Zheng et al. (2018), Zhang et al. (2022), and Bello et al. (2022) can all be interpreted as being based on analytic functions from $\mathcal{F}$. Furthermore, we establish that the function class $\mathcal{F}$ remains closed under various function operators, including differentiation, function addition, and function multiplication. Leveraging this insight, we can construct novel DAG constraints based on pre-existing ones. Additionally, we can analyze the performance of these derived DAG constraints using techniques rooted in analytic functions.

## 2 PRELIMINARIES

**DAG and Linear SEM**    Given a directed acyclic graph (DAG) $\mathcal{G}$ defined over a random vector $\mathbf{x} = [x_1, x_2, \ldots, x_d]^\top$, the corresponding distribution $P(\mathbf{x})$ is assumed to satisfy the Markov assumption (Spirtes et al., 2000; Pearl, 2000). We consider $\mathbf{x}$ to follow a linear Structural Equation Model (SEM):

$$\mathbf{x} = \mathbf{B}^\top \mathbf{x} + \mathbf{e}. \tag{1}$$

Here, $\mathbf{B} \in \mathbb{R}^{d \times d}$ represents the weighted adjacency matrix that characterizes the DAG $\mathcal{G}$, and $\mathbf{e} = [e_1, e_2, \ldots, e_d]^\top$ represents the exogenous noise vector, comprising $d$ independent random variables. To simplify notation, we use $\mathcal{G}(\mathbf{B})$ to denote the graph induced by the weighted adjacency matrix $\mathbf{B}$, and we interchangeably use the terms 'random variables' and 'vertices' or 'nodes'.

We aim to estimate the DAG $\mathcal{G}$ from $n$ i.i.d. observational examples of $\mathbf{x}$, denoted by $\mathbf{X} \in \mathbb{R}^{n \times d}$. Generally, the DAG $\mathcal{G}$ can be identified only up to its Markov equivalence class under the faithfulness (Spirtes et al., 2000) or the sparsest Markov representation assumption (Raskutti and Uhler, 2018). It has been demonstrated that for linear SEMs with homoscedastic errors, where the noise terms are specified up to a constant (Loh and Bühlmann, 2013), and for linear non-Gaussian SEMs, where no more than one of the noise terms is Gaussian (Shimizu et al., 2006), the true DAG can be fully identified. In our study, we specifically focus on linear SEMs with equal noise variances (Peters and Bühlmann, 2013), where the scale of the data may be either known or unknown. When the scale is known, it is possible to fully recover the DAG. However, in the case of an unknown scale, the DAG may only be identified up to its Markov equivalence class.

**Continuous DAG learning**    In recent years, a series of continuous Directed Acyclic Graph (DAG) learning algorithms Bello et al. (2022); Ng et al. (2020); Zhang et al. (2022); Yu et al. (2021; 2019); Zheng et al. (2018) have been introduced, demonstrating superior performance when applied to linear Structural Equation Models (SEMs) with equal noise variances and known data scales. These methods can be expressed as follows:

$$\arg\min_{\mathbf{B}} S(\mathbf{B}, \mathbf{X}), \text{ s.t. } h(\mathbf{B}) = 0. \tag{2}$$

Here, $S$ is a scoring function, which can take the form of mean squared error (Zheng et al., 2018) or negative log-likelihood (Ng et al., 2020). The function $h$ is continuous and equals to 0 if and only if

the weighted adjacency matrix $\mathbf{B}$ defines a valid DAG. Previous approaches have employed various techniques, such as matrix exponential (Zheng et al., 2018), log-determinants (Bello et al., 2022), and polynomials (Zhang et al., 2022), to construct the function $h$. However, these methods are known to perform poorly, when applied to normalized data since they rely on scale information across variables for complete DAG recovery (Reisach et al., 2021).

## 3 ANALYTIC DAG CONSTRAINTS

In this section, we demonstrate that the diverse set of continuous DAG constraints proposed in previous work can be unified through the use of analytic functions. We begin by providing a brief introduction to analytic functions and then illustrate how they can be used to establish DAG constraints.

### 3.1 ANALYTIC FUNCTIONS AS DAG CONSTRAINTS

In mathematics, a power series

$$f(x) = c_0 + \sum_{i=1}^{\infty} c_i x^i, \tag{3}$$

which converges for $|x| < r = \lim_{i \to \infty} |c_i/c_{i+1}|$, defines an analytic function $f$ on the open interval $(-r, r)$, and $r$ is known as the radius of convergence. When we replace $x$ with a square matrix $\mathbf{A}$, we obtain an analytic function $f$ of a matrix as follows:

$$f(\mathbf{A}) = c_0 \mathbf{I} + \sum_{i=1}^{\infty} c_i \, \mathbf{A}^i, \tag{4}$$

where $\mathbf{I}$ is the identity matrix. Equation (4) would converge if the largest absolute value of the eigenvalues of $\mathbf{A}$, known as the spectral radius and denoted by $\rho(\mathbf{A})$, is smaller than $r$.

We are particularly interested in the following specific class of analytic functions

$$\mathcal{F} = \{f | f(x) = c_0 + \sum_{i=1}^{\infty} c_i x^i; \ \forall i > 0, c_i > 0; \lim_{i \to \infty} c_i/c_{i+1} > 0\}, \tag{5}$$

as any analytic function belonging to $\mathcal{F}$ can be applied to construct a continuous DAG constraint.

**Proposition 1.** *Let $\tilde{\mathbf{B}} \in \mathbb{R}_{\geq 0}^{d \times d}$ with $\rho(\tilde{\mathbf{B}}) < r$ be the weighted adjacency matrix of a directed graph $\mathcal{G}$, and let $f$ be an analytic function in the form of equation 4, where we further assume $\forall i > 0$ we have $c_i > 0$, then $\mathcal{G}$ is acyclic if and only if*

$$\mathrm{tr}\left[f(\tilde{\mathbf{B}})\right] = c_0 d. \tag{6}$$

An interesting property the DAG constraint equation 6 is that its gradients can also be represented as transpose of an analytic function as follows, which allows us to use analytic functions as the gradients of DAG constraints.

**Proposition 2.** *There exists some real number $r$, where for all $\{\tilde{\mathbf{B}} \in \mathbb{R}_{\geq 0}^{d \times d} | \rho(\tilde{\mathbf{B}}) < r\}$, the derivative of $\mathrm{tr}\left[f(\tilde{\mathbf{B}})\right]$ w.r.t. $\tilde{\mathbf{B}}$ is*

$$\nabla_{\tilde{\mathbf{B}}}\mathrm{tr}\left[f(\tilde{\mathbf{B}})\right] = \left[\nabla_x f(x)|_{x=\tilde{\mathbf{B}}}\right]^{\top}. \tag{7}$$

It is notable that for a $d \times d$ weighted adjacency matrix $\tilde{\mathbf{B}}$, an order-$d$ polynomial of $\tilde{\mathbf{B}}$ is sufficient and necessary to enforce DAGness (Wei et al., 2020; Ng et al., 2022). Meanwhile, evaluating matrix polynomials efficiently is highly nontrivial (Higham, 2008). For matrix analytic functions such as exponentials or logarithms, however, efficient algorithms exist (Higham, 2008).

The connection between matrix analytic functions and real analytic functions means that various properties of the matrix function can be obtained from a simple real-valued function. To pursue DAG constraints with better computational efficiency, we seek an analytic function whose derivative can be represented by itself to reduce the computation of different analytic functions. If a function has

such property, various intermediate results can be saved for future computation of gradients. The exponential function $\exp(x)$ with $\partial \exp(x)/\partial x = \exp(x)$, is a natural contender, and this leads to the well-known exponential-based DAG constraints (Zheng et al., 2018)

$$\textbf{Constraints: } \text{tr}\big[\exp(\tilde{\mathbf{B}})\big] = \sum_{i=0}^{\infty} \tilde{\mathbf{B}}^i/i! = d, \quad \textbf{Gradient: } \nabla_{\tilde{\mathbf{B}}}\exp(\tilde{\mathbf{B}}) = \exp(\tilde{\mathbf{B}})^{\top}, \quad (8)$$

which will converge for any $\tilde{\mathbf{B}}$.

Recently Bello et al. (2022) and Zhang et al. (2022) have suggested that exponential-based DAG constraints suffer from gradient vanishing. One cause of gradient vanishing arises from the small coefficients of high-order terms. The convergence radius for the exponential is $\infty$, that is $\lim_{i\to\infty} |c_i/c_{i+1}| = \lim_{i\to\infty} |(i+1)!/i!| = \infty$, which suggests that, compared to the lower order terms, the higher order terms contribute almost nothing in the DAG constraints, which indicates that it would not be efficient to prohibit possible long loops in candidate adjacency matrices.

Due to the fact that the adjacency matrix of a DAG must form a nilpotent matrix, whose spectural radius are acutally 0, naturally the spectral radius of candidate adjacency matrices would be close to 0. As a result, we do not need a function with infinite convergence radius. Instead, we can use an analytic function with finite convergence radius $r = \lim_{i\to\infty} |c_i/c_{i+1}| < \infty$. Thus by using a sequence $c_i$ with geometric progression $c_i = 1/s^{i-1}$ or harmonic-geometric progression $c_i = 1/(is^{i-1})$ we can obtain two analytic functions,

$$f_{inv}^s(x) = (s-x)^{-1} = \sum_{i=0}^{\infty} x^i/s^{i-1}, \quad f_{log}^s(x) = -s\log(s-x) = \sum_{i=1}^{\infty} \frac{x^i}{is^{i-1}} - s\log s. \quad (9)$$

Then by our Proposition 1 and Proposition 2, two dag constraints can be obtained as follows:

$$\textbf{Constraints: } \text{tr} f_{inv}^s(\tilde{\mathbf{B}}) = d, \quad \textbf{Gradient: } \nabla_{\tilde{\mathbf{B}}}\text{tr} f_{inv}^s(\tilde{\mathbf{B}}) = [f_{inv}^s(\tilde{\mathbf{B}})^2]^{\top}, \quad (10a)$$

$$\textbf{Constraints:} \text{tr} f_{log}^s(\tilde{\mathbf{B}}) = 0, \quad \textbf{Gradient: } \nabla_{\tilde{\mathbf{B}}}\text{tr} f_{log}^s(\tilde{\mathbf{B}}) = [f_{inv}^s(\tilde{\mathbf{B}})]^{\top}, \quad (10b)$$

where a truncated version of $f_{inv}^s$ is applied in Zhang et al. (2022), and the $f_{log}^s$ based constraints are equivalent to those in Bello et al. (2022). One key difference between Zhang et al. (2022); Bello et al. (2022) and the exponential-based DAG constraints (Zheng et al., 2018) is their finite convergence radius, which requires an additional constraints $\rho(\tilde{\mathbf{B}}) < s$. Meanwhile, the adjacency matrix of a DAG must be nilpotent, and thus its spectral radius must be 0. In this case, such additional constraints would not affect the feasible set.

## 3.2 CONSTRUCTING DAG CONSTRAINTS BY FUNCTIONAL OPERATOR

One can easily observe a coincidence between $f_{log}$ and $f_{inv}$ as follows,

$$\frac{\partial f_{log}^s(x)}{\partial x} \propto f_{inv}^s(x), \ f_{log}^s(x) \propto \int_{-\infty}^{x} f_{inv}^s(t)\mathbf{d}t + C, \quad (11)$$

which suggests that it may be possible to derive a group of DAG constraints from an analytic function by applying integration or differentiation. This is because derivatives of any order of an analytic function is also analytic. More formally, if a function is analytic at some point $x_0$, then its $n^{\text{th}}$ derivative for any integer $n$ exists and is also analytic at $x_0$. Thus we can derive DAG constraints from any $f \in \mathcal{F}$ as follows.

**Proposition 3.** *Let $f(x) = c_0 + \sum_{i=1}^{\infty} c_i x^i \in \mathcal{F}$ be analytic on $(-r, r)$, and let $n$ be arbitary integer larger than 1, then $\tilde{\mathbf{B}} \in \mathbb{R}_{\geq 0}^{d \times d}$ with spectral radius $\rho(\hat{\mathbf{B}}) \leqslant r$ forms a DAG if and only if*

$$\text{tr}\left[\frac{\partial^n f(x)}{\partial x^n}\bigg|_{x=\tilde{\mathbf{B}}}\right] = n!c_n. \quad (12)$$

The above proposition suggests that the differential operator can be applied to an analytic function to form a new DAG constraints. Besides the differential operator, the addition and multiplication of analytic functions can also be applied to generate new DAG constraints. That is

**Proposition 4.** *Let $f_1(x) = c_0^1 + \sum_{i=1}^\infty c_i^1 x^i \in \mathcal{F}$, and $f_2(x) = c_0^2 + \sum_{i=1}^\infty c_i^2 x^i \in \mathcal{F}$. Then for an adjancency matrix $\tilde{\mathbf{B}} \in \mathbb{R}_{\geqslant 0}^{d \times d}$ with spectral radius $\rho(\tilde{\mathbf{B}}) \leqslant \min(\lim_{i\to\infty} c_i^1/c_{i+1}^1, \lim_{i\to\infty} c_i^2/c_{i+1}^2)\}$, the following three statements are equivalent: 1) $\tilde{\mathbf{B}}$ forms a DAG; 2) $\operatorname{tr}[f_1(\tilde{\mathbf{B}}) + f_2(\tilde{\mathbf{B}})] = (c_0^1 + c_0^2)d$; 3) $\operatorname{tr}[f_1(\tilde{\mathbf{B}})f_2(\tilde{\mathbf{B}})] = c_0^1 c_0^2 d$.*

Particularly for $f_{log}^s(x)$ and $f_{inv}^s(x)$, due to the specific property of $f_{inv}^s(x)$, we have

$$\frac{\partial^{n+1} f_{log}^s(x)}{\partial x^{n+1}} = \frac{\partial^n f_{inv}^s(x)}{\partial x^n} \propto (s-x)^{-(n+1)} = [f_{inv}^s(x)]^{n+1}, \tag{13}$$

which implies that the $n^{\text{th}}$ derivative of function $1/(s-x)$ is propositional to the the order-$(n+1)$ power of $1/(s-x)$. Using this property, the value of $(\mathbf{I} - \tilde{\mathbf{B}}/s)^{-1}$ can be cached and then used to generate a series of DAG constraints as well as their gradients. Similarly, the value of matrix exponential $\exp((\tilde{\mathbf{B}})/s)$ can also be cached during the evaluation of DAG constraints to accelerate the computation. Furthermore, the gradients of the DAG constraints will also increase as $n$ increases.

**Proposition 5.** *Let $n$ be any positive integer, the adjacency matrix $\tilde{\mathbf{B}} \in \{\hat{\mathbf{B}} \in \mathbb{R}_{\geqslant 0}^{d \times d} | \rho(\hat{\mathbf{B}}) < s\}$ forms a DAG if and only if*

$$\operatorname{tr}\left[(\mathbf{I} - \tilde{\mathbf{B}}/s)^{-n}\right] = d.$$

*Furthermore, the gradients of the DAG constraints satisfies that $\forall \tilde{\mathbf{B}} \in \{\hat{\mathbf{B}} \in \mathbb{R}_{\geqslant 0}^{d \times d} | \rho(\hat{\mathbf{B}}) < s\}$*

$$\|\nabla_{\tilde{\mathbf{B}}} \operatorname{tr}(\mathbf{I} - \tilde{\mathbf{B}}/s)^{-n}\| \leqslant \|\nabla_{\tilde{\mathbf{B}}} \operatorname{tr}(\mathbf{I} - \tilde{\mathbf{B}}/s)^{-n-k}\|,$$

*where $k$ is an arbitrary positive integer, and $\|\cdot\|$ denote an arbitrary matrix norm induced by vector $p$-norm.*

**Gradient Vanishing and Numeric Stability**   For the series of DAG constraints constructed from Equation (13), as gradient vanishing is one of the main challenges for differentiable DAG learning, according to Proposition 5 we may prefer larger $n$ to achieve better performance in practice. Furthermore, choosing a smaller $s$ may also help to amplify the gradient of DAG constraints. Therefore, Bello et al. (2022) applied an annealing strategy on $s$ to improve performance, while Zhang et al. (2022) used a fixed $s = 1.0$ in their implementation. However, in practice, especially when incorporating the DAG constraints with first-order optimizers, the spectral radius of the candidate $\tilde{\mathbf{B}}$ can often be larger than $s$. Bello et al. (2022) applied a simple heuristics to search for the proper $s$, while Zhang et al. (2022) truncated the power series to avoid numerical issues in higher-order terms. However, in practice, we observed that Zhang et al. (2022) encountered some numerical issues for large graphs, and the simple heuristics used by Bello et al. (2022) may result in a sacrifice in performance. Based on our analysis, if $\tilde{\mathbf{B}}$ goes out, it can be verified by checking if the power series $\sum_{i=0}^\infty (\tilde{\mathbf{B}}/s)^i$ converges to $(\mathbf{I} - \tilde{\mathbf{B}}/s)^{-1}$, and $s$ can be chosen based on the spectral radius of $\tilde{\mathbf{B}}$.

**Efficiently Computation**   The specific structure of the power series $\sum_{i=0}^\infty (\tilde{\mathbf{B}}/s)^i$ allows for fast evaluation. Let

$$\mathbf{L}_t = \sum_{i=0}^t (\tilde{\mathbf{B}}/s)^i, \tag{14}$$

then it is evident that

$$\mathbf{L}_{2t} = \mathbf{L}_t + (\tilde{\mathbf{B}}/s)^t \mathbf{L}_t, \tag{15}$$

which indicates that the term $\mathbf{L}_t$ can be obtained with $\mathcal{O}(\log t)$ time complexity. Furthermore, using Equation (13), the gradient of $\operatorname{tr}(\mathbf{I} - \tilde{\mathbf{B}}/s)^{-n}$ can also be easily derived from $\mathbf{L}_\infty$. Along with the strategy for searching $s$, we can use Algorithm 1 to efficiently compute the DAG constraints.

### 3.3 OVERALL OPTIMIZATION FRAMEWORK

The DAG constraints above are applicable only to positive adjacency matrices, so we use the Hadamard product to map a real adjacency matrix to a positive one. Thus Equation (2) becomes:

$$\arg\min_{\mathbf{B}} S(\mathbf{B}, \mathbf{X}), \quad \text{s.t. } \operatorname{tr} f(\mathbf{B} \odot \mathbf{B}) = c_0 d, \ \rho(\mathbf{B} \odot \mathbf{B}) < r, \tag{16}$$

where the analytic function $f(x) = c_0 + \sum_{i=1}^{\infty} x^i \in \mathcal{F}$, and $\odot$ denotes the Hadamard product.

| **Algorithm 1** Efficient Evaluation of Gradients | **Algorithm 2** Path following algorithm |
|---|---|
| **Input:** $\tilde{\mathbf{B}}, s, d, \epsilon > 0, \xi > 0$ 
 **Output:** $\nabla_{\tilde{\mathbf{B}}} \mathrm{tr} f_{\log}^s(\tilde{\mathbf{B}})$ or $\nabla_{\tilde{\mathbf{B}}} \mathrm{tr} \left[ f_{inv}^s(\tilde{\mathbf{B}}) \right]^n$ | **Input:** $\mathbf{X} \in \mathbb{R}^{n \times d}; \quad S; f \in \mathcal{F}; \lambda_1; \mu_0; \alpha \in (0, 1);$ 
 $T_{outer}; T_{inner}; \gamma > 0$ 
 **Output:** Estimated $\mathbf{B}$ |

| | |
|---|---|
| 1: $\mathbf{D} \leftarrow \mathbf{I} + \tilde{\mathbf{B}}/s$ , $\mathbf{W} \leftarrow \tilde{\mathbf{B}}/s$ | 1: $i \leftarrow 0, \mu \leftarrow \mu_0, \mathbf{B}_0 = \mathbf{0}$ |
| 2: $k = 1$ | 2: **for** $i = 0; i < T_{outer}; i + +$ **do** |
| 3: **while** $\|\mathbf{D}(\mathbf{I} - \tilde{\mathbf{B}}) - \mathbf{I}\| > \epsilon$ and $k < 2d$ **do** | 3: $\quad \mathbf{B}_{i+1} \leftarrow \mathbf{B}_i$ $\qquad \triangleright$ Optimize over |
| 4: $\quad \mathbf{W} \leftarrow \mathbf{W} \times \mathbf{W}$ | $\quad \mu[S(\mathbf{B}, \mathbf{X}) + \lambda_1 \|\mathbf{B}\|_1] + 1/2 f(\mathbf{B} \odot \mathbf{B})$ |
| 5: $\quad \mathbf{D} \leftarrow \mathbf{D} \times (\mathbf{W} + \mathbf{I})$ | 4: $\quad$ **for** $j = 0; j < T_{inner}; j + +$ **do** |
| 6: $\quad k \leftarrow 2k$ | 5: $\quad\quad \tilde{\mathbf{B}} \leftarrow \mathbf{B}_{i+1} \odot \mathbf{B}_{i+1}$ |
| 7: **end while** | 6: $\quad\quad \mathbf{B}_{i+1} \leftarrow \mathbf{B}_{i+1} - \gamma\mu[\nabla_{\mathbf{B}} S(\mathbf{B}, \mathbf{X}) +$ |
| 8: **if** $\|\mathbf{D}(\mathbf{I} - \tilde{\mathbf{B}}) - \mathbf{I}\| > \epsilon$ **then** | $\quad \lambda_1 \mathrm{sign}(\mathbf{B})] - \gamma\nabla_{\tilde{\mathbf{B}}} f(\tilde{\mathbf{B}}) \odot \mathbf{B}_{i+1}$ |
| 9: $\quad s \leftarrow \rho(\tilde{\mathbf{B}}) + \xi$, goto line 1 | 7: $\quad$ **end for** |
| 10: **else** | 8: $\quad \mu \leftarrow \mu \times \alpha$ |
| 11: $\quad$ For $f_{\log}^s$ return $\mathbf{D}^\top / s$ | 9: $\quad \hat{\mathbf{B}} \leftarrow \mathbf{B}_{i+1}$ |
| 12: $\quad$ For $[f_{inv}^s(\tilde{\mathbf{B}})]^n$ return $n[\mathbf{D}^\top / s]^{n+1}$ | 10: **end for** |
| 13: **end if** | 11: **Return** $\hat{\mathbf{B}}$ |

In our work, we choose to use the path-following approach with an $\ell_1$ regularizer, as in Bello et al. (2022). This is because in the Lagrange approaches applied in Zhang et al. (2022); Yu et al. (2021); Zheng et al. (2018); Yu et al. (2019), the Lagrangian multiplier must be set to very large value to enforce DAGness, which may result in numerical instability. In the path-following approach, instead of using large Lagrangian multipliers, a small coefficients are added to the score function $S$ as follows[1]

$$\arg\min_{\mathbf{B}} \ \mu[S(\mathbf{B}, \mathbf{X}) + \lambda_1 \|\mathbf{B}\|_1] + \mathrm{tr} f(\mathbf{B} \odot \mathbf{B}), \quad \text{s.t.} \quad \rho(\mathbf{B} \odot \mathbf{B}) < r, \tag{17}$$

where $\lambda_1$ is the user-specified weight for the $\ell_1$ regularizer. For the additional constraints $\rho(\mathbf{B} \odot \mathbf{B}) < r$, with properly chosen initial value and step-length, it can usually be satisfied. Also it is notable that $\|\mathbf{B}\|_1 < r$ is a sufficient condition for $\rho(\mathbf{B} \odot \mathbf{B}) < r$, and thus the sparsity constraints also encourage this condition to be satisfied. Based on Bello et al. (2022), we implemented a path-following shown in Algorithm 2.

The optimization model equation 17 is observed to perform well for linear Gaussian SEMs with equal variance as well as other equal variance SEMs. Meanwhile, for unequal variance, or normalized data from linear Gaussian SEMs with equal variance where the scale information are missing, MSE score function is not consistent and often provides misleading information about the underlying DAG. Additionally, as observed by Ng et al. (2023), the initialization of adjacency matrices in cases of unequal variance can significantly affect performance, suggesting that non-convexity may pose a serious challenge in such scenarios.

## 4 NON-CONVEXITY ANALYSIS OF ANALYTIC DAG CONSTRAINTS

The non-convexity of a function can be analyzed through the analysis of its Hessian. Particularly for our analytic DAG constraints, its Hessian can be obtained using the following proposition and then the non-convexity can be analyzed by analysis the spectral radius of the Hessian.

**Proposition 6.** *The Hessian of DAG constraints equation 6 can be obtained as follows:*

$$\nabla_{\tilde{\mathbf{B}}}^2 \mathrm{tr} f(\tilde{\mathbf{B}}) = \mathbf{K}_{dd} \sum_{i=2}^{\infty} i c_i \sum_{j=0}^{i-2} \left[ \tilde{\mathbf{B}}^j \right]^\top \otimes \left[ \tilde{\mathbf{B}}^{i-2-j} \right], \tag{18}$$

*where $\otimes$ denotes the Kronecker product, and $\mathbf{K}_{dd} \in \{0, 1\}^{d^2 \times d^2}$ is the commutation matrix satisfies that for any $d \times d$ matrix $\mathbf{A}$*

$$\mathbf{K}_{d,d} \mathrm{vec}(\mathbf{A}) = \mathrm{vec}(\mathbf{A}^\top).$$

---

[1] the constant $c_0 d$ can be dropped because $\mathrm{tr} f(\mathbf{B} \odot \mathbf{B})$ is bounded below by $c_0 d$, detailed derivation is provided in the supplementary file.

Obviously, the Hessian Equation (18) is symmetric and not positive semi-definite. One widely used way to convexify Hessian is to find a positive scalar $\eta$ such that

$$\Delta = \nabla_{\tilde{\mathbf{B}}}^2 \operatorname{tr} f(\tilde{\mathbf{B}}) + \eta \mathbb{I}, \tag{19}$$

becomes positive semi-definite. It require $\eta$ to be no less than the absolute value of the most negative eigenvalue of $\nabla_{\tilde{\mathbf{B}}}^2 \operatorname{tr} f(\tilde{\mathbf{B}})$. Here the Hessian are symmetric matrix with all non-negative entries. For this kind of matrices the absolute value of the most negative eigenvalue of $\nabla_{\tilde{\mathbf{B}}}^2 \operatorname{tr} f(\tilde{\mathbf{B}})$ is upper bounded by the spectral radius of the Hessian, and the bound is tight under certain conditions (Spielman, 2012). Thus it would be nature to use the spectral radius of the Hessian to measure the level of non-convexity of the analytic DAG constraints.

The Hessian $\nabla_{\tilde{\mathbf{B}}}^2 \operatorname{tr} f(\tilde{\mathbf{B}})$ can be viewed as linear combinations of a series of symmetric matrices $i\mathbf{K}_{dd} \sum_{j=0}^{i-2} \left[\tilde{\mathbf{B}}^j\right]^\top \otimes \left[\tilde{\mathbf{B}}^{i-2-j}\right]$ with all non-negative entries. The commutation matrix $\mathbf{K}_{dd}$ (Magnus and Neudecker, 1979) would not have any effects on the spectral radius as it is orthonormal. Thus larger $c_i$ would result the spectral radius of a single term $ic_i\mathbf{K}_{dd} \sum_{j=0}^{i-2} \left[\tilde{\mathbf{B}}^j\right]^\top \otimes \left[\tilde{\mathbf{B}}^{i-2-j}\right]$ to increase, and finally lead the spectral radius of the Hessian to increase as the following proposition.

**Proposition 7.** *For two analytic function $f_1(x) = c_{0,1} + \sum_{i=1}^\infty c_{i,1} x^i$ and $f_2(x) = c_{0,2} + \sum_{i=1}^\infty c_{i,2} x^i$, if $\forall i \geqslant 1$ we have $c_{i,1} \geqslant c_{i,2} > 0$, then*

$$\rho(\nabla_{\tilde{\mathbf{B}}}^2 \operatorname{tr} f_1(\tilde{\mathbf{B}})) \geqslant \rho(\nabla_{\tilde{\mathbf{B}}}^2 \operatorname{tr} f_2(\tilde{\mathbf{B}})), \tag{20}$$

*where $\rho(\cdot)$ denotes the spectral radius of a matrix.*

Proposition 7 suggests that the spectral radius of the Hessian would increase if the coefficients $c_i$ in the analytic function increase. This implies that DAG constraints with larger $c_i$ may gain benefits from gradient vanishing, but suffer from non-convexity. In fact, using Proposition 7 it would be straightforward to get the following corollary, which provides the level of non-convexity comparison for several DAG constraints.

**Corollary 8.**

$$\rho(\nabla_{\tilde{\mathbf{B}}}^2 \operatorname{tr} \exp(\tilde{\mathbf{B}})) \leqslant \rho(\nabla_{\tilde{\mathbf{B}}}^2 \operatorname{tr} f_{\log}^s(\tilde{\mathbf{B}})) \leqslant \rho(\nabla_{\tilde{\mathbf{B}}}^2 \operatorname{tr} f_{inv}^s(\tilde{\mathbf{B}})). \tag{21}$$

The optimization problem equation 17 can be viewed as a convex objective plus one non-convex constraint, and the convex mean square error (MSE) loss may play different roles in different scenarios. For data with known scale, the MSE loss is consistent and thus it provides enough information to identify the underlying model and thus the non-convexity may not be a serious issue. This is because in the path-following optimization framework (provided in Algorithm 2), at the beginning the optimization direction are dominated by the MSE loss so that it will push the candidates to a point that is not far from global optimal. Thus DAG constraints with finite convergence radius is preferred to escape from gradient vanishing. Meanwhile, for DAG learning problem with unknown scale, the MSE loss may not be very informative to the underlying graph structure. In this case, the highly non-convex DAG constraints may lead to the optimizer to get trapped into a local minimum easily, and thus we may need additional constraints to reduce the search space, which may possibly make the objective flatter. In our experiments, we find that by allowing only edges to exist between nodes with strong correlation can significantly improve the performance.

## 5 EXPERIMENTS

In the experiment, we compared the performance of different analytic DAG constraints in the same path-following optimization framework. We implemented the path-following algorithm (provided in Algorithm 2) using cupy and numpy based on the path-following optimizer in Bello et al. (2022). For analytic DAG constraints with infinite convergence radius, we consider the exponential-based DAG constraints. For analytic DAG constraints with finite convergence radius, we consider the following 4 different DAG constraints generated by the differentiation operator or multiply operator:

- Order-1: $\operatorname{tr} f_{\log}^s(\mathbf{B} \odot \mathbf{B}) = 0$;

| Graphs | #Nodes | DAGMA | Order 1 | Order 2 | Order 3 | Order 4 |
|---|---|---|---|---|---|---|
| ER2-Gaussian | 500 | $44.90 \pm 32.95$ | $33.40 \pm 23.46$ | $31.70 \pm 19.47$ | $30.60 \pm 19.07$ | $\mathbf{29.80 \pm 20.97}$ |
| | 1000 | $94.80 \pm 35.80$ | $69.60 \pm 27.64$ | $55.60 \pm 19.13$ | $\mathbf{52.40 \pm 19.86}$ | $57.30 \pm 21.39$ |
| | 2000 | $235.40 \pm 62.76$ | $176.00 \pm 47.77$ | $153.30 \pm 35.92$ | $135.60 \pm 38.91$ | $\mathbf{131.00 \pm 28.65}$ |
| ER3-Gaussian | 500 | $125.30 \pm 44.55$ | $101.10 \pm 39.03$ | $\mathbf{90.30 \pm 39.56}$ | $93.90 \pm 31.26$ | $92.90 \pm 45.56$ |
| | 1000 | $339.60 \pm 67.80$ | $242.80 \pm 72.21$ | $210.30 \pm 60.98$ | $184.90 \pm 47.44$ | $\mathbf{165.80 \pm 35.95}$ |
| | 2000 | $669.50 \pm 140.61$ | $610.70 \pm 136.84$ | $555.30 \pm 106.01$ | $479.50 \pm 88.72$ | $\mathbf{424.90 \pm 64.39}$ |
| ER4-Gaussian | 500 | $307.60 \pm 116.53$ | $261.40 \pm 102.81$ | $263.00 \pm 122.34$ | $246.70 \pm 110.28$ | $\mathbf{223.80 \pm 97.46}$ |
| | 1000 | $878.50 \pm 174.96$ | $689.20 \pm 165.62$ | $695.50 \pm 134.41$ | $\mathbf{619.80 \pm 150.43}$ | $626.30 \pm 157.59$ |
| | 2000 | $1922.30 \pm 187.69$ | $1785.40 \pm 184.47$ | $1779.30 \pm 211.23$ | $1655.40 \pm 181.75$ | $\mathbf{1574.10 \pm 152.23}$ |
| ER2-Exp | 500 | $58.20 \pm 31.58$ | $40.50 \pm 26.93$ | $\mathbf{28.90 \pm 16.60}$ | $31.00 \pm 25.67$ | $35.20 \pm 34.32$ |
| | 1000 | $93.90 \pm 33.96$ | $68.70 \pm 23.20$ | $54.00 \pm 16.26$ | $\mathbf{50.90 \pm 17.29}$ | $57.90 \pm 24.93$ |
| ER3-Exp | 500 | $142.70 \pm 50.13$ | $106.00 \pm 39.15$ | $\mathbf{95.10 \pm 32.68}$ | $99.60 \pm 37.94$ | $100.30 \pm 47.52$ |
| | 1000 | $321.10 \pm 83.82$ | $242.80 \pm 68.51$ | $212.40 \pm 67.87$ | $187.60 \pm 61.87$ | $\mathbf{173.10 \pm 49.03}$ |
| ER4-Exp | 500 | $336.00 \pm 124.19$ | $292.70 \pm 123.41$ | $294.90 \pm 130.66$ | $254.40 \pm 133.05$ | $\mathbf{214.70 \pm 84.29}$ |
| | 1000 | $879.40 \pm 162.98$ | $718.20 \pm 127.12$ | $710.60 \pm 151.41$ | $640.50 \pm 148.24$ | $\mathbf{619.70 \pm 133.44}$ |
| ER2-Gumbel | 500 | $45.10 \pm 33.28$ | $22.60 \pm 20.04$ | $21.30 \pm 18.41$ | $19.80 \pm 16.39$ | $\mathbf{16.20 \pm 11.91}$ |
| | 1000 | $80.50 \pm 42.65$ | $49.90 \pm 24.54$ | $39.90 \pm 14.04$ | $\mathbf{36.80 \pm 15.18}$ | $45.90 \pm 23.18$ |
| ER3-Gumbel | 500 | $147.10 \pm 54.19$ | $94.10 \pm 40.87$ | $76.60 \pm 60.77$ | $\mathbf{60.60 \pm 31.34}$ | $85.30 \pm 50.75$ |
| | 1000 | $297.90 \pm 72.40$ | $215.40 \pm 52.35$ | $185.00 \pm 71.98$ | $173.90 \pm 57.09$ | $\mathbf{147.70 \pm 40.51}$ |
| ER4-Gumbel | 500 | $338.80 \pm 127.56$ | $273.70 \pm 131.13$ | $257.50 \pm 111.06$ | $\mathbf{232.40 \pm 121.98}$ | $234.70 \pm 149.59$ |
| | 1000 | $919.90 \pm 182.38$ | $722.80 \pm 177.86$ | $734.80 \pm 177.78$ | $620.70 \pm 187.56$ | $\mathbf{564.10 \pm 170.46}$ |

Table 1: DAG learning performance (measured in structural hamming distance, the lower the better, best results in **bold**) of different algorithms on ER{2,3,4} graphs with different noise distributions. All our algorithms performs better than the previous state-of-the-arts DAGMA (Bello et al., 2022), and as higher order DAG constraints suffers less to gradient vanishing, it tends to have better performance.

| Graphs | #Nodes | DAGMA(Bello et al., 2022) | Order 1 | Order 2 | Order 3 | Order 4 |
|---|---|---|---|---|---|---|
| SF2 | 500 | $31.40 \pm 43.51$ | $\mathbf{24.30 \pm 43.90}$ | $32.40 \pm 49.38$ | $34.20 \pm 45.56$ | $41.50 \pm 48.45$ |
| | 1000 | $44.90 \pm 34.38$ | $41.20 \pm 36.02$ | $\mathbf{22.50 \pm 13.21}$ | $29.20 \pm 20.07$ | $58.10 \pm 27.58$ |
| | 2000 | $189.80 \pm 99.47$ | $162.90 \pm 73.30$ | $172.10 \pm 74.35$ | $\mathbf{152.20 \pm 90.29}$ | $172.60 \pm 124.12$ |
| SF3 | 500 | $58.10 \pm 33.90$ | $51.10 \pm 32.10$ | $\mathbf{41.10 \pm 17.91}$ | $49.80 \pm 24.58$ | $71.00 \pm 23.46$ |
| | 1000 | $169.40 \pm 60.82$ | $\mathbf{158.10 \pm 46.70}$ | $161.20 \pm 55.25$ | $162.50 \pm 57.54$ | $195.40 \pm 75.60$ |
| | 2000 | $928.70 \pm 148.70$ | $\mathbf{896.00 \pm 101.85}$ | $897.10 \pm 146.78$ | $891.50 \pm 143.40$ | $999.70 \pm 206.36$ |
| SF4 | 500 | $131.20 \pm 42.63$ | $136.80 \pm 41.71$ | $134.40 \pm 39.34$ | $\mathbf{128.90 \pm 36.68}$ | $151.60 \pm 37.05$ |
| | 1000 | $431.70 \pm 119.22$ | $404.00 \pm 88.89$ | $400.30 \pm 76.49$ | $\mathbf{386.90 \pm 93.16}$ | $394.50 \pm 111.57$ |
| | 2000 | $1525.10 \pm 299.02$ | $1500.50 \pm 297.88$ | $1444.70 \pm 291.45$ | $\mathbf{1395.60 \pm 264.90}$ | $1418.90 \pm 228.86$ |
| SF2 | 500 | $25.90 \pm 44.45$ | $\mathbf{23.40 \pm 44.41}$ | $32.10 \pm 49.08$ | $35.00 \pm 48.84$ | $37.20 \pm 45.21$ |
| | 1000 | $43.70 \pm 34.48$ | $41.20 \pm 36.02$ | $32.00 \pm 34.13$ | $\mathbf{29.10 \pm 19.68}$ | $59.10 \pm 30.34$ |
| SF3 | 500 | $57.70 \pm 33.68$ | $57.70 \pm 33.64$ | $\mathbf{41.80 \pm 20.37}$ | $43.20 \pm 15.75$ | $66.70 \pm 24.36$ |
| | 1000 | $177.10 \pm 67.53$ | $\mathbf{165.40 \pm 57.70}$ | $171.60 \pm 66.80$ | $175.10 \pm 69.09$ | $195.90 \pm 80.37$ |
| SF4 | 500 | $\mathbf{127.50 \pm 40.84}$ | $132.80 \pm 40.39$ | $129.90 \pm 42.07$ | $135.50 \pm 44.21$ | $152.40 \pm 39.94$ |
| | 1000 | $408.80 \pm 119.71$ | $419.70 \pm 108.01$ | $\mathbf{388.50 \pm 53.01}$ | $394.30 \pm 88.95$ | $395.30 \pm 109.37$ |
| SF2 | 500 | $23.10 \pm 44.78$ | $17.70 \pm 44.51$ | $\mathbf{16.70 \pm 44.15}$ | $20.00 \pm 45.75$ | $33.40 \pm 49.30$ |
| | 1000 | $29.20 \pm 24.77$ | $24.70 \pm 24.85$ | $\mathbf{12.50 \pm 11.40}$ | $16.20 \pm 13.96$ | $47.90 \pm 25.69$ |
| SF3 | 500 | $33.50 \pm 32.98$ | $25.20 \pm 27.85$ | $19.40 \pm 12.37$ | $\mathbf{19.00 \pm 7.44}$ | $50.00 \pm 22.41$ |
| | 1000 | $107.50 \pm 50.50$ | $114.50 \pm 59.80$ | $106.60 \pm 64.77$ | $\mathbf{103.70 \pm 58.15}$ | $133.60 \pm 88.43$ |
| SF4 | 500 | $77.70 \pm 41.43$ | $76.20 \pm 41.86$ | $\mathbf{67.90 \pm 26.47}$ | $79.20 \pm 23.87$ | $101.40 \pm 22.37$ |
| | 1000 | $333.10 \pm 118.06$ | $348.80 \pm 110.93$ | $\mathbf{309.20 \pm 51.86}$ | $321.50 \pm 83.13$ | $339.70 \pm 111.17$ |

Table 2: DAG learning performance (measured in structural hamming distance, the lower the better, best results in **bold**) of different algorithms on SF{2,3,4} graphs with different noise distributions. Our algorithms usually performs better than the previous state-of-the-arts DAGMA(Bello et al., 2022).

- Order-2: $\mathrm{tr} f_{inv}^s(\mathbf{B} \odot \mathbf{B} / s) = d$;
- Order-3: $\mathrm{tr}[f_{inv}^s(\mathbf{B} \odot \mathbf{B} / s)]^2 = d$;
- Order-4: $\mathrm{tr}[f_{inv}^s(\mathbf{B} \odot \mathbf{B} / s)]^3 = d$.

In our experiments, we use the same annealing strategy for $s$ as Bello et al. (2022). During the optimization, the spectral radius of $\mathbf{B} \odot \mathbf{B}$ may be larger than $s$, which make the DAG constraints

| | PC | GES | DAGMA | Exponential | Order 1 | Order 2 | Order 3 | Order 4 |
|---|---|---|---|---|---|---|---|---|
| SHD | $563.9 \pm 23.84$ | $4490.2 \pm 62.52$ | $588.8 \pm 18.33$ | $488.6 \pm 24.29$ | $429.6 \pm 24.73$ | $410.6 \pm 15.25$ | $401.0 \pm 16.64$ | $\mathbf{389.4 \pm 16.70}$ |

Table 3: DAG learning performance (measured in structural hamming distance, the lower the better, best results in **bold**) of different algorithms on 1000-node ER1 graphs with Gaussian noise with observation data normalized. Our algorithms performs better than the previous approaches, and as higher order DAG constraints suffers less to gradient vanishing, it tends to have better performance. We compare differential DAG learning approaches with conditional independent test based PC (Spirtes and Glymour, 1991) algorithm and score based GES (Chickering, 2002) algorithm. The result is reported in the format of average$\pm$ standard derivation gathered from 10 different simulations.

invalid. In this case, we use the strategy provided in Algorithm 1 to reset $s$. We also tried the DAG constraints (Zhang et al., 2022) with their code provided in their appendix, but it do have some numeric issues for large scale graphs.

We compare the performance of these DAG constraints using two different settings: linear SEM with known ground truth scale and with unknown ground truth scale. We also compare these methods with constraint based PC (Spirtes and Glymour, 1991) algorithm and score based combinatorial search algorithm GES (Chickering, 2002) implemented by Kalainathan et al. (2020).

## 5.1 LINEAR SEM WITH KNOWN GROUND TRUTH SCALE

For linear SEM with a known ground truth scale, our experimental setting is similar to Bello et al. (2022); Zhang et al. (2022); Zheng et al. (2018). We generated two different types of random graphs: ER (Erdős-Rényi) and SF (Scale-Free) graphs with different numbers of expected edges. We use ER$n$ (SF$n$) to denote graphs with $d$ nodes and $nd$ expected edges. Edge weights generated from a uniform distribution over the union of two intervals $[-2, -0.5] \cup [0.5, 2.0]$ are assigned to each edge to form a weighted adjacency matrix $\mathbf{B}$. Then, $n$ samples are generated from the linear SEM $\mathbf{x} = \mathbf{B}\,\mathbf{x} + \mathbf{e}$ to form an $n \times d$ data matrix $\mathbf{X}$, where the noise $\mathbf{e}$ is iid sampled from Gaussian, Exponential, or Gumbel distribution. As Bello et al. (2022); Zhang et al. (2022); Zheng et al. (2018) achieved nearly perfect results on small and sparse graphs, we considered more challenging large and denser graphs in our experiments. We set the sample size $n = 1000$ and consider 3 different numbers of nodes $d = 500, 1000, 2000$. For each setting, we conducted 10 random simulations to obtain an average performance. All these experiments were performed using an A100 GPU, and all computations were done in double precision. Our algorithms were compared with the previous state-of-the-art approach DAGMA Bello et al. (2022). The original version of DAGMA Bello et al. (2022) used numpy and ran on CPU; we replaced numpy with cupy to get a GPU version of DAGMA, which performed identically to the CPU version.

The results on ER2, ER3, and ER4 graphs are shown in Table 1. In all cases, our algorithms outperformed the previous state-of-the-art DAGMA. Our Order-1 algorithm is very similar to DAGMA, except for our annealing strategy of $s$ derived from our theory, which indicates the efficiency of our theory. Furthermore, our theory shows that higher-order constraints suffer less from gradient vanishing, and in the experimental results, we observed that the performance of higher-order DAG constraints outperformed lower-order ones in most cases. The results of SF2, SF3, and SF4 graphs are shown in Table 2. On scale-free graphs, our algorithms usually performed better than DAGMA, and the higher-order constraints, Order-2 and Order-3, often outperformed Order-1. The performance of Order-4 constraints was not good, possibly due to stronger non-convexity.

The DAGMA algorithm actually employed the same DAG constraints as our Order-1 method, but with a different strategy to search for $s$. Our search strategy, derived from properties of analytic functions, provides a tight bound for $s$, allowing a smaller $s$ to be used than DAGMA without sacrificing the numeric stability. As a result, our algorithm suffers less from gradient vanishing and achieves better performance.

In terms of running time, all algorithms had similar running times, typically about 5 minutes for a 500-node graph, 10-20 minutes for a 1000-node graph, and around 2 hours for a 2000-node graph. Due to limited time and resources, we only considered $d = 2000$ for Gaussian noises, and for other cases, we only considered $d = 500, 1000$.

|  | Original GRAN | Order 1 | Order 2 | Order 3 | Order 4 |
|---|---|---|---|---|---|
| SHD | 15 | 13 | 13 | 13 | 13 |
| SHD-CPDAG | 10 | 9 | 9 | 9 | 9 |
| SHD | 13 | 13 | 13 | 13 | 13 |
| SHD-CPDAG | 10 | 9 | 9 | 9 | 9 |

Table 4: Nonlinear DAG learning performance (measured in structural hamming distance on DAG ad CPDAG, the lower the better) of different DAG constraints on Sachs et al. (2005)'s dataset . Different DAG constraints was plugged into the GRAN (Lachapelle et al., 2020) framework. **Top two rows**: results obtained from single-precision mode. **Bottom two rows**: results obtained from double-precision mode.

## 5.2 LINEAR SEM WITH UNKNOWN GROUND TRUTH SCALE

For linear SEM with an unknown ground truth scale, we applied the same data generation process as for the linear SEM with a known ground truth scale and Gaussian noise, but normalized the generated data $\mathbf{X}$ to have zero mean and unit variance. In this normalization procedure, the scale information of the variables is removed from the data. Particularly for Gaussian noise, in this case, the true DAG is not identifiable, and we may only identify it to a Markov Equivalent Class. Previously, it has been observed that direct optimization over equation 17 may result in poor performance, mainly due to the non-convex nature. In our experiments, we added an additional constraint to only allow edges to exist between highly correlated nodes. We first computed the Pearson correlation coefficients between every pair of nodes, and if the absolute value of the coefficient between two nodes is larger than 0.1, then we allow the edge to exist. During optimization, at every gradient descent step, we removed the disallowed edges from the candidate graph. We generated 10 instances of 1000-node ER1 graphs with Gaussian noise, and 1000 observational samples were generated for each instance.

The results are shown in Table 3. Our algorithm outperforms PC (Spirtes and Glymour, 1991) and GES (Chickering, 2002) in terms of SHD. Although higher-order DAG constraints may suffer more from non-convexity, by adding proper constraints on the candidate graphs, we can still achieve satisfactory results. In the results, we can see that the performance of higher-order constraints is better than that of lower-order ones, and also better than the exponential-based DAG constraints, which suggests that gradient vanishing may still be one important reason for poor performance.

## 5.3 EXPERIMENTAL RESULTS ON NONLINEAR CASES

Our DAG constraints can also be extended to continuous nonlinear DAG learning approaches by replacing their original DAG constraints. We incorporated our DAG constraints into Lachapelle et al. (2020) to model nonlinear Structural Equation Models (SEMs) and conducted experiments using Sachs et al. (2005)'s dataset pre-processed by Lachapelle et al. (2020)[2]. The GraN-DAG algorithm can operate in both single-precision and double-precision modes. The experimental results are shown in Table 4. The results suggest that DAG constraints with a finite spectral radius suffer less from gradient vanishing and, consequently, from numerical truncation errors. In contrast, the original GraN-DAG algorithm experiences gradient vanishing, particularly when running in single-precision mode, as higher-order constraints that prevent long loops are truncated due to limited machine precision.

## 6 CONCLUSION

The continuous differentiable DAG constraints play an important role in the continuous DAG learning algorithms. We show that many of these DAG constraints can be formulated using analytic functions. Several functional operators, including differentiation, summation, and multiplication, can be leveraged to create novel DAG constraints based on existing ones. Using these properties, we designed a series of DAG constraints and designed an efficient algorithm to evaluate these DAG constraints. Experiments on various settings show that our DAG constraints outperform previous state-of-the-arts approaches.

---

[2]Available at `https://github.com/kurowasan/GraN-DAG`.

## ACKNOWLEDGMENTS

This work was partially funded by the Centre for Augmented Reasoning (CAR), the Responsible Artificial Intelligence Research Centre (RAIR), the Australian Research Council (ARC) Discovery Early Career Researcher Award (DECRA) project DE210101624 to M. Gong and DE230101591 to D. Gong, and the Discovery Projects DP240102088 and DP240103278. We would also like to acknowledge the support from NSF Award No. 2229881, AI Institute for Societal Decision Making (AI-SDM), the National Institutes of Health (NIH) under Contract R01HL159805, and grants from Quris AI, Florin Court Capital, and MBZUAI-WIS Joint Program.

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

# Appendices

## A Proof of convergence of matrix analytic function

To prove that $f(\mathbf{A})$ converges for any matrix $\mathbf{A}$ with spectral radius $\rho(\mathbf{A}) < r$, we begin by considering the power series representation of the analytic function $f(x)$. Since $f(x)$ converges for $|x| < r$, it can be expressed as:

$$f(x) = \sum_{k=0}^{\infty} c_k x^k,$$

where the coefficients $c_k$ are such that the series converges absolutely for $|x| < r$. Now, let $\mathbf{A}$ be an $n \times n$ matrix with spectral radius $\rho(\mathbf{A}) < r$. By the definition of the spectral radius, all eigenvalues $\lambda$ of $\mathbf{A}$ satisfy $|\lambda| < r$.

We define the matrix function $f(\mathbf{A})$ by substituting $\mathbf{A}$ into the power series:

$$f(\mathbf{A}) = \sum_{k=0}^{\infty} c_k \mathbf{A}^k.$$

To show that this series converges, we use the fact that the spectral radius $\rho(\mathbf{A})$ is the infimum of all sub-multiplicative matrix norms of $\mathbf{A}$. Specifically, for any $\epsilon > 0$, there exists a sub-multiplicative matrix norm $\|\cdot\|$ such that $\|\mathbf{A}\| \leq \rho(\mathbf{A}) + \epsilon$. Since $\rho(\mathbf{A}) < r$, we can choose $\epsilon$ small enough so that $\|\mathbf{A}\| < r$.

To justify this claim, we provide a proof that the spectral radius $\rho(\mathbf{A})$ is indeed the infimum of all sub-multiplicative norms of $\mathbf{A}$. Let $\|\cdot\|$ be any sub-multiplicative matrix norm. For any eigenvalue $\lambda$ of $\mathbf{A}$ with corresponding eigenvector $v$, we have:

$$|\lambda|\|v\| = \|\lambda v\| = \|\mathbf{A}v\| \leq \|\mathbf{A}\|\|v\|.$$

Since $\|v\| \neq 0$, it follows that $|\lambda| \leq \|\mathbf{A}\|$. Taking the supremum over all eigenvalues $\lambda$, we obtain $\rho(\mathbf{A}) \leq \|\mathbf{A}\|$. This shows that $\rho(\mathbf{A})$ is a lower bound for all sub-multiplicative norms of $\mathbf{A}$.

To show that $\rho(\mathbf{A})$ is the infimum, we use the fact that for any $\epsilon > 0$, there exists a sub-multiplicative norm $\|\cdot\|_\epsilon$ such that $\|\mathbf{A}\|_\epsilon \leq \rho(\mathbf{A}) + \epsilon$. This can be achieved, for example, by using the Jordan canonical form of $\mathbf{A}$ and constructing an appropriate norm. Thus, $\rho(\mathbf{A})$ is the infimum of all sub-multiplicative norms of $\mathbf{A}$.

Now, consider the norm of the series:

$$\left\|\sum_{k=0}^{\infty} c_k \mathbf{A}^k\right\| \leq \sum_{k=0}^{\infty} |c_k|\|\mathbf{A}\|^k.$$

Since $\|\mathbf{A}\| < r$ and the original series $\sum_{k=0}^{\infty} c_k x^k$ converges absolutely for $|x| < r$, the series $\sum_{k=0}^{\infty} |c_k|\|\mathbf{A}\|^k$ also converges. Therefore, the matrix series $\sum_{k=0}^{\infty} c_k \mathbf{A}^k$ converges absolutely, and $f(\mathbf{A})$ is well-defined.

This completes the proof that $f(\mathbf{A})$ converges for any matrix $\mathbf{A}$ with spectral radius $\rho(\mathbf{A}) < r$.

## B Proof of Propositions

Our proof are also based on the well-known properties of analytic functions listed as follows:

1. Let $f_1(x)$, $f_2(x)$ be analytic functions on $(-r_1, r_1)$ and $(-r_2, r_2)$, then $f_1(x) + f_2(x)$ and $f_1(x)f_2(x)$ are analytic functions on $(-\min(r_1, r_2), \min(r_1, r_2))$;
2. Let $f(x)$ be analytic function on $(-r, r)$, then $\partial f(x)/\partial x$ is an analytic function on $(-r, r)$.

### B.1 Lemmas required for proofs

**Lemma 9.** *Let $\tilde{\mathbf{B}} \in \mathbb{R}_{\geq 0}^{d \times d}$ be the weighted adjacency matrix of a graph $\mathcal{G}$ with $d$ vertices, $\mathcal{G}$ is a DAG if and only if $\tilde{\mathbf{B}}^d = \mathbf{0}$.*

*Proof.* See Proposition 3.1 of Zhang et al. (2022). □

**Lemma 10.** *Let $\tilde{\mathbf{B}} \in \mathbb{R}_{\geqslant 0}^{d \times d}$ be the weighted adjacency matrix of a graph $\mathcal{G}$ with $d$ vertices, $\mathcal{G}$ is a DAG if and only*

$$\mathrm{tr}(\sum_{i=1}^{d} c_i \tilde{\mathbf{B}}^i) = 0,$$

*where $c_i > 0 \forall i$.*

*Proof.* See Wei et al. (2020). □

### B.2 PROOF OF PROPOSITION 1

**Proposition 1.** *Let $\tilde{\mathbf{B}} \in \mathbb{R}_{\geq 0}^{d \times d}$ with $\rho(\tilde{\mathbf{B}}) \leqslant r$ be the weighted adjacency matrix of a directed graph $\mathcal{G}$, and let $f$ be an analytic function in the form of equation 3, where we further assume $\forall i > 0$ we have $c_i > 0$, then $\mathcal{G}$ is acyclic if and only if*

$$\mathrm{tr}\left[f(\tilde{\mathbf{B}})\right] = c_0 d.$$

*Proof.* Without loss of generality, assume that $f$ can be formulated as:

$$f(x) = c_0 + \sum_{i=1}^{\infty} c_i x^i; \forall i, c_i > 0; \lim_{i \to \infty} c_i/c_{i+1} > 0. \tag{22}$$

First if $\mathcal{G}$ is acyclic, by Lemma 9 we must have

$$\tilde{\mathbf{B}}^k = \mathbf{0} \forall k \geqslant d, \tag{23}$$

which also indicates that $\rho(\tilde{\mathbf{B}}) = 0$. Thus we have

$$
\begin{aligned}
\mathrm{tr}\left[f(\tilde{\mathbf{B}})\right] &= \mathrm{tr}\left[c_0\mathbf{I} + \sum_{i=1}^{d} c_i \tilde{\mathbf{B}}^i + \underbrace{\sum_{i=d+1}^{\infty} c_i \tilde{\mathbf{B}}^i}_{\text{Equals } \mathbf{0}, \text{ By Lemma 9}}\right] \\
&= \mathrm{tr}[c_0\mathbf{I}] + \underbrace{\mathrm{tr}\left[\sum_{i=1}^{d} c_i \tilde{\mathbf{B}}^i\right]}_{\text{Equals 0, By Lemma 10}} \\
&= c_0 d.
\end{aligned}
\tag{24}
$$

On the other hand, if $\mathrm{tr}\left[f(\tilde{\mathbf{B}})\right] = c_0 d$, we must have that

$$\mathrm{tr}\left[\sum_{i=1}^{\infty} c_i \tilde{\mathbf{B}}^i\right] = 0.$$

By the fact all entries of $\tilde{\mathbf{B}}$ are positive, we have that

$$0 \leqslant \mathrm{tr}\left[\sum_{i=1}^{d} c_i \tilde{\mathbf{B}}^i\right] \leqslant \left[\sum_{i=1}^{\infty} c_i \tilde{\mathbf{B}}^i\right] = 0. \tag{25}$$

Then we must have

$$\mathrm{tr}\left[\sum_{i=1}^{d} c_i \tilde{\mathbf{B}}^i\right] = 0.$$

Finally by Lemma 10 we have that $\mathcal{G}$ is a DAG. □

### B.3 PROOF OF PROPOSITION 2

In all the paper, we consider analytic functions $f$ from the functional class $\mathcal{F}$ defined in equation 5.

**Proposition 2.** *There exists some real number $r$, where for all $\{\tilde{\mathbf{B}} \in \mathbb{R}_{\geqslant 0}^{d \times d} | \rho(\tilde{\mathbf{B}}) < r\}$, the derivative of $\operatorname{tr}\left[f(\tilde{\mathbf{B}})\right]$ w.r.t. $\tilde{\mathbf{B}}$ is*

$$\nabla_{\tilde{\mathbf{B}}} \operatorname{tr}\left[f(\tilde{\mathbf{B}})\right] = \left[\nabla_x f(x)|_{x=\tilde{\mathbf{B}}}\right]^\top.$$

*Proof.* Without loss of generality, assume that $f$ can be formulated as:

$$f(x) = c_0 + \sum_{i=1}^{\infty} c_i x^i; \forall i, c_i > 0; \lim_{i \to \infty} c_i / c_{i+1} > 0. \tag{26}$$

For some $i$ by basic matrix differentiation we have

$$\frac{\partial \operatorname{tr} \tilde{\mathbf{B}}^i}{\partial \tilde{\mathbf{B}}} = (i \, \mathbf{B}^{i-1})^\top, \tag{27}$$

and then by the properties of power series we have

$$
\begin{aligned}
\nabla_{\tilde{\mathbf{B}}} \operatorname{tr}\left[f(\tilde{\mathbf{B}})\right] &= \nabla_{\tilde{\mathbf{B}}} \operatorname{tr}\left[c_0 \mathbf{I} + \sum_{i=1}^{\infty} c_i \tilde{\mathbf{B}}^i\right] \\
&= \sum_{i=1}^{\infty} \nabla_{\tilde{\mathbf{B}}} \operatorname{tr} c_i \tilde{\mathbf{B}}^i \\
&= \left[\sum_{i=1}^{\infty} c_i i \tilde{\mathbf{B}}^{i-1}\right]^\top \\
&= \left[\sum_{i=1}^{\infty} c_i i x^{i-1}\bigg|_{x=\tilde{\mathbf{B}}}\right]^\top = \left[\nabla_x f(x)|_{x=\tilde{\mathbf{B}}}\right]^\top,
\end{aligned}
\tag{28}
$$

where we can exchange $\nabla$ and $\sum_{i=1}^{\infty}$ because after the exchanging the new power series will still converge (by properties of analytic functions). $\qquad\square$

### B.4 PROOF OF PROPOSITION 3

**Proposition 3.** *Let $f(x) = c_0 + \sum_{i=1}^{\infty} c_i x^i \in \mathcal{F}$ be a analytic function on $(-r, r)$, and let $n$ be arbitary integer larger than 1, then $\tilde{\mathbf{B}} \in \mathbb{R}_{\geqslant 0}^{d \times d}|$ with spectral radius $\rho(\hat{\mathbf{B}}) \leqslant r$ forms a DAG if and only if*

$$\operatorname{tr}\left[\frac{\partial^n f(x)}{\partial x^n}\bigg|_{x=\tilde{\mathbf{B}}}\right] = n! c_n.$$

*Proof.* By properties of analytic functions, the $n^{\text{th}}$ order derivative of an analytic function $f(x)$ on $(-r, r)$ is still an analytic function on $(-r, r)$. Particularly for $f(x) = c_0 + \sum_{i=1}^{\infty} c_i x^i \in \mathcal{F}$, we have

$$
\begin{aligned}
\frac{\partial^n f(x)}{\partial x^n} &= \sum_{i=1}^{\infty} \frac{\partial^n c_i x^i}{\partial x^n} \\
&= \sum_{i=n}^{\infty} \frac{\partial^n c_i x^i}{\partial x^n} \\
&= \sum_{i=n}^{\infty} \left[c_i x^{i-n} \prod_{k=i-n+1}^{n} k\right] \\
&= n! c_n + \sum_{i=1}^{\infty} \left[c_{i+n} x^i \prod_{k=i}^{n+i} k\right],
\end{aligned}
\tag{29}
$$

where by the fact $c_i > 0, \forall i > 1$, we have that $\frac{\partial^n f(x)}{\partial x^n} \in \mathcal{F}$. Then by Proposition 1 we immediately proved the proposition. □

## B.5 PROOF OF PROPOSITION 4

**Proposition 4.** *Let $f_1(x) = c_0^1 + \sum_{i=1}^{\infty} c_i^1 x^i \in \mathcal{F}$, and $f_2(x) = c_0^2 + \sum_{i=1}^{\infty} c_i^2 x^i \in \mathcal{F}$. Then for an adjancency matrix $\tilde{\mathbf{B}} \in \mathbb{R}_{\geq 0}^{d \times d}$ with spectral radius $\rho(\tilde{\mathbf{B}}) \leq \min(\lim_{i \to \infty} c_i^1/c_{i+1}^1, \lim_{i \to \infty} c_i^2/c_{i+1}^2)\}$, the following three statements are equivalent:*

1. *$\tilde{\mathbf{B}}$ forms a DAG;*

2. *$\mathrm{tr}[f_1(\tilde{\mathbf{B}}) + f_2(\tilde{\mathbf{B}})] = (c_0^1 + c_0^2)d$;*

3. *$\mathrm{tr}[f_1(\tilde{\mathbf{B}})f_2(\tilde{\mathbf{B}})] = c_0^1 c_0^2 d$.*

*Proof.* By properties of analytic functions, we have

$$f_1(x) + f_2(x) = c_0^1 + c_0^2 + \sum_{i=1}^{\infty}(c_i^1 + c_i^2)x^i \tag{30}$$

is an analytic function, and its convergence radius is given by

$$\lim_{i \to \infty}(c_i^1 + c_i^2)/(c_{i+1}^1 + c_{i+1}^2) = \min(\lim_{i \to \infty} c_i^1/c_{i+1}^1, \lim_{i \to \infty} c_i^2/c_{i+1}^2), \tag{31}$$

and thus by Proposition 1 the statement 1 and 2 are equivalent. Similarly by properties of analytic functions statement 1 and 3 are equivalent. Thus the 3 statements are equivalent. □

## B.6 PROOF OF PROPOSITION 5

**Proposition 5.** *Let $n$ be any positive integer, the adjacency matrix $\tilde{\mathbf{B}} \in \{\hat{\mathbf{B}} \in \mathbb{R}_{\geq 0}^{d \times d} | \rho(\hat{\mathbf{B}}) \leq s\}$ if and only if*

$$\mathrm{tr}(\mathbf{I} - \tilde{\mathbf{B}}/s)^{-n} = d,$$

*and the gradients of the DAG constraints satisfies that $\forall \tilde{\mathbf{B}} \in \{\hat{\mathbf{B}} \in \mathbb{R}_{\geq 0}^{d \times d} | \rho(\hat{\mathbf{B}}) \leq s\}$*

$$\|\nabla_{\tilde{\mathbf{B}}} \mathrm{tr}(\mathbf{I} - \tilde{\mathbf{B}}/s)^{-n}\| \leq \|\nabla_{\tilde{\mathbf{B}}} \mathrm{tr}(\mathbf{I} - \tilde{\mathbf{B}}/s)^{-n-k}\|,$$

*where $k$ is an arbitrary positive integer, and $\|\cdot\|$ denote an arbitrary matrix norm induced by vector p-norm.*

*Proof.* By Proposition 4 or Proposition 3, it would be straightforward that $\mathrm{tr}(\mathbf{I} - \tilde{\mathbf{B}})^{-n} = d$ is a necessary and sufficient condition for an adjacency matrix $\tilde{\mathbf{B}} \in \{\hat{\mathbf{B}} \in \mathbb{R}_{\geq 0}^{d \times d} | \rho(\hat{\mathbf{B}}) \leq s\}$ to form a DAG.

For the norm of gradients, it is straightforward that

$$\frac{\partial(1 - x)^{-n}}{\partial x} = n(1 - x)^{-n-1}. \tag{32}$$

For arbitrary $n$ we have

$$(1 - x)^{-n} = 1 + \sum_{i=1}^{\infty} \left[ \prod_{j=n}^{n+i-1} j \right] x^i, \tag{33}$$

and obviously the coefficients is monotonic increasing w.r.t. $n$. Thus by the fact $\forall \tilde{\mathbf{B}} \in \{\hat{\mathbf{B}} \in \mathbb{R}_{\geq 0}^{d \times d} | \rho(\hat{\mathbf{B}}) \leq s\}$ we have for any $j > 0, k > 0$

$$\|(\mathbf{I} - \tilde{\mathbf{B}})^{-j}\| \leq \|(\mathbf{I} - \tilde{\mathbf{B}})^{-j-k}\|. \tag{34}$$

As a result, we have

$$\begin{aligned}
\|\nabla_{\tilde{\mathbf{B}}} \mathrm{tr}(\mathbf{I} - \tilde{\mathbf{B}})^{-n}\| &= n\|(\mathbf{I} - \tilde{\mathbf{B}})^{-n-1}\| \leq (n + k)\|(\mathbf{I} - \tilde{\mathbf{B}})^{-n-1}\| \\
&\leq (n + k)\|(\mathbf{I} - \tilde{\mathbf{B}})^{-n-k-1}\| = \|\nabla_{\tilde{\mathbf{B}}} \mathrm{tr}(\mathbf{I} - \tilde{\mathbf{B}})^{-n-k}\|.
\end{aligned} \tag{35}$$

□

## B.7 PROOF OF PROPOSITION 6

**Proposition 6.** *Thus the Hessian of the DAG constraints equation 6 can be obtained as follows:*

$$\nabla_{\tilde{\mathbf{B}}}^2 \mathrm{tr} f(\tilde{\mathbf{B}}) = \mathbf{K}_{dd} \sum_{i=2}^{\infty} i c_i \sum_{j=0}^{i-2} \left[\tilde{\mathbf{B}}^j\right]^\top \otimes \left[\tilde{\mathbf{B}}^{i-2-j}\right],$$

*where $\otimes$ denotes the Kronecker product.*

*Proof.* Firstly, the derivative of matrix power can be obtained using the following equation (Magnus and Neudecker, 2019),

$$\nabla_{\tilde{\mathbf{B}}}^2 \mathrm{tr} \tilde{\mathbf{B}}^k = k \mathbf{K}_{dd} \sum_{j=0}^{i-2} \left[\tilde{\mathbf{B}}^j\right]^\top \otimes \left[\tilde{\mathbf{B}}^{i-2-j}\right], \tag{36}$$

where $\otimes$ denotes the Kronecker product. Thus the Hessian of analytic DAG constraints can be obtained as follows:

$$\nabla_{\tilde{\mathbf{B}}}^2 \mathrm{tr} f(\tilde{\mathbf{B}}) = \sum_{i=0}^{\infty} c_i \mathrm{tr} \nabla_{\tilde{\mathbf{B}}}^2 \tilde{\mathbf{B}}^i$$

$$= \mathbf{K}_{dd} \sum_{i=2}^{\infty} i c_i \sum_{j=0}^{i-2} \left[\tilde{\mathbf{B}}^j\right]^\top \otimes \left[\tilde{\mathbf{B}}^{i-2-j}\right]. \tag{37}$$

$\square$

## B.8 PROOF OF PROPOSITION 7

**Proposition 7.** *For two analytic function $f_1(x) = c_{0,1} + \sum_{i=1}^{\infty} c_{i,1} x^i \in \mathcal{F}$ and $f_2(x) = c_{0,2} + \sum_{i=1}^{\infty} c_{i,2} x^i \in \mathcal{F}$, if $\forall i \geqslant 1$ we have $c_{i,1} \geqslant c_{i,2}$, then*

$$\rho(\nabla_{\tilde{\mathbf{B}}}^2 \mathrm{tr} f_1(\tilde{\mathbf{B}})) \geqslant \rho(\nabla_{\tilde{\mathbf{B}}}^2 \mathrm{tr} f_2(\tilde{\mathbf{B}})),$$

*where $\rho(\cdot)$ denotes the spectral radius of a matrix.*

*Proof.* Obviously, each entries in the Hessian of $\mathrm{tr} f_1(\tilde{\mathbf{B}}))$ is larger than the corresponding ones in $\mathrm{tr} f_2(\tilde{\mathbf{B}}))$. Thus for any unit length vector $\mathbf{u}$ with all positive entries we would have

$$\mathbf{u}^\top \nabla_{\tilde{\mathbf{B}}}^2 \mathrm{tr} f_1(\tilde{\mathbf{B}})) \mathbf{u} \geqslant \mathbf{u}^\top \nabla_{\tilde{\mathbf{B}}}^2 \mathrm{tr} f_2(\tilde{\mathbf{B}})) \mathbf{u},$$

and then it would be straightforward that

$$\rho(\nabla_{\tilde{\mathbf{B}}}^2 \mathrm{tr} f_1(\tilde{\mathbf{B}})) = \max_{\mathbf{u}:\mathbf{u} \geqslant 0, \|\mathbf{u}\|_2 = 1} \mathbf{u}^\top \nabla_{\tilde{\mathbf{B}}}^2 \mathrm{tr} f_1(\tilde{\mathbf{B}})) \mathbf{u}$$

$$\geqslant \max_{\mathbf{u}:\mathbf{u} \geqslant 0, \|\mathbf{u}\|_2 = 1} \mathbf{u}^\top \nabla_{\tilde{\mathbf{B}}}^2 \mathrm{tr} f_2(\tilde{\mathbf{B}})) \mathbf{u} = \rho(\nabla_{\tilde{\mathbf{B}}}^2 \mathrm{tr} f_2(\tilde{\mathbf{B}})).$$

$\square$

## C ERROR ANALYSIS OF THE ANALYTIC FUNCTION BASED DAG CONSTRAINTS

Without loss of generalization, we analyze the truncated error for a DAG constraints

$$f(\tilde{\mathbf{B}}) = c_0 \mathbf{I} + \sum_{i=1}^{\infty} c_i \tilde{\mathbf{B}}^i$$

which converges for $\rho(\tilde{\mathbf{B}}) < 1$, and the results can be easily generalized to normal analytic function based DAG constraints.

it would be obvious for a truncated approximation

$$f_k(\tilde{\mathbf{B}}) = c_0 \mathbb{I} + \sum_{i=1}^{k} c_i \tilde{\mathbf{B}}^i$$

the error between the truncated approximation and the original analytic function can be obtained by

$$\|f(\tilde{\mathbf{B}}) - f_k(\tilde{\mathbf{B}})\|_2 = \| \sum_{i=k+1}^{\infty} c_i \tilde{\mathbf{B}}^i \|_2$$

$$= \underbrace{\|\tilde{\mathbf{B}}^{k+1} \sum_{i=0}^{\infty} \hat{c}_i \tilde{\mathbf{B}}^i\|_2}_{\text{Let } \hat{c}_i = c_{k+1+i}}$$

$$\leqslant \|\tilde{\mathbf{B}}^{k+1}\|_2 \sup_{\tilde{\mathbf{B}}:\rho(\mathbf{B})<1} \| \sum_{i=0}^{\infty} \hat{c}_i \tilde{\mathbf{B}}^i \|_2$$

$$= \|\tilde{\mathbf{B}}^{k+1}\|_2 C(f, k)$$

where we define $C(f, k) = \sup_{\tilde{\mathbf{B}}:\rho(\mathbf{B})<1} \| \sum_{i=0}^{\infty} \hat{c}_i \tilde{\mathbf{B}}^i \|_2$. It is notable that $\|\tilde{\mathbf{B}}^{k+1}\|_2$ converges to $0$ exponentially w.r.t. the increase of $k$ due to the fact $\rho(\tilde{\mathbf{B}}) < 1$. Meanwhile, it is obvious that $C(f, k)$ increases at most in polynomial rate due to that fact $\sum_{i=0}^{\infty} \hat{c}_i \tilde{\mathbf{B}}^i \|_2$ converges for $\rho(\tilde{\mathbf{B}}) < 1$. Finally, the residual of Algorithm 2 converges to $0$ in a rate of a polynomial of $k$ divided by the exponential of $k$.

## D    HYPER PARAMETERS

In terms of hyper-parameters, our selection involves $\alpha = 0.1$, $\lambda_1 = 0.1$, and $T = 5$. For $s$ we use the same annealing approach as Bello et al. (2022), but with our strategy to reset $s$ when candidate graph goes out of the desired region.

In all experiments in this paper, for continuous based approaches we use exactly the same hyper parameter as Bello et al. (2022), for conditional independent test and score based approaches we use the default parameter in Causal Discovery Toolbox[3].

## E    EXTRA EXPERIMENTAL RESULTS

In this section, we provide additional experimental results, including true positive rate, false detection rate and running time for large scale graphs, as well as experimental results on small scale graphs.

---

[3]https://fentechsolutions.github.io/CausalDiscoveryToolbox/html/index.html

Table 5: DAG learning performance (measured in true positive rate, the higher the better, best results in **bold**) of different algorithms on large scale (500-2000 nodes) graphs with different noise distributions. Our algorithm performs better than previous approaches.

| Graphs | Nodes | DAGMA | Order-1 | Order-2 | Order-3 | Order-4 |
|---|---|---|---|---|---|---|
| ER2-gauss | 500 | $0.97 \pm 0.01$ | $0.98 \pm 0.01$ | $0.98 \pm 0.01$ | $0.98 \pm 0.01$ | $\mathbf{0.98 \pm 0.01}$ |
| ER2-gauss | 1000 | $0.97 \pm 0.01$ | $0.98 \pm 0.01$ | $0.98 \pm 0.00$ | $0.98 \pm 0.00$ | $\mathbf{0.98 \pm 0.00}$ |
| ER2-gauss | 2000 | $0.96 \pm 0.01$ | $0.97 \pm 0.01$ | $0.98 \pm 0.00$ | $0.98 \pm 0.00$ | $\mathbf{0.98 \pm 0.00}$ |
| ER3-gauss | 500 | $0.95 \pm 0.01$ | $0.96 \pm 0.01$ | $0.97 \pm 0.01$ | $0.97 \pm 0.01$ | $\mathbf{0.97 \pm 0.01}$ |
| ER3-gauss | 1000 | $0.94 \pm 0.01$ | $0.96 \pm 0.01$ | $0.96 \pm 0.01$ | $0.97 \pm 0.01$ | $\mathbf{0.97 \pm 0.01}$ |
| ER3-gauss | 2000 | $0.94 \pm 0.01$ | $0.95 \pm 0.01$ | $0.95 \pm 0.01$ | $0.96 \pm 0.01$ | $\mathbf{0.96 \pm 0.00}$ |
| ER4-gauss | 500 | $0.92 \pm 0.02$ | $0.94 \pm 0.02$ | $0.94 \pm 0.02$ | $0.94 \pm 0.02$ | $\mathbf{0.95 \pm 0.02}$ |
| ER4-gauss | 1000 | $0.89 \pm 0.02$ | $0.92 \pm 0.01$ | $0.92 \pm 0.01$ | $0.93 \pm 0.01$ | $\mathbf{0.93 \pm 0.01}$ |
| ER4-gauss | 2000 | $0.89 \pm 0.01$ | $0.90 \pm 0.01$ | $0.90 \pm 0.01$ | $0.91 \pm 0.01$ | $\mathbf{0.91 \pm 0.01}$ |
| ER2-exp | 500 | $0.96 \pm 0.01$ | $0.97 \pm 0.01$ | $0.98 \pm 0.01$ | $\mathbf{0.98 \pm 0.01}$ | $0.98 \pm 0.01$ |
| ER2-exp | 1000 | $0.97 \pm 0.01$ | $0.98 \pm 0.00$ | $0.98 \pm 0.00$ | $\mathbf{0.98 \pm 0.00}$ | $0.98 \pm 0.00$ |
| ER2-exp | 2000 | $0.00 \pm 0.00$ | $\mathbf{0.58 \pm 0.48}$ | $0.10 \pm 0.29$ | $0.10 \pm 0.29$ | $0.10 \pm 0.29$ |
| ER3-exp | 500 | $0.95 \pm 0.01$ | $0.96 \pm 0.01$ | $0.97 \pm 0.01$ | $0.97 \pm 0.01$ | $\mathbf{0.97 \pm 0.01}$ |
| ER3-exp | 1000 | $0.94 \pm 0.01$ | $0.96 \pm 0.01$ | $0.96 \pm 0.01$ | $0.97 \pm 0.01$ | $\mathbf{0.97 \pm 0.01}$ |
| ER3-exp | 2000 | $0.00 \pm 0.00$ | $0.09 \pm 0.28$ | $0.09 \pm 0.28$ | $\mathbf{0.10 \pm 0.29}$ | $0.00 \pm 0.00$ |
| ER4-exp | 500 | $0.91 \pm 0.02$ | $0.93 \pm 0.02$ | $0.93 \pm 0.02$ | $0.94 \pm 0.02$ | $\mathbf{0.95 \pm 0.01}$ |
| ER4-exp | 1000 | $0.89 \pm 0.01$ | $0.92 \pm 0.01$ | $0.92 \pm 0.01$ | $0.93 \pm 0.01$ | $\mathbf{0.94 \pm 0.01}$ |
| ER4-exp | 2000 | $0.00 \pm 0.00$ | $0.09 \pm 0.27$ | $0.09 \pm 0.27$ | $\mathbf{0.09 \pm 0.27}$ | $0.00 \pm 0.00$ |
| ER2-gumbel | 500 | $0.98 \pm 0.01$ | $0.99 \pm 0.01$ | $0.99 \pm 0.01$ | $0.99 \pm 0.01$ | $\mathbf{0.99 \pm 0.00}$ |
| ER2-gumbel | 1000 | $0.98 \pm 0.01$ | $0.99 \pm 0.01$ | $0.99 \pm 0.00$ | $\mathbf{0.99 \pm 0.00}$ | $0.99 \pm 0.00$ |
| ER2-gumbel | 2000 | $0.00 \pm 0.00$ | $\mathbf{0.59 \pm 0.48}$ | $0.10 \pm 0.29$ | $0.10 \pm 0.30$ | $0.10 \pm 0.30$ |
| ER3-gumbel | 500 | $0.95 \pm 0.01$ | $0.97 \pm 0.01$ | $0.98 \pm 0.01$ | $\mathbf{0.98 \pm 0.01}$ | $0.98 \pm 0.01$ |
| ER3-gumbel | 1000 | $0.95 \pm 0.01$ | $0.97 \pm 0.01$ | $0.97 \pm 0.01$ | $0.97 \pm 0.01$ | $\mathbf{0.98 \pm 0.01}$ |
| ER3-gumbel | 2000 | $0.00 \pm 0.00$ | $0.00 \pm 0.00$ | $0.00 \pm 0.00$ | $0.00 \pm 0.00$ | $\mathbf{0.10 \pm 0.29}$ |
| ER4-gumbel | 500 | $0.93 \pm 0.02$ | $0.95 \pm 0.01$ | $0.96 \pm 0.01$ | $0.96 \pm 0.01$ | $\mathbf{0.96 \pm 0.02}$ |
| ER4-gumbel | 1000 | $0.90 \pm 0.01$ | $0.94 \pm 0.01$ | $0.94 \pm 0.01$ | $0.95 \pm 0.01$ | $\mathbf{0.95 \pm 0.01}$ |
| ER4-gumbel | 2000 | $0.00 \pm 0.00$ | $0.00 \pm 0.00$ | $0.00 \pm 0.00$ | $0.00 \pm 0.00$ | $\mathbf{0.09 \pm 0.28}$ |
| SF2-gauss | 500 | $0.97 \pm 0.01$ | $0.98 \pm 0.01$ | $0.98 \pm 0.01$ | $0.98 \pm 0.01$ | $\mathbf{0.98 \pm 0.01}$ |
| SF2-gauss | 1000 | $0.97 \pm 0.01$ | $0.98 \pm 0.01$ | $0.98 \pm 0.00$ | $0.98 \pm 0.00$ | $\mathbf{0.98 \pm 0.00}$ |
| SF2-gauss | 2000 | $0.96 \pm 0.01$ | $0.97 \pm 0.01$ | $0.98 \pm 0.00$ | $0.98 \pm 0.00$ | $\mathbf{0.98 \pm 0.00}$ |
| SF3-gauss | 500 | $0.95 \pm 0.01$ | $0.96 \pm 0.01$ | $0.97 \pm 0.01$ | $0.97 \pm 0.01$ | $\mathbf{0.97 \pm 0.01}$ |
| SF3-gauss | 1000 | $0.94 \pm 0.01$ | $0.96 \pm 0.01$ | $0.96 \pm 0.01$ | $0.97 \pm 0.01$ | $\mathbf{0.97 \pm 0.01}$ |
| SF3-gauss | 2000 | $0.94 \pm 0.01$ | $0.95 \pm 0.01$ | $0.95 \pm 0.01$ | $0.96 \pm 0.01$ | $\mathbf{0.96 \pm 0.00}$ |
| SF4-gauss | 500 | $0.92 \pm 0.02$ | $0.94 \pm 0.02$ | $0.94 \pm 0.02$ | $0.94 \pm 0.02$ | $\mathbf{0.95 \pm 0.02}$ |
| SF4-gauss | 1000 | $0.89 \pm 0.02$ | $0.92 \pm 0.01$ | $0.92 \pm 0.01$ | $0.93 \pm 0.01$ | $\mathbf{0.93 \pm 0.01}$ |
| SF4-gauss | 2000 | $0.89 \pm 0.01$ | $0.90 \pm 0.01$ | $0.90 \pm 0.01$ | $0.91 \pm 0.01$ | $\mathbf{0.91 \pm 0.01}$ |
| SF2-exp | 500 | $0.96 \pm 0.01$ | $0.97 \pm 0.01$ | $0.98 \pm 0.01$ | $\mathbf{0.98 \pm 0.01}$ | $0.98 \pm 0.01$ |
| SF2-exp | 1000 | $0.97 \pm 0.01$ | $0.98 \pm 0.00$ | $0.98 \pm 0.00$ | $\mathbf{0.98 \pm 0.00}$ | $0.98 \pm 0.00$ |
| SF3-exp | 500 | $0.95 \pm 0.01$ | $0.96 \pm 0.01$ | $0.97 \pm 0.01$ | $0.97 \pm 0.01$ | $\mathbf{0.97 \pm 0.01}$ |
| SF3-exp | 1000 | $0.94 \pm 0.01$ | $0.96 \pm 0.01$ | $0.96 \pm 0.01$ | $0.97 \pm 0.01$ | $\mathbf{0.97 \pm 0.01}$ |
| SF4-exp | 500 | $0.91 \pm 0.02$ | $0.93 \pm 0.02$ | $0.93 \pm 0.02$ | $0.94 \pm 0.02$ | $\mathbf{0.95 \pm 0.01}$ |
| SF4-exp | 1000 | $0.89 \pm 0.01$ | $0.92 \pm 0.01$ | $0.92 \pm 0.01$ | $0.93 \pm 0.01$ | $\mathbf{0.94 \pm 0.01}$ |
| SF2-gumbel | 500 | $0.98 \pm 0.01$ | $0.99 \pm 0.01$ | $0.99 \pm 0.01$ | $0.99 \pm 0.01$ | $\mathbf{0.99 \pm 0.00}$ |
| SF2-gumbel | 1000 | $0.98 \pm 0.01$ | $0.99 \pm 0.01$ | $0.99 \pm 0.00$ | $\mathbf{0.99 \pm 0.00}$ | $0.99 \pm 0.00$ |
| SF3-gumbel | 500 | $0.95 \pm 0.01$ | $0.97 \pm 0.01$ | $0.98 \pm 0.01$ | $\mathbf{0.98 \pm 0.01}$ | $0.98 \pm 0.01$ |
| SF3-gumbel | 1000 | $0.95 \pm 0.01$ | $0.97 \pm 0.01$ | $0.97 \pm 0.01$ | $0.97 \pm 0.01$ | $\mathbf{0.98 \pm 0.01}$ |
| SF4-gumbel | 500 | $0.93 \pm 0.02$ | $0.95 \pm 0.01$ | $0.96 \pm 0.01$ | $0.96 \pm 0.01$ | $\mathbf{0.96 \pm 0.02}$ |
| SF4-gumbel | 1000 | $0.90 \pm 0.01$ | $0.94 \pm 0.01$ | $0.94 \pm 0.01$ | $0.95 \pm 0.01$ | $\mathbf{0.95 \pm 0.01}$ |

Table 6: DAG learning performance (measured in false detection rate, the lower the better, best results in **bold**) of different algorithms on large scale (500-2000 nodes) graphs with different noise distributions. Our algorithm performs better than previous approaches.

| Graphs | Nodes | DAGMA | Order-1 | Order-2 | Order-3 | Order-4 |
|---|---|---|---|---|---|---|
| ER2-gauss | 500 | $0.02 \pm 0.02$ | $0.02 \pm 0.01$ | $\mathbf{0.01 \pm 0.01}$ | $\mathbf{0.01 \pm 0.01}$ | $\mathbf{0.01 \pm 0.01}$ |
| ER2-gauss | 1000 | $0.02 \pm 0.01$ | $0.02 \pm 0.01$ | $\mathbf{0.01 \pm 0.01}$ | $\mathbf{0.01 \pm 0.01}$ | $\mathbf{0.01 \pm 0.01}$ |
| ER2-gauss | 2000 | $0.02 \pm 0.01$ | $0.02 \pm 0.01$ | $0.02 \pm 0.01$ | $0.02 \pm 0.01$ | $\mathbf{0.02 \pm 0.00}$ |
| ER3-gauss | 500 | $0.04 \pm 0.02$ | $0.04 \pm 0.02$ | $0.03 \pm 0.02$ | $\mathbf{0.03 \pm 0.01}$ | $0.04 \pm 0.02$ |
| ER3-gauss | 1000 | $0.06 \pm 0.01$ | $0.05 \pm 0.01$ | $0.04 \pm 0.01$ | $\mathbf{0.03 \pm 0.01}$ | $\mathbf{0.03 \pm 0.01}$ |
| ER3-gauss | 2000 | $0.06 \pm 0.01$ | $0.06 \pm 0.02$ | $0.05 \pm 0.01$ | $\mathbf{0.04 \pm 0.01}$ | $\mathbf{0.04 \pm 0.01}$ |
| ER4-gauss | 500 | $0.08 \pm 0.03$ | $0.08 \pm 0.03$ | $0.08 \pm 0.04$ | $0.07 \pm 0.04$ | $\mathbf{0.07 \pm 0.03}$ |
| ER4-gauss | 1000 | $0.12 \pm 0.03$ | $0.11 \pm 0.03$ | $0.10 \pm 0.02$ | $\mathbf{0.09 \pm 0.03}$ | $0.10 \pm 0.03$ |
| ER4-gauss | 2000 | $0.14 \pm 0.01$ | $0.13 \pm 0.01$ | $0.13 \pm 0.02$ | $0.12 \pm 0.02$ | $\mathbf{0.12 \pm 0.01}$ |
| ER2-exp | 500 | $0.03 \pm 0.02$ | $0.02 \pm 0.01$ | $\mathbf{0.01 \pm 0.01}$ | $0.01 \pm 0.02$ | $0.02 \pm 0.02$ |
| ER2-exp | 1000 | $0.02 \pm 0.01$ | $0.02 \pm 0.01$ | $\mathbf{0.01 \pm 0.01}$ | $\mathbf{0.01 \pm 0.01}$ | $\mathbf{0.01 \pm 0.01}$ |
| ER3-exp | 500 | $0.05 \pm 0.02$ | $0.04 \pm 0.02$ | $\mathbf{0.03 \pm 0.01}$ | $0.04 \pm 0.02$ | $0.04 \pm 0.02$ |
| ER3-exp | 1000 | $0.06 \pm 0.01$ | $0.05 \pm 0.01$ | $0.04 \pm 0.01$ | $\mathbf{0.03 \pm 0.01}$ | $\mathbf{0.03 \pm 0.01}$ |
| ER4-exp | 500 | $0.09 \pm 0.04$ | $0.09 \pm 0.04$ | $0.09 \pm 0.05$ | $0.07 \pm 0.04$ | $\mathbf{0.06 \pm 0.03}$ |
| ER4-exp | 1000 | $0.12 \pm 0.03$ | $0.11 \pm 0.02$ | $0.11 \pm 0.02$ | $\mathbf{0.09 \pm 0.03}$ | $0.10 \pm 0.02$ |
| ER2-gumbel | 500 | $0.03 \pm 0.02$ | $\mathbf{0.01 \pm 0.01}$ | $\mathbf{0.01 \pm 0.01}$ | $\mathbf{0.01 \pm 0.01}$ | $\mathbf{0.01 \pm 0.01}$ |
| ER2-gumbel | 1000 | $0.02 \pm 0.01$ | $0.02 \pm 0.01$ | $\mathbf{0.01 \pm 0.00}$ | $0.01 \pm 0.01$ | $0.02 \pm 0.01$ |
| ER3-gumbel | 500 | $0.06 \pm 0.02$ | $0.04 \pm 0.02$ | $0.03 \pm 0.03$ | $\mathbf{0.03 \pm 0.01}$ | $0.04 \pm 0.02$ |
| ER3-gumbel | 1000 | $0.06 \pm 0.01$ | $0.05 \pm 0.01$ | $0.04 \pm 0.02$ | $0.04 \pm 0.01$ | $\mathbf{0.03 \pm 0.01}$ |
| ER4-gumbel | 500 | $0.10 \pm 0.04$ | $0.09 \pm 0.05$ | $0.09 \pm 0.04$ | $\mathbf{0.08 \pm 0.04}$ | $0.08 \pm 0.05$ |
| ER4-gumbel | 1000 | $0.14 \pm 0.03$ | $0.12 \pm 0.03$ | $0.12 \pm 0.03$ | $0.11 \pm 0.03$ | $\mathbf{0.10 \pm 0.03}$ |
| SF2-gauss | 500 | $0.02 \pm 0.02$ | $0.02 \pm 0.01$ | $\mathbf{0.01 \pm 0.01}$ | $\mathbf{0.01 \pm 0.01}$ | $\mathbf{0.01 \pm 0.01}$ |
| SF2-gauss | 1000 | $0.02 \pm 0.01$ | $0.02 \pm 0.01$ | $\mathbf{0.01 \pm 0.01}$ | $\mathbf{0.01 \pm 0.01}$ | $\mathbf{0.01 \pm 0.01}$ |
| SF2-gauss | 2000 | $0.02 \pm 0.01$ | $0.02 \pm 0.01$ | $0.02 \pm 0.01$ | $0.02 \pm 0.01$ | $\mathbf{0.02 \pm 0.00}$ |
| SF3-gauss | 500 | $0.04 \pm 0.02$ | $0.04 \pm 0.02$ | $0.03 \pm 0.02$ | $\mathbf{0.03 \pm 0.01}$ | $0.04 \pm 0.02$ |
| SF3-gauss | 1000 | $0.06 \pm 0.01$ | $0.05 \pm 0.01$ | $0.04 \pm 0.01$ | $\mathbf{0.03 \pm 0.01}$ | $\mathbf{0.03 \pm 0.01}$ |
| SF3-gauss | 2000 | $0.06 \pm 0.01$ | $0.06 \pm 0.02$ | $0.05 \pm 0.01$ | $\mathbf{0.04 \pm 0.01}$ | $\mathbf{0.04 \pm 0.01}$ |
| SF4-gauss | 500 | $0.08 \pm 0.03$ | $0.08 \pm 0.03$ | $0.08 \pm 0.04$ | $0.07 \pm 0.04$ | $\mathbf{0.07 \pm 0.03}$ |
| SF4-gauss | 1000 | $0.12 \pm 0.03$ | $0.11 \pm 0.03$ | $0.10 \pm 0.02$ | $\mathbf{0.09 \pm 0.03}$ | $0.10 \pm 0.03$ |
| SF4-gauss | 2000 | $0.14 \pm 0.01$ | $0.13 \pm 0.01$ | $0.13 \pm 0.02$ | $0.12 \pm 0.02$ | $\mathbf{0.12 \pm 0.01}$ |
| SF2-exp | 500 | $0.03 \pm 0.02$ | $0.02 \pm 0.01$ | $\mathbf{0.01 \pm 0.01}$ | $0.01 \pm 0.02$ | $0.02 \pm 0.02$ |
| SF2-exp | 1000 | $0.02 \pm 0.01$ | $0.02 \pm 0.01$ | $\mathbf{0.01 \pm 0.01}$ | $\mathbf{0.01 \pm 0.01}$ | $\mathbf{0.01 \pm 0.01}$ |
| SF3-exp | 500 | $0.05 \pm 0.02$ | $0.04 \pm 0.02$ | $0.03 \pm 0.01$ | $\mathbf{0.04 \pm 0.02}$ | $\mathbf{0.04 \pm 0.02}$ |
| SF3-exp | 1000 | $0.06 \pm 0.01$ | $0.05 \pm 0.01$ | $0.04 \pm 0.01$ | $\mathbf{0.03 \pm 0.01}$ | $\mathbf{0.03 \pm 0.01}$ |
| SF4-exp | 500 | $0.09 \pm 0.04$ | $0.09 \pm 0.04$ | $0.09 \pm 0.05$ | $0.07 \pm 0.04$ | $\mathbf{0.06 \pm 0.03}$ |
| SF4-exp | 1000 | $0.12 \pm 0.03$ | $0.11 \pm 0.02$ | $0.11 \pm 0.02$ | $\mathbf{0.09 \pm 0.03}$ | $0.10 \pm 0.02$ |
| SF2-gumbel | 500 | $0.03 \pm 0.02$ | $\mathbf{0.01 \pm 0.01}$ | $\mathbf{0.01 \pm 0.01}$ | $\mathbf{0.01 \pm 0.01}$ | $\mathbf{0.01 \pm 0.01}$ |
| SF2-gumbel | 1000 | $0.02 \pm 0.01$ | $0.02 \pm 0.01$ | $0.01 \pm 0.00$ | $\mathbf{0.01 \pm 0.01}$ | $0.02 \pm 0.01$ |
| SF3-gumbel | 500 | $0.06 \pm 0.02$ | $0.04 \pm 0.02$ | $0.03 \pm 0.03$ | $\mathbf{0.03 \pm 0.01}$ | $0.04 \pm 0.02$ |
| SF3-gumbel | 1000 | $0.06 \pm 0.01$ | $0.05 \pm 0.01$ | $0.04 \pm 0.02$ | $0.04 \pm 0.01$ | $\mathbf{0.03 \pm 0.01}$ |
| SF4-gumbel | 500 | $0.10 \pm 0.04$ | $0.09 \pm 0.05$ | $0.09 \pm 0.04$ | $\mathbf{0.08 \pm 0.04}$ | $0.08 \pm 0.05$ |
| SF4-gumbel | 1000 | $0.14 \pm 0.03$ | $0.12 \pm 0.03$ | $0.12 \pm 0.03$ | $0.11 \pm 0.03$ | $\mathbf{0.10 \pm 0.03}$ |

Table 7: DAG learning performance (measured in running time (seconds), the lower the better, best results in **bold**) of different algorithms on large scale (500-2000 nodes) graphs with different noise distributions. Our algorithm performs better than previous approaches.

| Graphs | Nodes | DAGMA | Order-1 | Order-2 | Order-3 | Order-4 |
|---|---|---|---|---|---|---|
| ER2-gauss | 500 | **171.79 ± 18.30** | 294.19 ± 62.47 | 493.73 ± 28.79 | 482.01 ± 17.59 | 512.86 ± 33.87 |
| ER2-gauss | 1000 | **364.95 ± 38.47** | 687.38 ± 69.45 | 1261.33 ± 47.66 | 1256.89 ± 33.76 | 1329.92 ± 315.91 |
| ER2-gauss | 2000 | **1187.17 ± 108.06** | 3515.56 ± 310.54 | 6063.65 ± 554.57 | 6300.09 ± 197.69 | 6360.65 ± 321.65 |
| ER3-gauss | 500 | **238.54 ± 58.34** | 394.95 ± 110.87 | 516.49 ± 14.45 | 524.93 ± 48.79 | 528.08 ± 49.59 |
| ER3-gauss | 1000 | **501.24 ± 59.30** | 1004.85 ± 162.25 | 1365.24 ± 21.10 | 1355.23 ± 139.98 | 1349.25 ± 84.01 |
| ER3-gauss | 2000 | **1636.23 ± 101.64** | 4903.56 ± 782.06 | 6037.67 ± 765.47 | 7072.04 ± 2168.01 | 6969.54 ± 724.52 |
| ER4-gauss | 500 | **347.06 ± 55.40** | 519.77 ± 56.88 | 532.34 ± 10.16 | 517.46 ± 12.95 | 534.98 ± 52.22 |
| ER4-gauss | 1000 | **798.01 ± 27.84** | 1328.32 ± 48.32 | 1318.45 ± 62.71 | 1348.08 ± 15.79 | 1312.53 ± 138.14 |
| ER4-gauss | 2000 | **2461.67 ± 1.49** | 6520.96 ± 311.23 | 6560.10 ± 13.55 | 7666.47 ± 2151.48 | 7067.87 ± 1011.11 |
| ER2-exp | 500 | **167.08 ± 25.75** | 291.87 ± 63.66 | 492.14 ± 30.49 | 496.83 ± 20.55 | 504.17 ± 22.59 |
| ER2-exp | 1000 | **359.97 ± 28.62** | 708.23 ± 61.00 | 1245.71 ± 71.43 | 1279.88 ± 41.63 | 1277.89 ± 45.39 |
| ER3-exp | 500 | **235.71 ± 56.44** | 440.70 ± 189.91 | 564.88 ± 145.09 | 559.54 ± 144.42 | 550.44 ± 129.03 |
| ER3-exp | 1000 | **515.88 ± 83.18** | 1100.18 ± 226.98 | 1371.65 ± 136.38 | 1401.29 ± 283.39 | 1503.85 ± 321.35 |
| ER4-exp | 500 | **358.94 ± 46.65** | 510.53 ± 46.91 | 513.70 ± 49.38 | 522.16 ± 15.09 | 508.01 ± 34.87 |
| ER4-exp | 1000 | **778.40 ± 51.69** | 1344.07 ± 26.71 | 1324.00 ± 111.40 | 1347.33 ± 21.06 | 1298.91 ± 100.30 |
| ER2-gumbel | 500 | **161.36 ± 24.90** | 255.86 ± 33.56 | 501.55 ± 85.14 | 490.99 ± 11.06 | 501.44 ± 23.26 |
| ER2-gumbel | 1000 | **330.53 ± 39.10** | 656.98 ± 62.61 | 1245.11 ± 47.50 | 1276.58 ± 15.41 | 1266.35 ± 38.53 |
| ER3-gumbel | 500 | **232.45 ± 50.71** | 381.03 ± 85.67 | 521.98 ± 8.54 | 514.79 ± 12.92 | 506.95 ± 18.01 |
| ER3-gumbel | 1000 | **525.92 ± 93.20** | 1013.67 ± 168.84 | 1331.88 ± 99.25 | 1302.93 ± 101.85 | 1369.44 ± 39.47 |
| ER4-gumbel | 500 | **366.06 ± 33.40** | 514.95 ± 57.57 | 540.93 ± 17.35 | 530.18 ± 20.49 | 519.87 ± 13.98 |
| ER4-gumbel | 1000 | **805.91 ± 31.96** | 1367.02 ± 36.36 | 1260.93 ± 137.27 | 1434.41 ± 154.30 | 1335.20 ± 61.78 |
| SF2-gauss | 500 | **171.79 ± 18.30** | 294.19 ± 62.47 | 493.73 ± 28.79 | 482.01 ± 17.59 | 512.86 ± 33.87 |
| SF2-gauss | 1000 | **364.95 ± 38.47** | 687.38 ± 69.45 | 1261.33 ± 47.66 | 1256.89 ± 33.76 | 1329.92 ± 315.91 |
| SF2-gauss | 2000 | **1187.17 ± 108.06** | 3515.56 ± 310.54 | 6063.65 ± 554.57 | 6300.09 ± 197.69 | 6360.65 ± 321.65 |
| SF3-gauss | 500 | **238.54 ± 58.34** | 394.95 ± 110.87 | 516.49 ± 14.45 | 524.93 ± 48.79 | 528.08 ± 49.59 |
| SF3-gauss | 1000 | **501.24 ± 59.30** | 1004.85 ± 162.25 | 1365.24 ± 21.10 | 1355.23 ± 139.98 | 1349.25 ± 84.01 |
| SF3-gauss | 2000 | **1636.23 ± 101.64** | 4903.56 ± 782.06 | 6037.67 ± 765.47 | 7072.04 ± 2168.01 | 6969.54 ± 724.52 |
| SF4-gauss | 500 | **347.06 ± 55.40** | 519.77 ± 56.88 | 532.34 ± 10.16 | 517.46 ± 12.95 | 534.98 ± 52.22 |
| SF4-gauss | 1000 | **798.01 ± 27.84** | 1328.32 ± 48.32 | 1318.45 ± 62.71 | 1348.08 ± 15.79 | 1312.53 ± 138.14 |
| SF4-gauss | 2000 | **2461.67 ± 1.49** | 6520.96 ± 311.23 | 6560.10 ± 13.55 | 7666.47 ± 2151.48 | 7067.87 ± 1011.11 |
| SF2-exp | 500 | **167.08 ± 25.75** | 291.87 ± 63.66 | 492.14 ± 30.49 | 496.83 ± 20.55 | 504.17 ± 22.59 |
| SF2-exp | 1000 | **359.97 ± 28.62** | 708.23 ± 61.00 | 1245.71 ± 71.43 | 1279.88 ± 41.63 | 1277.89 ± 45.39 |
| SF3-exp | 500 | **235.71 ± 56.44** | 440.70 ± 189.91 | 564.88 ± 145.09 | 559.54 ± 144.42 | 550.44 ± 129.03 |
| SF3-exp | 1000 | **515.88 ± 83.18** | 1100.18 ± 226.98 | 1371.65 ± 136.38 | 1401.29 ± 283.39 | 1503.85 ± 321.35 |
| SF4-exp | 500 | **358.94 ± 46.65** | 510.53 ± 46.91 | 513.70 ± 49.38 | 522.16 ± 15.09 | 508.01 ± 34.87 |
| SF4-exp | 1000 | **778.40 ± 51.69** | 1344.07 ± 26.71 | 1324.00 ± 111.40 | 1347.33 ± 21.06 | 1298.91 ± 100.30 |
| SF2-gumbel | 500 | **161.36 ± 24.90** | 255.86 ± 33.56 | 501.55 ± 85.14 | 490.99 ± 11.06 | 501.44 ± 23.26 |
| SF2-gumbel | 1000 | **330.53 ± 39.10** | 656.98 ± 62.61 | 1245.11 ± 47.50 | 1276.58 ± 15.41 | 1266.35 ± 38.53 |
| SF3-gumbel | 500 | **232.45 ± 50.71** | 381.03 ± 85.67 | 521.98 ± 8.54 | 514.79 ± 12.92 | 506.95 ± 18.01 |
| SF3-gumbel | 1000 | **525.92 ± 93.20** | 1013.67 ± 168.84 | 1331.88 ± 99.25 | 1302.93 ± 101.85 | 1369.44 ± 39.47 |
| SF4-gumbel | 500 | **366.06 ± 33.40** | 514.95 ± 57.57 | 540.93 ± 17.35 | 530.18 ± 20.49 | 519.87 ± 13.98 |
| SF4-gumbel | 1000 | **805.91 ± 31.96** | 1367.02 ± 36.36 | 1260.93 ± 137.27 | 1434.41 ± 154.30 | 1335.20 ± 61.78 |

Table 8: DAG learning performance (measured in structural hamming distance, the lower the better, best results in **bold**) of different algorithms on small scale (10-100 nodes) ER{2,3,4} graphs with different noise distributions. Our algorithm performs better than previous approaches.

| Graphs | Nodes | MMPC | GES | NOTEARS | DAGMA | Order-1 | Order-2 | Order-3 | Order-4 |
|---|---|---|---|---|---|---|---|---|---|
| ER2-gauss | 10 | 21.01 ± 3.77 | 14.66 ± 9.25 | 3.01 ± 2.67 | 2.01 ± 1.91 | **0.74 ± 1.22** | 0.93 ± 1.49 | 0.97 ± 1.65 | 1.27 ± 2.11 |
| ER2-gauss | 30 | 65.23 ± 9.26 | 57.95 ± 44.65 | 6.09 ± 6.31 | 3.76 ± 3.70 | 1.88 ± 2.97 | 1.57 ± 2.29 | **1.49 ± 2.32** | 1.87 ± 2.89 |
| ER2-gauss | 50 | 106.74 ± 11.35 | 90.49 ± 58.75 | 11.59 ± 10.25 | 4.21 ± 3.66 | 2.28 ± 3.06 | **2.05 ± 2.96** | 2.21 ± 2.98 | 2.43 ± 3.69 |
| ER2-gauss | 100 | 215.36 ± 14.83 | 155.61 ± 76.87 | 22.58 ± 17.69 | 7.45 ± 6.70 | 3.49 ± 3.97 | 3.64 ± 4.48 | 4.09 ± 4.40 | 4.01 ± 4.54 |
| ER3-gauss | 10 | 32.04 ± 3.39 | 26.39 ± 6.85 | 8.91 ± 4.33 | 6.96 ± 3.69 | 3.36 ± 2.91 | 3.35 ± 2.97 | **3.21 ± 2.83** | 3.51 ± 2.83 |
| ER3-gauss | 30 | 97.56 ± 9.71 | 158.57 ± 49.92 | 17.69 ± 12.29 | 9.22 ± 6.24 | **4.83 ± 4.56** | 5.33 ± 5.52 | 5.54 ± 5.71 | 4.96 ± 5.94 |
| ER3-gauss | 50 | 160.21 ± 12.67 | 285.88 ± 96.26 | 30.70 ± 18.48 | 13.05 ± 8.68 | 6.60 ± 6.56 | **6.43 ± 5.72** | 6.55 ± 6.09 | 6.44 ± 5.54 |
| ER3-gauss | 100 | 321.30 ± 18.01 | 506.91 ± 202.63 | 64.27 ± 33.28 | 21.66 ± 15.48 | 11.59 ± 9.90 | **10.69 ± 9.92** | 10.86 ± 9.02 | 11.28 ± 9.99 |
| ER4-gauss | 10 | 40.37 ± 2.24 | 29.07 ± 5.77 | 13.71 ± 3.93 | 11.57 ± 3.27 | 6.89 ± 3.39 | **6.75 ± 3.15** | 7.08 ± 3.40 | 7.04 ± 3.31 |
| ER4-gauss | 30 | 126.27 ± 10.70 | 222.49 ± 38.99 | 39.18 ± 21.79 | 20.67 ± 9.26 | 11.12 ± 7.76 | **10.54 ± 7.35** | 12.58 ± 8.81 | 12.81 ± 10.74 |
| ER4-gauss | 50 | 210.43 ± 13.13 | 495.14 ± 81.60 | 60.61 ± 25.25 | 24.96 ± 12.48 | 14.39 ± 9.72 | **12.89 ± 9.52** | 15.15 ± 10.31 | 16.22 ± 10.17 |
| ER4-gauss | 100 | 424.49 ± 20.13 | 1047.05 ± 250.05 | 118.16 ± 51.53 | 42.81 ± 23.74 | 28.31 ± 25.58 | 24.47 ± 24.31 | 23.22 ± 23.60 | **23.03 ± 21.51** |
| ER2-exp | 10 | 20.88 ± 3.70 | 15.23 ± 9.49 | 3.05 ± 2.64 | 2.19 ± 1.87 | **0.80 ± 1.34** | 0.90 ± 1.57 | 1.03 ± 1.74 | 1.35 ± 2.30 |
| ER2-exp | 30 | 64.98 ± 9.21 | 59.37 ± 46.71 | 6.59 ± 7.08 | 3.83 ± 4.10 | 2.22 ± 3.63 | 1.85 ± 2.91 | **1.40 ± 2.37** | 2.09 ± 3.28 |
| ER2-exp | 50 | 106.77 ± 10.85 | 95.43 ± 56.48 | 11.51 ± 9.70 | 4.21 ± 3.43 | 2.12 ± 2.75 | **1.97 ± 2.47** | 2.62 ± 3.01 | 2.80 ± 3.53 |
| ER2-exp | 100 | 215.70 ± 14.80 | 159.55 ± 77.19 | 21.87 ± 17.48 | 7.31 ± 6.17 | **3.71 ± 4.77** | 4.11 ± 4.58 | 4.54 ± 4.72 | 4.82 ± 5.15 |
| ER3-exp | 10 | 31.90 ± 3.41 | 26.29 ± 6.80 | 9.32 ± 4.42 | 6.98 ± 3.43 | 3.57 ± 2.91 | 3.43 ± 2.96 | **3.10 ± 2.89** | 3.68 ± 2.82 |
| ER3-exp | 30 | 97.74 ± 9.81 | 154.20 ± 48.45 | 17.54 ± 11.51 | 9.11 ± 5.51 | 5.07 ± 4.47 | 5.51 ± 5.51 | 5.60 ± 5.73 | **4.81 ± 5.76** |
| ER3-exp | 50 | 160.16 ± 12.49 | 288.43 ± 99.00 | 31.32 ± 19.96 | 14.16 ± 9.73 | 7.92 ± 7.38 | **6.73 ± 5.94** | 7.15 ± 6.32 | 7.18 ± 5.69 |
| ER3-exp | 100 | 321.66 ± 17.89 | 494.33 ± 188.84 | 66.39 ± 30.84 | 22.02 ± 15.03 | 12.26 ± 9.06 | 12.44 ± 11.03 | 11.52 ± 9.37 | **11.14 ± 8.36** |
| ER4-exp | 10 | 40.44 ± 2.15 | 28.79 ± 5.56 | 13.92 ± 3.89 | 11.85 ± 3.42 | 7.01 ± 3.44 | **6.93 ± 3.36** | 7.07 ± 3.55 | 7.10 ± 3.32 |
| ER4-exp | 30 | 125.98 ± 10.62 | 221.86 ± 39.40 | 36.55 ± 18.74 | 21.00 ± 10.26 | **10.73 ± 7.83** | 11.55 ± 7.70 | 13.19 ± 9.95 | 12.87 ± 11.09 |
| ER4-exp | 50 | 210.42 ± 13.43 | 494.91 ± 77.20 | 60.90 ± 25.93 | 27.10 ± 12.94 | 14.76 ± 9.50 | **13.83 ± 9.46** | 14.92 ± 10.98 | 16.44 ± 12.95 |
| ER4-exp | 100 | 424.65 ± 19.82 | 1043.47 ± 239.69 | 116.91 ± 48.63 | 42.64 ± 23.78 | 27.02 ± 24.07 | 24.32 ± 22.06 | **23.39 ± 22.89** | 24.12 ± 22.40 |
| ER2-gumbel | 10 | 20.89 ± 3.72 | 16.07 ± 8.92 | 1.69 ± 2.04 | 1.11 ± 1.52 | **0.67 ± 1.36** | 0.82 ± 1.60 | 0.79 ± 1.65 | 1.00 ± 2.24 |
| ER2-gumbel | 30 | 64.88 ± 9.55 | 61.33 ± 43.70 | 5.65 ± 7.35 | 2.10 ± 2.92 | 1.76 ± 2.79 | **1.69 ± 2.90** | 2.13 ± 3.06 | 2.28 ± 3.74 |
| ER2-gumbel | 50 | 106.77 ± 10.76 | 92.35 ± 57.81 | 10.24 ± 10.44 | 2.81 ± 3.30 | **2.43 ± 3.03** | 2.64 ± 3.81 | 3.41 ± 4.63 | 3.77 ± 4.56 |
| ER2-gumbel | 100 | 215.52 ± 15.57 | 161.75 ± 89.91 | 22.39 ± 19.17 | 4.89 ± 5.94 | **4.21 ± 5.17** | 4.78 ± 5.64 | 5.59 ± 5.42 | 6.63 ± 6.00 |
| ER3-gumbel | 10 | 31.95 ± 3.37 | 26.29 ± 6.69 | 5.93 ± 3.95 | 4.05 ± 2.96 | 1.90 ± 2.39 | 2.44 ± 2.71 | 2.09 ± 2.31 | **1.75 ± 2.23** |
| ER3-gumbel | 30 | 97.72 ± 9.72 | 158.60 ± 49.62 | 13.44 ± 11.65 | 4.97 ± 4.07 | **3.72 ± 4.11** | 4.13 ± 4.29 | 5.99 ± 6.27 | 5.96 ± 6.92 |
| ER3-gumbel | 50 | 160.35 ± 12.66 | 281.98 ± 102.11 | 27.28 ± 20.76 | 9.43 ± 7.56 | 7.29 ± 7.07 | **5.79 ± 5.76** | 7.89 ± 6.40 | 8.17 ± 8.16 |
| ER3-gumbel | 100 | 321.87 ± 18.02 | 496.90 ± 193.46 | 60.12 ± 29.08 | 15.06 ± 13.15 | 12.62 ± 12.83 | 12.46 ± 12.84 | **12.45 ± 9.95** | 13.87 ± 10.60 |
| ER4-gumbel | 10 | 40.48 ± 2.13 | 28.69 ± 5.93 | 10.24 ± 4.08 | 7.75 ± 2.94 | **3.96 ± 3.11** | 4.55 ± 3.54 | 4.92 ± 3.38 | 4.90 ± 3.46 |
| ER4-gumbel | 30 | 126.04 ± 10.54 | 219.39 ± 43.15 | 29.45 ± 19.08 | 12.35 ± 7.64 | 8.37 ± 6.70 | **8.14 ± 7.53** | 11.29 ± 9.49 | 11.84 ± 9.91 |
| ER4-gumbel | 50 | 210.16 ± 13.34 | 499.02 ± 78.43 | 55.57 ± 28.67 | 18.03 ± 15.20 | 12.07 ± 9.36 | **10.92 ± 9.04** | 13.44 ± 10.94 | 15.06 ± 11.37 |
| ER4-gumbel | 100 | 424.19 ± 20.06 | 1031.32 ± 243.38 | 114.40 ± 52.86 | 29.94 ± 22.69 | 27.00 ± 28.51 | 22.36 ± 24.07 | **21.24 ± 23.58** | 25.37 ± 25.67 |
| SF2-gauss | 10 | 14.82 ± 1.99 | 5.01 ± 6.58 | 3.01 ± 2.67 | 2.01 ± 1.91 | **0.74 ± 1.22** | 0.93 ± 1.49 | 0.97 ± 1.65 | 1.27 ± 2.11 |
| SF2-gauss | 30 | 54.01 ± 3.54 | 19.59 ± 19.48 | 6.09 ± 6.31 | 3.76 ± 3.70 | 1.88 ± 2.97 | 1.57 ± 2.29 | **1.49 ± 2.32** | 1.87 ± 2.89 |
| SF2-gauss | 50 | 97.30 ± 5.13 | 43.39 ± 40.68 | 11.59 ± 10.25 | 4.21 ± 3.66 | 2.28 ± 3.06 | **2.05 ± 2.96** | 2.21 ± 2.98 | 2.43 ± 3.69 |
| SF2-gauss | 100 | 215.89 ± 8.24 | 106.51 ± 62.29 | 22.58 ± 17.69 | 7.45 ± 6.70 | 3.49 ± 3.97 | 3.64 ± 4.48 | 4.09 ± 4.40 | 4.01 ± 4.54 |
| SF3-gauss | 10 | 16.61 ± 2.77 | 7.78 ± 8.34 | 8.91 ± 4.33 | 6.96 ± 3.69 | 3.36 ± 2.91 | 3.35 ± 2.97 | **3.21 ± 2.83** | 3.51 ± 2.83 |
| SF3-gauss | 30 | 65.90 ± 7.29 | 31.51 ± 32.61 | 17.69 ± 12.29 | 9.22 ± 6.24 | **4.83 ± 4.56** | 5.33 ± 5.52 | 5.54 ± 5.71 | 4.96 ± 5.94 |
| SF3-gauss | 50 | 119.92 ± 10.81 | 69.43 ± 58.72 | 30.70 ± 18.48 | 13.05 ± 8.68 | 6.60 ± 6.56 | **6.43 ± 5.72** | 6.55 ± 6.09 | 6.44 ± 5.54 |
| SF3-gauss | 100 | 271.55 ± 16.84 | 157.11 ± 99.22 | 64.27 ± 33.28 | 21.66 ± 15.48 | 11.59 ± 9.90 | **10.69 ± 9.92** | 10.86 ± 9.02 | 11.28 ± 9.99 |
| SF4-gauss | 10 | 17.53 ± 3.20 | 6.56 ± 6.84 | 13.71 ± 3.93 | 11.57 ± 3.27 | 6.89 ± 3.39 | **6.75 ± 3.15** | 7.08 ± 3.40 | 7.04 ± 3.31 |
| SF4-gauss | 30 | 73.81 ± 9.41 | 46.72 ± 41.14 | 39.18 ± 21.79 | 20.67 ± 9.26 | 11.12 ± 7.76 | **10.54 ± 7.35** | 12.58 ± 8.81 | 12.81 ± 10.74 |
| SF4-gauss | 50 | 137.43 ± 13.92 | 90.76 ± 69.01 | 60.61 ± 25.25 | 24.96 ± 12.48 | 14.39 ± 9.72 | **12.89 ± 9.52** | 15.15 ± 10.31 | 16.22 ± 10.17 |
| SF4-gauss | 100 | 315.31 ± 24.17 | 174.28 ± 109.72 | 118.16 ± 51.53 | 42.81 ± 23.74 | 28.31 ± 25.58 | 24.47 ± 24.31 | 23.22 ± 23.60 | **23.03 ± 21.51** |
| SF2-exp | 10 | 14.72 ± 1.88 | 5.18 ± 6.51 | 3.05 ± 2.64 | 2.19 ± 1.87 | **0.80 ± 1.34** | 0.90 ± 1.57 | 1.03 ± 1.74 | 1.35 ± 2.30 |
| SF2-exp | 30 | 54.19 ± 3.69 | 23.86 ± 24.42 | 6.59 ± 7.08 | 3.83 ± 4.10 | 2.22 ± 3.63 | 1.85 ± 2.91 | **1.40 ± 2.37** | 2.09 ± 3.28 |
| SF2-exp | 50 | 97.15 ± 5.42 | 47.29 ± 42.59 | 11.51 ± 9.70 | 4.21 ± 3.43 | 2.12 ± 2.75 | **1.97 ± 2.47** | 2.62 ± 3.01 | 2.80 ± 3.53 |
| SF2-exp | 100 | 216.20 ± 7.98 | 113.30 ± 63.41 | 21.87 ± 17.48 | 7.31 ± 6.17 | **3.71 ± 4.77** | 4.11 ± 4.58 | 4.54 ± 4.72 | 4.82 ± 5.15 |
| SF3-exp | 10 | 16.56 ± 2.77 | 6.54 ± 7.60 | 9.32 ± 4.42 | 6.98 ± 3.43 | 3.57 ± 2.91 | 3.43 ± 2.96 | **3.10 ± 2.89** | 3.68 ± 2.82 |
| SF3-exp | 30 | 65.72 ± 7.38 | 32.67 ± 36.15 | 17.54 ± 11.51 | 9.11 ± 5.51 | 5.07 ± 4.47 | 5.51 ± 5.51 | 5.60 ± 5.73 | **4.81 ± 5.76** |
| SF3-exp | 50 | 119.86 ± 10.63 | 64.53 ± 55.26 | 31.32 ± 19.96 | 14.16 ± 9.73 | 7.92 ± 7.38 | **6.73 ± 5.94** | 7.15 ± 6.32 | 7.18 ± 5.69 |
| SF3-exp | 100 | 272.14 ± 16.44 | 157.53 ± 109.02 | 66.39 ± 30.84 | 22.02 ± 15.03 | 12.26 ± 9.06 | 12.44 ± 11.03 | 11.52 ± 9.37 | **11.14 ± 8.36** |
| SF4-exp | 10 | 17.46 ± 3.13 | 6.86 ± 6.78 | 13.92 ± 3.89 | 11.85 ± 3.42 | 7.01 ± 3.44 | **6.93 ± 3.36** | 7.07 ± 3.55 | 7.10 ± 3.32 |
| SF4-exp | 30 | 73.49 ± 8.97 | 47.03 ± 40.43 | 36.55 ± 18.74 | 21.00 ± 10.26 | **10.73 ± 7.83** | 11.55 ± 7.70 | 13.19 ± 9.95 | 12.87 ± 11.09 |
| SF4-exp | 50 | 137.96 ± 14.04 | 84.43 ± 63.15 | 60.90 ± 25.93 | 27.10 ± 12.94 | 14.76 ± 9.50 | **13.83 ± 9.46** | 14.92 ± 10.98 | 16.44 ± 12.95 |
| SF4-exp | 100 | 314.36 ± 24.83 | 176.43 ± 114.42 | 116.91 ± 48.63 | 42.64 ± 23.78 | 27.02 ± 24.07 | 24.32 ± 22.06 | **23.39 ± 22.89** | 24.12 ± 22.40 |
| SF2-gumbel | 10 | 14.72 ± 1.91 | 5.28 ± 6.51 | 1.69 ± 2.04 | 1.11 ± 1.52 | **0.67 ± 1.36** | 0.82 ± 1.60 | 0.79 ± 1.65 | 1.00 ± 2.24 |
| SF2-gumbel | 30 | 54.01 ± 3.73 | 23.38 ± 23.10 | 5.65 ± 7.35 | 2.10 ± 2.92 | 1.76 ± 2.79 | **1.69 ± 2.90** | 2.13 ± 3.06 | 2.28 ± 3.74 |
| SF2-gumbel | 50 | 97.21 ± 5.45 | 43.98 ± 37.99 | 10.24 ± 10.44 | 2.81 ± 3.30 | **2.43 ± 3.03** | 2.64 ± 3.81 | 3.41 ± 4.63 | 3.77 ± 4.56 |
| SF2-gumbel | 100 | 217.44 ± 8.29 | 112.49 ± 66.23 | 22.39 ± 19.17 | 4.89 ± 5.94 | **4.21 ± 5.17** | 4.78 ± 5.64 | 5.59 ± 5.42 | 6.63 ± 6.00 |
| SF3-gumbel | 10 | 16.51 ± 2.72 | 6.94 ± 7.96 | 5.93 ± 3.95 | 4.05 ± 2.96 | 1.90 ± 2.39 | 2.44 ± 2.71 | 2.09 ± 2.31 | **1.75 ± 2.23** |
| SF3-gumbel | 30 | 65.91 ± 7.36 | 33.87 ± 35.72 | 13.44 ± 11.65 | 4.97 ± 4.07 | **3.72 ± 4.11** | 4.13 ± 4.29 | 5.99 ± 6.27 | 5.96 ± 6.92 |
| SF3-gumbel | 50 | 120.11 ± 11.19 | 65.40 ± 55.66 | 27.28 ± 20.76 | 9.43 ± 7.56 | 7.29 ± 7.07 | **5.79 ± 5.76** | 7.89 ± 6.40 | 8.17 ± 8.16 |
| SF3-gumbel | 100 | 272.00 ± 16.44 | 149.45 ± 100.23 | 60.12 ± 29.08 | 15.06 ± 13.15 | 12.62 ± 12.83 | 12.46 ± 12.84 | **12.45 ± 9.95** | 13.87 ± 10.60 |
| SF4-gumbel | 10 | 17.47 ± 3.17 | 7.27 ± 7.12 | 10.24 ± 4.08 | 7.75 ± 2.94 | **3.96 ± 3.11** | 4.55 ± 3.54 | 4.92 ± 3.38 | 4.90 ± 3.46 |
| SF4-gumbel | 30 | 73.54 ± 9.23 | 45.15 ± 41.75 | 29.45 ± 19.08 | 12.35 ± 7.64 | 8.37 ± 6.70 | **8.14 ± 7.53** | 11.29 ± 9.49 | 11.84 ± 9.91 |
| SF4-gumbel | 50 | 137.73 ± 13.67 | 81.88 ± 67.58 | 55.57 ± 28.67 | 18.03 ± 15.20 | 12.07 ± 9.36 | **10.92 ± 9.04** | 13.44 ± 10.94 | 15.06 ± 11.37 |
| SF4-gumbel | 100 | 315.13 ± 23.64 | 185.00 ± 111.88 | 114.40 ± 52.86 | 29.94 ± 22.69 | 27.00 ± 28.51 | 22.36 ± 24.07 | **21.24 ± 23.58** | 25.37 ± 25.67 |

Table 9: DAG learning performance (measured in true positive rate, the higher the better, best results in **bold**) of different algorithms on small scale (10-100 nodes) ER{2,3,4} graphs with different noise distributions. Our algorithm performs better than previous approaches.

| Graphs | Nodes | MMPC | GES | NOTEARS | DAGMA | Order-1 | Order-2 | Order-3 | Order-4 |
|---|---|---|---|---|---|---|---|---|---|
| ER2-gauss | 10 | 0.59 ± 0.14 | 0.70 ± 0.23 | 0.88 ± 0.09 | 0.91 ± 0.07 | **0.97 ± 0.04** | 0.97 ± 0.04 | 0.97 ± 0.04 | 0.96 ± 0.05 |
| ER2-gauss | 30 | 0.53 ± 0.11 | 0.80 ± 0.14 | 0.93 ± 0.05 | 0.95 ± 0.04 | 0.98 ± 0.02 | 0.99 ± 0.02 | **0.99 ± 0.02** | 0.98 ± 0.02 |
| ER2-gauss | 50 | 0.55 ± 0.09 | 0.83 ± 0.10 | 0.93 ± 0.05 | 0.97 ± 0.02 | 0.99 ± 0.01 | 0.99 ± 0.01 | 0.99 ± 0.01 | **0.99 ± 0.01** |
| ER2-gauss | 100 | 0.58 ± 0.07 | 0.87 ± 0.06 | 0.94 ± 0.04 | 0.97 ± 0.02 | 0.99 ± 0.01 | **0.99 ± 0.01** | 0.99 ± 0.01 | 0.99 ± 0.01 |
| ER3-gauss | 10 | 0.34 ± 0.07 | 0.48 ± 0.15 | 0.75 ± 0.11 | 0.80 ± 0.10 | 0.91 ± 0.08 | 0.91 ± 0.08 | **0.91 ± 0.07** | 0.91 ± 0.07 |
| ER3-gauss | 30 | 0.29 ± 0.07 | 0.63 ± 0.13 | 0.88 ± 0.06 | 0.92 ± 0.04 | 0.97 ± 0.02 | 0.97 ± 0.03 | 0.97 ± 0.03 | **0.97 ± 0.03** |
| ER3-gauss | 50 | 0.30 ± 0.06 | 0.69 ± 0.10 | 0.89 ± 0.06 | 0.93 ± 0.03 | 0.98 ± 0.02 | 0.98 ± 0.02 | **0.98 ± 0.02** | 0.98 ± 0.02 |
| ER3-gauss | 100 | 0.33 ± 0.04 | 0.78 ± 0.07 | 0.89 ± 0.04 | 0.95 ± 0.02 | 0.98 ± 0.01 | 0.98 ± 0.01 | 0.98 ± 0.01 | **0.98 ± 0.01** |
| ER4-gauss | 10 | 0.28 ± 0.05 | 0.43 ± 0.14 | 0.68 ± 0.09 | 0.73 ± 0.07 | 0.84 ± 0.07 | **0.85 ± 0.07** | 0.84 ± 0.07 | 0.84 ± 0.07 |
| ER4-gauss | 30 | 0.18 ± 0.04 | 0.54 ± 0.10 | 0.80 ± 0.08 | 0.87 ± 0.04 | 0.94 ± 0.03 | **0.95 ± 0.03** | 0.94 ± 0.03 | 0.94 ± 0.03 |
| ER4-gauss | 50 | 0.18 ± 0.03 | 0.57 ± 0.09 | 0.83 ± 0.05 | 0.91 ± 0.03 | 0.96 ± 0.02 | **0.96 ± 0.02** | 0.96 ± 0.02 | 0.96 ± 0.02 |
| ER4-gauss | 100 | 0.20 ± 0.03 | 0.68 ± 0.07 | 0.85 ± 0.06 | 0.93 ± 0.03 | 0.97 ± 0.02 | 0.97 ± 0.02 | 0.97 ± 0.02 | **0.97 ± 0.01** |
| ER2-exp | 10 | 0.58 ± 0.14 | 0.67 ± 0.24 | 0.87 ± 0.09 | 0.91 ± 0.07 | **0.97 ± 0.04** | 0.97 ± 0.04 | 0.97 ± 0.04 | 0.96 ± 0.05 |
| ER2-exp | 30 | 0.54 ± 0.11 | 0.80 ± 0.14 | 0.93 ± 0.05 | 0.95 ± 0.04 | 0.98 ± 0.02 | 0.98 ± 0.02 | **0.99 ± 0.02** | 0.98 ± 0.02 |
| ER2-exp | 50 | 0.56 ± 0.10 | 0.82 ± 0.09 | 0.93 ± 0.05 | 0.97 ± 0.02 | 0.99 ± 0.01 | **0.99 ± 0.01** | 0.99 ± 0.01 | 0.99 ± 0.01 |
| ER2-exp | 100 | 0.59 ± 0.07 | 0.87 ± 0.06 | 0.94 ± 0.04 | 0.97 ± 0.02 | **0.99 ± 0.01** | 0.99 ± 0.01 | 0.99 ± 0.01 | 0.99 ± 0.01 |
| ER3-exp | 10 | 0.34 ± 0.08 | 0.48 ± 0.16 | 0.75 ± 0.11 | 0.81 ± 0.09 | 0.90 ± 0.07 | 0.91 ± 0.08 | **0.92 ± 0.07** | 0.90 ± 0.07 |
| ER3-exp | 30 | 0.29 ± 0.07 | 0.64 ± 0.13 | 0.88 ± 0.06 | 0.92 ± 0.04 | 0.97 ± 0.02 | 0.96 ± 0.03 | 0.97 ± 0.03 | **0.97 ± 0.03** |
| ER3-exp | 50 | 0.31 ± 0.06 | 0.69 ± 0.10 | 0.88 ± 0.06 | 0.93 ± 0.03 | 0.97 ± 0.02 | **0.98 ± 0.02** | 0.97 ± 0.02 | 0.97 ± 0.01 |
| ER3-exp | 100 | 0.33 ± 0.04 | 0.78 ± 0.07 | 0.89 ± 0.04 | 0.95 ± 0.02 | 0.98 ± 0.01 | 0.98 ± 0.01 | 0.98 ± 0.01 | **0.98 ± 0.01** |
| ER4-exp | 10 | 0.28 ± 0.05 | 0.44 ± 0.13 | 0.68 ± 0.09 | 0.72 ± 0.07 | 0.84 ± 0.08 | **0.84 ± 0.07** | 0.84 ± 0.08 | 0.84 ± 0.07 |
| ER4-exp | 30 | 0.17 ± 0.04 | 0.54 ± 0.11 | 0.81 ± 0.07 | 0.87 ± 0.05 | **0.94 ± 0.03** | 0.94 ± 0.03 | 0.94 ± 0.03 | 0.94 ± 0.04 |
| ER4-exp | 50 | 0.18 ± 0.03 | 0.58 ± 0.09 | 0.83 ± 0.05 | 0.90 ± 0.03 | 0.96 ± 0.02 | **0.96 ± 0.02** | 0.96 ± 0.02 | 0.96 ± 0.02 |
| ER4-exp | 100 | 0.20 ± 0.03 | 0.68 ± 0.07 | 0.85 ± 0.06 | 0.93 ± 0.03 | 0.97 ± 0.02 | 0.97 ± 0.02 | **0.97 ± 0.02** | 0.97 ± 0.01 |
| ER2-gumbel | 10 | 0.57 ± 0.15 | 0.64 ± 0.24 | 0.93 ± 0.08 | 0.96 ± 0.05 | **0.98 ± 0.04** | 0.98 ± 0.04 | 0.98 ± 0.04 | 0.98 ± 0.04 |
| ER2-gumbel | 30 | 0.54 ± 0.11 | 0.78 ± 0.13 | 0.95 ± 0.05 | 0.98 ± 0.02 | 0.99 ± 0.02 | **0.99 ± 0.02** | 0.99 ± 0.02 | 0.99 ± 0.02 |
| ER2-gumbel | 50 | 0.56 ± 0.10 | 0.83 ± 0.09 | 0.95 ± 0.04 | 0.98 ± 0.02 | 0.99 ± 0.01 | **0.99 ± 0.01** | 0.99 ± 0.01 | 0.99 ± 0.01 |
| ER2-gumbel | 100 | 0.58 ± 0.07 | 0.87 ± 0.06 | 0.95 ± 0.03 | 0.99 ± 0.02 | **0.99 ± 0.01** | 0.99 ± 0.01 | 0.99 ± 0.01 | 0.99 ± 0.01 |
| ER3-gumbel | 10 | 0.35 ± 0.08 | 0.49 ± 0.16 | 0.84 ± 0.10 | 0.89 ± 0.07 | 0.95 ± 0.06 | 0.94 ± 0.06 | 0.95 ± 0.06 | **0.95 ± 0.06** |
| ER3-gumbel | 30 | 0.28 ± 0.07 | 0.63 ± 0.13 | 0.92 ± 0.06 | 0.96 ± 0.02 | **0.98 ± 0.02** | 0.98 ± 0.02 | 0.97 ± 0.02 | 0.97 ± 0.02 |
| ER3-gumbel | 50 | 0.30 ± 0.06 | 0.69 ± 0.10 | 0.92 ± 0.05 | 0.96 ± 0.02 | 0.98 ± 0.01 | **0.98 ± 0.01** | 0.98 ± 0.01 | 0.98 ± 0.02 |
| ER3-gumbel | 100 | 0.33 ± 0.04 | 0.78 ± 0.08 | 0.92 ± 0.03 | 0.97 ± 0.02 | 0.98 ± 0.01 | 0.99 ± 0.01 | **0.99 ± 0.01** | 0.99 ± 0.01 |
| ER4-gumbel | 10 | 0.28 ± 0.05 | 0.44 ± 0.13 | 0.77 ± 0.09 | 0.82 ± 0.07 | **0.91 ± 0.07** | 0.90 ± 0.07 | 0.89 ± 0.07 | 0.89 ± 0.08 |
| ER4-gumbel | 30 | 0.17 ± 0.04 | 0.54 ± 0.11 | 0.87 ± 0.07 | 0.93 ± 0.03 | 0.97 ± 0.02 | **0.97 ± 0.02** | 0.96 ± 0.03 | 0.96 ± 0.03 |
| ER4-gumbel | 50 | 0.18 ± 0.03 | 0.57 ± 0.09 | 0.87 ± 0.05 | 0.95 ± 0.03 | 0.98 ± 0.02 | **0.98 ± 0.01** | 0.97 ± 0.02 | 0.97 ± 0.02 |
| ER4-gumbel | 100 | 0.20 ± 0.03 | 0.69 ± 0.07 | 0.88 ± 0.06 | 0.96 ± 0.02 | 0.98 ± 0.02 | 0.98 ± 0.02 | **0.98 ± 0.01** | 0.98 ± 0.01 |
| SF2-gauss | 10 | 0.78 ± 0.11 | 0.91 ± 0.19 | 0.88 ± 0.09 | 0.91 ± 0.07 | **0.97 ± 0.04** | 0.97 ± 0.04 | 0.97 ± 0.04 | 0.96 ± 0.05 |
| SF2-gauss | 30 | 0.62 ± 0.09 | 0.95 ± 0.08 | 0.93 ± 0.05 | 0.95 ± 0.04 | 0.98 ± 0.02 | 0.99 ± 0.02 | **0.99 ± 0.02** | 0.98 ± 0.02 |
| SF2-gauss | 50 | 0.57 ± 0.08 | 0.92 ± 0.11 | 0.93 ± 0.05 | 0.97 ± 0.02 | 0.99 ± 0.01 | 0.99 ± 0.01 | 0.99 ± 0.01 | **0.99 ± 0.01** |
| SF2-gauss | 100 | 0.54 ± 0.06 | 0.92 ± 0.08 | 0.94 ± 0.04 | 0.97 ± 0.02 | 0.99 ± 0.01 | **0.99 ± 0.01** | 0.99 ± 0.01 | 0.99 ± 0.01 |
| SF3-gauss | 10 | 0.73 ± 0.13 | 0.86 ± 0.19 | 0.75 ± 0.11 | 0.80 ± 0.10 | 0.91 ± 0.08 | 0.91 ± 0.08 | **0.91 ± 0.07** | 0.91 ± 0.07 |
| SF3-gauss | 30 | 0.53 ± 0.10 | 0.92 ± 0.09 | 0.88 ± 0.06 | 0.92 ± 0.04 | 0.97 ± 0.02 | 0.97 ± 0.03 | 0.97 ± 0.03 | **0.97 ± 0.03** |
| SF3-gauss | 50 | 0.47 ± 0.07 | 0.90 ± 0.10 | 0.89 ± 0.06 | 0.93 ± 0.03 | 0.98 ± 0.02 | 0.98 ± 0.02 | 0.98 ± 0.02 | **0.98 ± 0.02** |
| SF3-gauss | 100 | 0.43 ± 0.05 | 0.90 ± 0.09 | 0.89 ± 0.04 | 0.95 ± 0.02 | 0.98 ± 0.01 | 0.98 ± 0.01 | 0.98 ± 0.01 | **0.98 ± 0.01** |
| SF4-gauss | 10 | 0.71 ± 0.13 | 0.89 ± 0.15 | 0.68 ± 0.09 | 0.73 ± 0.07 | 0.84 ± 0.07 | **0.85 ± 0.07** | 0.84 ± 0.07 | 0.84 ± 0.07 |
| SF4-gauss | 30 | 0.49 ± 0.08 | 0.87 ± 0.12 | 0.80 ± 0.08 | 0.87 ± 0.04 | 0.94 ± 0.03 | **0.95 ± 0.03** | 0.94 ± 0.03 | 0.94 ± 0.03 |
| SF4-gauss | 50 | 0.42 ± 0.07 | 0.88 ± 0.10 | 0.83 ± 0.05 | 0.91 ± 0.03 | 0.96 ± 0.02 | **0.96 ± 0.02** | 0.96 ± 0.02 | 0.96 ± 0.02 |
| SF4-gauss | 100 | 0.39 ± 0.05 | 0.90 ± 0.09 | 0.85 ± 0.06 | 0.93 ± 0.03 | 0.97 ± 0.02 | 0.97 ± 0.02 | 0.97 ± 0.02 | **0.97 ± 0.01** |
| SF2-exp | 10 | 0.79 ± 0.11 | 0.91 ± 0.18 | 0.87 ± 0.09 | 0.91 ± 0.07 | **0.97 ± 0.04** | 0.97 ± 0.04 | 0.97 ± 0.04 | 0.96 ± 0.05 |
| SF2-exp | 30 | 0.62 ± 0.09 | 0.93 ± 0.11 | 0.93 ± 0.05 | 0.95 ± 0.04 | 0.98 ± 0.02 | 0.98 ± 0.02 | **0.99 ± 0.02** | 0.98 ± 0.02 |
| SF2-exp | 50 | 0.57 ± 0.08 | 0.92 ± 0.11 | 0.93 ± 0.05 | 0.97 ± 0.02 | 0.99 ± 0.01 | **0.99 ± 0.01** | 0.99 ± 0.01 | 0.99 ± 0.01 |
| SF2-exp | 100 | 0.54 ± 0.06 | 0.91 ± 0.08 | 0.94 ± 0.04 | 0.97 ± 0.02 | **0.99 ± 0.01** | 0.99 ± 0.01 | 0.99 ± 0.01 | 0.99 ± 0.01 |
| SF3-exp | 10 | 0.74 ± 0.13 | 0.90 ± 0.17 | 0.75 ± 0.11 | 0.81 ± 0.09 | 0.90 ± 0.07 | 0.91 ± 0.08 | **0.92 ± 0.07** | 0.90 ± 0.07 |
| SF3-exp | 30 | 0.53 ± 0.10 | 0.92 ± 0.10 | 0.88 ± 0.06 | 0.92 ± 0.04 | 0.97 ± 0.02 | 0.96 ± 0.03 | 0.97 ± 0.03 | **0.97 ± 0.03** |
| SF3-exp | 50 | 0.47 ± 0.07 | 0.91 ± 0.09 | 0.88 ± 0.06 | 0.93 ± 0.03 | 0.97 ± 0.02 | **0.98 ± 0.02** | 0.97 ± 0.02 | 0.97 ± 0.01 |
| SF3-exp | 100 | 0.43 ± 0.05 | 0.89 ± 0.10 | 0.89 ± 0.04 | 0.95 ± 0.02 | 0.98 ± 0.01 | 0.98 ± 0.01 | 0.98 ± 0.01 | **0.98 ± 0.01** |
| SF4-exp | 10 | 0.71 ± 0.13 | 0.88 ± 0.16 | 0.68 ± 0.09 | 0.72 ± 0.07 | 0.84 ± 0.08 | **0.84 ± 0.07** | 0.84 ± 0.08 | 0.84 ± 0.07 |
| SF4-exp | 30 | 0.49 ± 0.08 | 0.88 ± 0.12 | 0.81 ± 0.07 | 0.87 ± 0.05 | **0.94 ± 0.03** | 0.94 ± 0.03 | 0.94 ± 0.03 | 0.94 ± 0.04 |
| SF4-exp | 50 | 0.42 ± 0.07 | 0.89 ± 0.10 | 0.83 ± 0.05 | 0.90 ± 0.03 | 0.96 ± 0.02 | **0.96 ± 0.02** | 0.96 ± 0.02 | 0.96 ± 0.02 |
| SF4-exp | 100 | 0.39 ± 0.05 | 0.90 ± 0.10 | 0.85 ± 0.06 | 0.93 ± 0.03 | 0.97 ± 0.02 | 0.97 ± 0.02 | **0.97 ± 0.02** | 0.97 ± 0.01 |
| SF2-gumbel | 10 | 0.78 ± 0.11 | 0.91 ± 0.18 | 0.93 ± 0.08 | 0.96 ± 0.05 | **0.98 ± 0.04** | 0.98 ± 0.04 | 0.98 ± 0.04 | 0.98 ± 0.04 |
| SF2-gumbel | 30 | 0.62 ± 0.09 | 0.94 ± 0.08 | 0.95 ± 0.05 | 0.98 ± 0.02 | 0.99 ± 0.02 | **0.99 ± 0.02** | 0.99 ± 0.02 | 0.99 ± 0.02 |
| SF2-gumbel | 50 | 0.57 ± 0.08 | 0.92 ± 0.10 | 0.95 ± 0.04 | 0.98 ± 0.02 | 0.99 ± 0.01 | **0.99 ± 0.01** | 0.99 ± 0.01 | 0.99 ± 0.01 |
| SF2-gumbel | 100 | 0.53 ± 0.06 | 0.92 ± 0.08 | 0.95 ± 0.03 | 0.99 ± 0.02 | **0.99 ± 0.01** | 0.99 ± 0.01 | 0.99 ± 0.01 | 0.99 ± 0.01 |
| SF3-gumbel | 10 | 0.73 ± 0.13 | 0.89 ± 0.17 | 0.84 ± 0.10 | 0.89 ± 0.07 | 0.95 ± 0.06 | 0.94 ± 0.06 | 0.95 ± 0.06 | **0.95 ± 0.06** |
| SF3-gumbel | 30 | 0.54 ± 0.10 | 0.92 ± 0.10 | 0.92 ± 0.06 | 0.96 ± 0.02 | **0.98 ± 0.02** | 0.98 ± 0.02 | 0.97 ± 0.02 | 0.97 ± 0.02 |
| SF3-gumbel | 50 | 0.48 ± 0.07 | 0.91 ± 0.09 | 0.92 ± 0.05 | 0.96 ± 0.02 | 0.98 ± 0.01 | **0.98 ± 0.01** | 0.98 ± 0.01 | 0.98 ± 0.02 |
| SF3-gumbel | 100 | 0.44 ± 0.05 | 0.90 ± 0.09 | 0.92 ± 0.03 | 0.97 ± 0.02 | 0.98 ± 0.01 | 0.99 ± 0.01 | **0.99 ± 0.01** | 0.99 ± 0.01 |
| SF4-gumbel | 10 | 0.71 ± 0.13 | 0.87 ± 0.16 | 0.77 ± 0.09 | 0.82 ± 0.07 | **0.91 ± 0.07** | 0.90 ± 0.07 | 0.89 ± 0.07 | 0.89 ± 0.08 |
| SF4-gumbel | 30 | 0.48 ± 0.08 | 0.88 ± 0.13 | 0.87 ± 0.07 | 0.93 ± 0.03 | 0.97 ± 0.02 | **0.97 ± 0.02** | 0.96 ± 0.03 | 0.96 ± 0.03 |
| SF4-gumbel | 50 | 0.42 ± 0.07 | 0.89 ± 0.11 | 0.87 ± 0.05 | 0.95 ± 0.03 | 0.98 ± 0.02 | **0.98 ± 0.01** | 0.97 ± 0.02 | 0.97 ± 0.02 |
| SF4-gumbel | 100 | 0.39 ± 0.05 | 0.89 ± 0.10 | 0.88 ± 0.06 | 0.96 ± 0.02 | 0.98 ± 0.02 | 0.98 ± 0.02 | **0.98 ± 0.01** | 0.98 ± 0.01 |

Table 10: DAG learning performance (measured in true positive rate, the higher the better, best results in **bold**) of different algorithms on small scale (10-100 nodes) ER{2,3,4} graphs with different noise distributions. Our algorithm performs better than previous approaches.

| Graphs | Nodes | MMPC | GES | NOTEARS | DAGMA | Order-1 | Order-2 | Order-3 | Order-4 |
|---|---|---|---|---|---|---|---|---|---|
| ER2-gauss | 10 | 0.55 ± 0.05 | 0.46 ± 0.23 | 0.05 ± 0.07 | 0.02 ± 0.05 | **0.02 ± 0.04** | 0.02 ± 0.05 | 0.02 ± 0.06 | 0.03 ± 0.08 |
| ER2-gauss | 30 | 0.56 ± 0.03 | 0.46 ± 0.21 | 0.04 ± 0.06 | 0.02 ± 0.04 | 0.02 ± 0.04 | 0.02 ± 0.03 | **0.02 ± 0.03** | 0.02 ± 0.04 |
| ER2-gauss | 50 | 0.56 ± 0.03 | 0.47 ± 0.16 | 0.05 ± 0.06 | 0.01 ± 0.02 | 0.02 ± 0.02 | **0.01 ± 0.02** | 0.02 ± 0.02 | 0.02 ± 0.03 |
| ER2-gauss | 100 | 0.55 ± 0.02 | 0.44 ± 0.11 | 0.05 ± 0.05 | 0.01 ± 0.02 | **0.01 ± 0.02** | 0.01 ± 0.02 | 0.02 ± 0.02 | 0.02 ± 0.02 |
| ER3-gauss | 10 | 0.57 ± 0.05 | 0.60 ± 0.14 | 0.10 ± 0.08 | 0.06 ± 0.06 | 0.04 ± 0.05 | 0.04 ± 0.05 | **0.03 ± 0.05** | 0.04 ± 0.05 |
| ER3-gauss | 30 | 0.60 ± 0.04 | 0.70 ± 0.13 | 0.08 ± 0.08 | 0.03 ± 0.04 | 0.03 ± 0.04 | 0.03 ± 0.04 | 0.03 ± 0.04 | **0.03 ± 0.04** |
| ER3-gauss | 50 | 0.59 ± 0.03 | 0.71 ± 0.10 | 0.10 ± 0.07 | 0.03 ± 0.03 | 0.03 ± 0.03 | 0.03 ± 0.03 | 0.03 ± 0.03 | **0.03 ± 0.03** |
| ER3-gauss | 100 | 0.58 ± 0.02 | 0.64 ± 0.12 | 0.11 ± 0.06 | 0.03 ± 0.03 | 0.03 ± 0.02 | **0.02 ± 0.02** | 0.02 ± 0.02 | 0.03 ± 0.03 |
| ER4-gauss | 10 | 0.53 ± 0.04 | 0.57 ± 0.13 | 0.09 ± 0.06 | 0.06 ± 0.05 | 0.05 ± 0.04 | 0.05 ± 0.04 | **0.04 ± 0.04** | 0.05 ± 0.04 |
| ER4-gauss | 30 | 0.61 ± 0.04 | 0.76 ± 0.07 | 0.14 ± 0.10 | 0.06 ± 0.05 | 0.05 ± 0.04 | **0.04 ± 0.04** | 0.05 ± 0.05 | 0.05 ± 0.06 |
| ER4-gauss | 50 | 0.62 ± 0.03 | 0.80 ± 0.05 | 0.15 ± 0.07 | 0.04 ± 0.04 | 0.04 ± 0.03 | **0.04 ± 0.03** | 0.04 ± 0.04 | 0.05 ± 0.04 |
| ER4-gauss | 100 | 0.61 ± 0.03 | 0.77 ± 0.06 | 0.15 ± 0.07 | 0.04 ± 0.04 | 0.04 ± 0.04 | 0.04 ± 0.04 | **0.04 ± 0.04** | 0.04 ± 0.04 |
| ER2-exp | 10 | 0.55 ± 0.05 | 0.48 ± 0.24 | 0.05 ± 0.07 | 0.03 ± 0.05 | **0.02 ± 0.04** | 0.02 ± 0.05 | 0.03 ± 0.06 | 0.04 ± 0.08 |
| ER2-exp | 30 | 0.56 ± 0.03 | 0.47 ± 0.21 | 0.05 ± 0.07 | 0.02 ± 0.04 | 0.02 ± 0.04 | 0.02 ± 0.04 | **0.01 ± 0.03** | 0.02 ± 0.04 |
| ER2-exp | 50 | 0.56 ± 0.03 | 0.49 ± 0.15 | 0.05 ± 0.06 | **0.01 ± 0.02** | 0.01 ± 0.02 | 0.01 ± 0.02 | 0.02 ± 0.02 | 0.02 ± 0.03 |
| ER2-exp | 100 | 0.55 ± 0.02 | 0.44 ± 0.12 | 0.05 ± 0.05 | **0.01 ± 0.02** | 0.01 ± 0.02 | 0.01 ± 0.02 | 0.02 ± 0.02 | 0.02 ± 0.02 |
| ER3-exp | 10 | 0.57 ± 0.06 | 0.60 ± 0.14 | 0.11 ± 0.08 | 0.06 ± 0.06 | 0.04 ± 0.05 | 0.04 ± 0.05 | **0.03 ± 0.04** | 0.04 ± 0.05 |
| ER3-exp | 30 | 0.60 ± 0.04 | 0.69 ± 0.13 | 0.08 ± 0.07 | 0.03 ± 0.04 | 0.03 ± 0.04 | 0.03 ± 0.04 | 0.03 ± 0.04 | **0.03 ± 0.04** |
| ER3-exp | 50 | 0.59 ± 0.03 | 0.70 ± 0.12 | 0.10 ± 0.08 | 0.03 ± 0.04 | 0.03 ± 0.04 | **0.03 ± 0.03** | 0.03 ± 0.03 | 0.03 ± 0.03 |
| ER3-exp | 100 | 0.58 ± 0.02 | 0.64 ± 0.11 | 0.11 ± 0.06 | 0.03 ± 0.03 | 0.03 ± 0.02 | 0.03 ± 0.03 | 0.03 ± 0.02 | **0.03 ± 0.02** |
| ER4-exp | 10 | 0.53 ± 0.03 | 0.57 ± 0.12 | 0.09 ± 0.06 | 0.07 ± 0.05 | 0.05 ± 0.04 | 0.05 ± 0.04 | **0.04 ± 0.04** | 0.05 ± 0.04 |
| ER4-exp | 30 | 0.61 ± 0.04 | 0.76 ± 0.07 | 0.13 ± 0.09 | 0.06 ± 0.05 | **0.04 ± 0.04** | 0.05 ± 0.04 | 0.06 ± 0.05 | 0.05 ± 0.06 |
| ER4-exp | 50 | 0.62 ± 0.03 | 0.80 ± 0.05 | 0.15 ± 0.08 | 0.05 ± 0.04 | 0.04 ± 0.03 | **0.04 ± 0.03** | 0.04 ± 0.04 | 0.05 ± 0.04 |
| ER4-exp | 100 | 0.61 ± 0.03 | 0.77 ± 0.06 | 0.15 ± 0.07 | 0.04 ± 0.04 | 0.04 ± 0.04 | 0.04 ± 0.04 | **0.04 ± 0.04** | 0.04 ± 0.04 |
| ER2-gumbel | 10 | 0.55 ± 0.04 | 0.51 ± 0.23 | 0.03 ± 0.06 | **0.02 ± 0.05** | 0.02 ± 0.06 | 0.03 ± 0.06 | 0.03 ± 0.06 | 0.03 ± 0.08 |
| ER2-gumbel | 30 | 0.56 ± 0.03 | 0.49 ± 0.19 | 0.05 ± 0.07 | **0.02 ± 0.03** | 0.02 ± 0.04 | 0.02 ± 0.04 | 0.03 ± 0.04 | 0.03 ± 0.05 |
| ER2-gumbel | 50 | 0.56 ± 0.03 | 0.48 ± 0.16 | 0.06 ± 0.06 | **0.02 ± 0.03** | 0.02 ± 0.03 | 0.02 ± 0.03 | 0.03 ± 0.04 | 0.03 ± 0.04 |
| ER2-gumbel | 100 | 0.55 ± 0.02 | 0.44 ± 0.12 | 0.07 ± 0.06 | **0.01 ± 0.02** | 0.02 ± 0.02 | 0.02 ± 0.02 | 0.02 ± 0.02 | 0.03 ± 0.03 |
| ER3-gumbel | 10 | 0.57 ± 0.05 | 0.60 ± 0.15 | 0.07 ± 0.07 | 0.04 ± 0.05 | 0.03 ± 0.04 | 0.04 ± 0.06 | 0.03 ± 0.05 | **0.02 ± 0.04** |
| ER3-gumbel | 30 | 0.60 ± 0.04 | 0.70 ± 0.13 | 0.08 ± 0.08 | **0.02 ± 0.03** | 0.03 ± 0.04 | 0.03 ± 0.04 | 0.05 ± 0.05 | 0.04 ± 0.05 |
| ER3-gumbel | 50 | 0.59 ± 0.03 | 0.70 ± 0.12 | 0.10 ± 0.08 | 0.03 ± 0.03 | 0.04 ± 0.03 | **0.03 ± 0.03** | 0.04 ± 0.04 | 0.04 ± 0.04 |
| ER3-gumbel | 100 | 0.58 ± 0.02 | 0.64 ± 0.12 | 0.12 ± 0.06 | **0.03 ± 0.03** | 0.03 ± 0.03 | 0.03 ± 0.03 | 0.03 ± 0.03 | 0.04 ± 0.03 |
| ER4-gumbel | 10 | 0.53 ± 0.04 | 0.56 ± 0.13 | 0.08 ± 0.06 | 0.05 ± 0.04 | **0.03 ± 0.04** | 0.04 ± 0.04 | 0.04 ± 0.04 | 0.04 ± 0.04 |
| ER4-gumbel | 30 | 0.61 ± 0.04 | 0.75 ± 0.08 | 0.13 ± 0.09 | 0.04 ± 0.04 | 0.04 ± 0.04 | **0.04 ± 0.04** | 0.06 ± 0.06 | 0.06 ± 0.06 |
| ER4-gumbel | 50 | 0.61 ± 0.03 | 0.80 ± 0.05 | 0.15 ± 0.08 | 0.05 ± 0.05 | 0.04 ± 0.04 | **0.04 ± 0.03** | 0.05 ± 0.04 | 0.06 ± 0.04 |
| ER4-gumbel | 100 | 0.61 ± 0.03 | 0.77 ± 0.06 | 0.16 ± 0.07 | 0.04 ± 0.04 | 0.05 ± 0.05 | 0.04 ± 0.04 | **0.04 ± 0.04** | 0.05 ± 0.04 |
| SF2-gauss | 10 | 0.78 ± 0.11 | 0.91 ± 0.19 | 0.88 ± 0.09 | 0.91 ± 0.07 | **0.97 ± 0.04** | 0.97 ± 0.04 | 0.97 ± 0.04 | 0.96 ± 0.05 |
| SF2-gauss | 30 | 0.62 ± 0.09 | 0.95 ± 0.08 | 0.93 ± 0.05 | 0.95 ± 0.04 | 0.98 ± 0.02 | 0.99 ± 0.02 | **0.99 ± 0.02** | 0.98 ± 0.02 |
| SF2-gauss | 50 | 0.57 ± 0.08 | 0.92 ± 0.11 | 0.93 ± 0.05 | 0.97 ± 0.02 | 0.99 ± 0.01 | 0.99 ± 0.01 | **0.99 ± 0.01** | 0.99 ± 0.01 |
| SF2-gauss | 100 | 0.54 ± 0.06 | 0.92 ± 0.08 | 0.94 ± 0.04 | 0.97 ± 0.02 | 0.99 ± 0.01 | **0.99 ± 0.01** | 0.99 ± 0.01 | 0.99 ± 0.01 |
| SF3-gauss | 10 | 0.73 ± 0.13 | 0.86 ± 0.19 | 0.75 ± 0.11 | 0.80 ± 0.10 | 0.91 ± 0.08 | 0.91 ± 0.08 | **0.91 ± 0.07** | 0.91 ± 0.07 |
| SF3-gauss | 30 | 0.53 ± 0.10 | 0.92 ± 0.09 | 0.88 ± 0.06 | 0.92 ± 0.04 | 0.97 ± 0.02 | 0.97 ± 0.03 | **0.97 ± 0.03** | 0.97 ± 0.03 |
| SF3-gauss | 50 | 0.47 ± 0.07 | 0.90 ± 0.10 | 0.89 ± 0.06 | 0.93 ± 0.03 | 0.98 ± 0.02 | 0.98 ± 0.02 | 0.98 ± 0.02 | **0.98 ± 0.02** |
| SF3-gauss | 100 | 0.43 ± 0.05 | 0.90 ± 0.09 | 0.89 ± 0.04 | 0.95 ± 0.02 | 0.98 ± 0.01 | 0.98 ± 0.01 | 0.98 ± 0.01 | **0.98 ± 0.01** |
| SF4-gauss | 10 | 0.71 ± 0.13 | 0.89 ± 0.15 | 0.68 ± 0.09 | 0.73 ± 0.07 | 0.84 ± 0.07 | **0.85 ± 0.07** | 0.84 ± 0.07 | 0.84 ± 0.07 |
| SF4-gauss | 30 | 0.49 ± 0.08 | 0.87 ± 0.12 | 0.80 ± 0.08 | 0.87 ± 0.04 | 0.94 ± 0.03 | **0.95 ± 0.03** | 0.94 ± 0.03 | 0.94 ± 0.03 |
| SF4-gauss | 50 | 0.42 ± 0.07 | 0.88 ± 0.10 | 0.83 ± 0.05 | 0.91 ± 0.03 | 0.96 ± 0.02 | **0.96 ± 0.02** | 0.96 ± 0.02 | 0.96 ± 0.02 |
| SF4-gauss | 100 | 0.39 ± 0.05 | 0.90 ± 0.09 | 0.85 ± 0.06 | 0.93 ± 0.03 | 0.97 ± 0.02 | 0.97 ± 0.02 | 0.97 ± 0.02 | **0.97 ± 0.01** |
| SF2-exp | 10 | 0.79 ± 0.11 | 0.91 ± 0.18 | 0.87 ± 0.09 | 0.91 ± 0.07 | **0.97 ± 0.04** | 0.97 ± 0.04 | 0.97 ± 0.04 | 0.96 ± 0.05 |
| SF2-exp | 30 | 0.62 ± 0.09 | 0.93 ± 0.11 | 0.93 ± 0.05 | 0.95 ± 0.04 | 0.98 ± 0.02 | 0.98 ± 0.02 | **0.99 ± 0.02** | 0.98 ± 0.02 |
| SF2-exp | 50 | 0.57 ± 0.08 | 0.92 ± 0.11 | 0.93 ± 0.05 | 0.99 ± 0.02 | 0.99 ± 0.01 | **0.99 ± 0.01** | 0.99 ± 0.01 | 0.99 ± 0.01 |
| SF2-exp | 100 | 0.54 ± 0.06 | 0.91 ± 0.08 | 0.94 ± 0.04 | 0.97 ± 0.02 | **0.99 ± 0.01** | 0.99 ± 0.01 | 0.99 ± 0.01 | 0.99 ± 0.01 |
| SF3-exp | 10 | 0.74 ± 0.13 | 0.90 ± 0.17 | 0.75 ± 0.11 | 0.81 ± 0.09 | 0.90 ± 0.07 | 0.91 ± 0.08 | **0.92 ± 0.07** | 0.90 ± 0.07 |
| SF3-exp | 30 | 0.53 ± 0.10 | 0.92 ± 0.10 | 0.88 ± 0.06 | 0.92 ± 0.04 | 0.97 ± 0.02 | 0.96 ± 0.03 | **0.97 ± 0.03** | 0.97 ± 0.03 |
| SF3-exp | 50 | 0.47 ± 0.07 | 0.91 ± 0.09 | 0.88 ± 0.06 | 0.93 ± 0.03 | 0.97 ± 0.02 | **0.98 ± 0.02** | 0.97 ± 0.02 | 0.97 ± 0.01 |
| SF3-exp | 100 | 0.43 ± 0.05 | 0.89 ± 0.10 | 0.89 ± 0.04 | 0.95 ± 0.02 | 0.98 ± 0.01 | 0.98 ± 0.01 | 0.98 ± 0.01 | **0.98 ± 0.01** |
| SF4-exp | 10 | 0.71 ± 0.13 | 0.88 ± 0.16 | 0.68 ± 0.09 | 0.72 ± 0.07 | 0.84 ± 0.08 | **0.84 ± 0.07** | 0.84 ± 0.08 | 0.84 ± 0.07 |
| SF4-exp | 30 | 0.49 ± 0.08 | 0.88 ± 0.12 | 0.81 ± 0.07 | 0.87 ± 0.05 | **0.94 ± 0.03** | 0.94 ± 0.03 | 0.94 ± 0.03 | 0.94 ± 0.04 |
| SF4-exp | 50 | 0.42 ± 0.07 | 0.89 ± 0.10 | 0.83 ± 0.05 | 0.90 ± 0.03 | 0.96 ± 0.02 | **0.96 ± 0.02** | 0.96 ± 0.02 | 0.96 ± 0.02 |
| SF4-exp | 100 | 0.39 ± 0.05 | 0.90 ± 0.10 | 0.85 ± 0.06 | 0.93 ± 0.03 | 0.97 ± 0.02 | 0.97 ± 0.02 | **0.97 ± 0.02** | 0.97 ± 0.01 |
| SF2-gumbel | 10 | 0.78 ± 0.11 | 0.91 ± 0.18 | 0.93 ± 0.08 | 0.96 ± 0.05 | **0.98 ± 0.04** | 0.98 ± 0.04 | 0.98 ± 0.04 | 0.98 ± 0.04 |
| SF2-gumbel | 30 | 0.62 ± 0.09 | 0.94 ± 0.08 | 0.95 ± 0.05 | 0.98 ± 0.02 | 0.99 ± 0.02 | **0.99 ± 0.02** | 0.99 ± 0.02 | 0.99 ± 0.02 |
| SF2-gumbel | 50 | 0.57 ± 0.08 | 0.92 ± 0.10 | 0.95 ± 0.04 | 0.98 ± 0.02 | 0.99 ± 0.01 | **0.99 ± 0.01** | 0.99 ± 0.01 | 0.99 ± 0.01 |
| SF2-gumbel | 100 | 0.53 ± 0.06 | 0.92 ± 0.08 | 0.95 ± 0.03 | 0.99 ± 0.02 | **0.99 ± 0.01** | 0.99 ± 0.01 | 0.99 ± 0.01 | 0.99 ± 0.01 |
| SF3-gumbel | 10 | 0.73 ± 0.13 | 0.89 ± 0.17 | 0.84 ± 0.10 | 0.89 ± 0.07 | 0.95 ± 0.06 | 0.94 ± 0.06 | 0.95 ± 0.06 | **0.95 ± 0.06** |
| SF3-gumbel | 30 | 0.54 ± 0.10 | 0.92 ± 0.10 | 0.92 ± 0.06 | 0.96 ± 0.02 | **0.98 ± 0.02** | 0.98 ± 0.02 | 0.97 ± 0.02 | 0.97 ± 0.02 |
| SF3-gumbel | 50 | 0.48 ± 0.07 | 0.91 ± 0.09 | 0.92 ± 0.05 | 0.96 ± 0.02 | 0.98 ± 0.01 | **0.98 ± 0.01** | 0.98 ± 0.01 | 0.98 ± 0.02 |
| SF3-gumbel | 100 | 0.44 ± 0.05 | 0.90 ± 0.09 | 0.92 ± 0.03 | 0.97 ± 0.02 | 0.98 ± 0.01 | 0.99 ± 0.01 | **0.99 ± 0.01** | 0.99 ± 0.01 |
| SF4-gumbel | 10 | 0.71 ± 0.13 | 0.87 ± 0.16 | 0.77 ± 0.09 | 0.82 ± 0.07 | **0.91 ± 0.07** | 0.90 ± 0.07 | 0.89 ± 0.07 | 0.89 ± 0.08 |
| SF4-gumbel | 30 | 0.48 ± 0.08 | 0.88 ± 0.13 | 0.87 ± 0.07 | 0.93 ± 0.03 | 0.97 ± 0.02 | **0.97 ± 0.02** | 0.96 ± 0.03 | 0.96 ± 0.03 |
| SF4-gumbel | 50 | 0.42 ± 0.07 | 0.89 ± 0.11 | 0.87 ± 0.05 | 0.95 ± 0.03 | 0.98 ± 0.02 | **0.98 ± 0.01** | 0.97 ± 0.02 | 0.97 ± 0.02 |
| SF4-gumbel | 100 | 0.39 ± 0.05 | 0.89 ± 0.10 | 0.88 ± 0.06 | 0.96 ± 0.02 | 0.98 ± 0.02 | 0.98 ± 0.02 | **0.98 ± 0.01** | 0.98 ± 0.01 |

| | PC | GES | DAGMA | Exponential | Order 1 | Order 2 | Order 3 | Order 4 |
|---|---|---|---|---|---|---|---|---|
| SHD | $563.9 \pm 23.84$ | $4490.2 \pm 62.52$ | $588.8 \pm 18.33$ | $488.6 \pm 24.29$ | $429.6 \pm 24.73$ | $410.6 \pm 15.25$ | $401.0 \pm 16.64$ | $\mathbf{389.4 \pm 16.70}$ |
| | | | | Exp MLE | Order 1 MLE | Order 2 MLE | Order 3 MLE | Order 4 MLE |
| SHD | | | | $518.00 \pm 23.02$ | $453.70 \pm 42.12$ | $447.30 \pm 51.85$ | $\mathbf{409.50 \pm 31.02}$ | $433.00 \pm 68.98$ |
| | | | | PC Exp | PC Order-1 | PC Order-2 | PC Order-3 | PC Order-4 |
| SHD | | | | $275.40 \pm 16.01$ | $274.40 \pm 15.44$ | $273.10 \pm 15.72$ | $\mathbf{271.80 \pm 14.75}$ | $276.00 \pm 14.66$ |
| | | | | PC Exp MLE | PC Order-1 MLE | PC Order-2 MLE | PC Order-3 MLE | PC Order-4 MLE |
| SHD | | | | $274.30 \pm 14.71$ | $284.30 \pm 19.43$ | $272.20 \pm 14.04$ | $273.00 \pm 17.79$ | $\mathbf{270.20 \pm 12.58}$ |
| | PC | GES | DAGMA | Exponential | Order 1 | Order 2 | Order 3 | Order 4 |
| SHDC | $321.30 \pm 27.77$ | $4626.20 \pm 69.05$ | $674.00 \pm 31.09$ | $588.60 \pm 59.81$ | $466.50 \pm 26.43$ | $458.40 \pm 30.85$ | $447.20 \pm 30.81$ | $\mathbf{439.90 \pm 37.06}$ |
| | | | | Exp MLE | Order 1 MLE | Order 2 MLE | Order 3 MLE | Order 4 MLE |
| SHDC | | | | $574.50 \pm 42.84$ | $490.80 \pm 66.99$ | $486.80 \pm 76.47$ | $\mathbf{444.30 \pm 42.22}$ | $479.50 \pm 100.71$ |
| | | | | PC Exp | PC Order-1 | PC Order-2 | PC Order-3 | PC Order-4 |
| SHDC | | | | $\mathbf{236.20 \pm 15.16}$ | $\mathbf{236.20 \pm 15.16}$ | $\mathbf{236.20 \pm 15.16}$ | $\mathbf{236.20 \pm 15.16}$ | $\mathbf{236.20 \pm 15.16}$ |
| | | | | PC Exp MLE | PC Order-1 MLE | PC Order-2 MLE | PC Order-3 MLE | PC Order-4 MLE |
| SHDC | | | | $231.30 \pm 13.64$ | $257.50 \pm 26.14$ | $236.10 \pm 16.23$ | $236.80 \pm 23.77$ | $\mathbf{231.60 \pm 13.17}$ |

Table 11: DAG learning performance (measured in structural hamming distance, the lower the better, best results in **bold**) of different algorithms on 1000-node ER1 graphs with Gaussian noise with observation data normalized. Our algorithms performs better than the previous approaches, and as higher order DAG constraints suffers less to gradient vanishing, it tends to have better performance. We compare differential DAG learning approaches with conditional independent test based PC (Spirtes and Glymour, 1991) algorithm and score based GES (Chickering, 2002) algorithm. The result is reported in the format of average± standard derivation gathered from 10 different simulations. The results are reported as averages ± standard deviations, calculated from 10 independent simulations. In addition to the MSE score function, we also applied the MLE score function described in Ng et al. (2020). Furthermore, rather than only considering edges between variables with correlation coefficients greater than 0.1, we also evaluated cases where edges are restricted to those in the PC-estimated CPDAG (algorithms whose names begin with 'PC').

# F IMPLEMENTATION FOR ALGORITHM 1

```python
def _h_grad(self, W, s, eps=1e-20):
    M_ = W * W / s
    Iw = self.Id - M_ # self.Id is identity matrix
    icnt = 1
    Inv = self.Id + M_
    while icnt < 2 * self.d:
        M_ = M_ @ M_
        Inv = Inv + Inv @ M_
        icnt *= 2
        if self.np.max(self.np.abs(M_)) < eps:
            break
        if self.np.any(self.np.isnan(Inv)):
            break

    if self.np.any(self.np.isinf(Inv)):
        return self.np.zeros_like(Inv)

    if self.np.any(self.np.isnan(Inv)):
        return self.np.zeros_like(Inv)

    return Inv / s

def compute_h_grad(self, W, s):
```

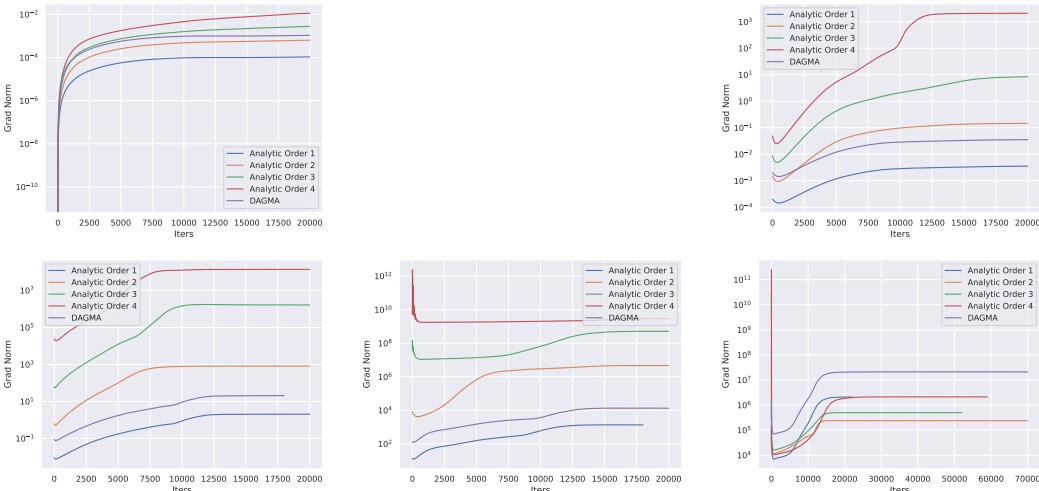

Figure 1: Gradient Norm v.s. Optimization Iterations. **Top:** Frobenius norm of gradients vs. gradient descent steps for the first two iterations in Algorithm 2. **Bottom:** Frobenius norm of gradients vs. gradient descent steps for the last three iterations in Algorithm 2. In most cases, our higher-order DAG constraints exhibit larger gradient norms compared to DAGMA, enabling our algorithm to often converge to better solutions than DAGMA.

```python
M = self._h_grad(W, s)
if self.np.any(self.np.isnan(M)) or self.np.linalg.norm(
        M @ (s * self.Id - W * W) - self.Id, ord='fro') >
        1e-6:
    if isinstance(W, cupy.ndarray):
        _, s, v = cupy.linalg.svd(W * W) # cupy does not
            have a eig lib, thus use spectral norm as an
            estimation
        cs = cupy.max(s) + 0.1 * self.h_order
    else:
        cs = np.max(np.abs(np.linalg.eigvals(
            W * W))) + 0.1 * self.h_order
else:
    cs = s
return M, cs
```

## G  EMPIRICAL RESULTS ON GRADIENTS VANISHING

In this section, we present empirical results on gradient issues in DAG learning. According to Proposition 5, the gradients of higher-order DAG constraints should have larger norms than those of lower-order constraints, given the same candidate adjacency matrix. Additionally, the behavior of our Order-1 DAG constraints is expected to align closely with that of DAGMA.

In the path-following algorithm described in Algorithm 2, five iterations are used to tune the scale of the score function. During each iteration, tens of thousands of gradient descent steps are performed. We recorded the gradient norms for various DAG constraints over the five iterations using a 1000-node ER3 DAG learning problem with Gaussian noise. The results are shown in Figure 1. In most cases, our higher-order DAG constraints exhibit larger gradient norms compared to DAGMA, enabling our algorithm to often converge to better solutions than DAGMA.

