# OpenReview forum: "Analytic DAG Constraints for Differentiable DAG Learning"
_ICLR.cc/2025/Conference — ICLR 2025 Poster_

### Official Review · Reviewer_Khru · 2024-10-24

**Soundness:** 3
**Presentation:** 3
**Contribution:** 3
**Rating:** 8
**Confidence:** 4

**Summary:**

The authors provide a theoretical analysis of analytic matrix functions serving as acyclicity constraints in the context of differentiable DAG learning.
They first show that analytic functions are valid DAG constraints, that the gradients of analytic functions are easily computable, and that analytic functions do not suffer from vanishing gradients (in contrast to other acyclicity constraints). Next, the authors show that the family of analytic functions is closed under differentiation, addition, and multiplication, thus introducing an entire family of new acyclicity constraints. Additionally, a theoretical analysis of the non-convexity of said constraints is provided, indicating that there is a trade-off between high/informative gradients and non-convexity that has to be considered when designing acyclicity constraints. Finally, an experimental evaluation shows that differentiable DAG learners equipped with acyclicity constraints based on analytic functions perform better than other constraint functions.

**Strengths:**

- the paper provides an interesting and important contribution to analyzing acyclicity constraints for differentiable DAG learners formally
- the theoretical contributions (i.e., the propositions in 3 and 4) are mathematically sound, and the proofs are correct. Note that I rated soundness with "fair" due to major weaknesses in the experimental section. The theory section is strong.
- the work provides a theory-backed "out-of-the-shelf" approach to design acyclicity constraints using differentiation, addition or multiplication of analytic functions. This simplifies the process of constructing new acyclicity constraints.
- it is shown that the proposed family of acyclicity constraints does not suffer from vanishing gradients
- the experimental section shows that analytic acyclicity constraints indeed can lead to better performances in DAG learning

**Weaknesses:**

- line 106-107: "However, these methods are known to perform poorly when applied to normalized data since they rely on scale information across variables for complete DAG recovery." This statement is a bit confusing since the findings in [1] and [2] (which generalized [1]) show that it is not the acyclicity constraint that leads to bad performance but the loss itself (MSE in this case).
- line 127-129: the claim that Eq. 4 converges if the spectral radius of $\mathbf{A}$ is smaller than $r$ is correct. However, it would be good to provide some intuition (and maybe proof in the appendix) since most propositions are based on that claim/assumption.
- line 154-156: while I agree that in terms of computational efficiency, it is desirable to have acyclicity constraints whose gradient is representable by itself, the evaluation of these constraint functions can still be costly if not carefully chosen. It would be good if the authors could elaborate a bit more on that and provide an intuition why analytic functions are computationally efficient beyond the gradient computation.
- Eq. 10: Could the authors provide a more detailed derivation of the constraints? It's easier to see where the constraints in Eq. 10 come from if Prop. 1 and 2 are actually applied to the functions in Eq. 9.
- Eq. 11: shouldn't it rather be $\frac{\partial f^s_{\text{log}}(x)}{\partial x} = -s (s -x)^{-1} = -s \cdot f^s_{inv}$?
- line 223: the authors claim that $(\mathbf{I} - \mathbf{B} / s)^{-1}$ can be cached. However, $\mathbf{B}$ will change in each iteration and thus hast to be re-computed, right?
- Prop. 5: Couldn't the fact that the gradient of the trace of $(\mathbf{I} - \mathbf{B} / s)^{-n}$ is smaller than that of $(\mathbf{I} - \mathbf{B} / s)^{-n-k}$ potentially lead to exploding, rather than vanishing gradients? A brief explanation on this would be appreciated.
- Algorithms 1 and 2 are not properly discussed in the main text. Especially, it is not entirely comprehensible from the text what the parameters of the algorithms mean.
- the text in Sec. 4 and 5 reads a bit "rushed" and is not polished.
- I think the experiments miss the main message the authors try to convey: In Sec. 3 and 4 the authors provide a strong theoretical analysis of the acyclicity constraints. However, the experiments seem to be mostly based on the paragraph in line 356-367 where the authors briefly discuss the learning dynamics of differentiable DAG learners when the scale of the data is known/unknown which is not the main contribution of this paper. The authors only show that analytical DAG constraints yield better results on simulated data (except Tab 4 where results are not significant and standard deviation is missing). While this is an important result and demonstrates that the proposed family of acyclicity constraints positively affects learning on simulated data, it remains unclear from an empirical perspective if the initial goal of solving the vanishing gradient problem was achieved. To underline the paper's important theoretical contributions, the authors should revise their experiments and rather analyze the learning dynamics and gradient statistics over learning instead of focusing on evaluating the methods' accuracies with the new acyclicity constraints. Also, since the acyclicity constraint is not primarily affected by data scale, the focus of the experimental evaluation is questionable.

# References
[1] Reisach et al. Beware of the simulated dag! causal discovery benchmarks may be easy to game. NeurIPS 2021.

[2] Seng et al. Learning Large DAGs is Harder Than You Think. ICLR 2024.

**Questions:**

- line 195-198: that's an interesting property but what is it actually good for? Because using the gradient/derivative of an analytic function essentially yields another analytic function with a shift by one in the polynomial degrees. As we only need a nilpotent matrix to ensure DAGness, this property may not necessarily yield a benefit? Could the authors please provide some more information on that?
-  paragraph in line 356-367: Since most of your experiments are built upon this paragraph, could the authors please provide more intuition on why the known/unknown variance has an influence on MSE. What about other losses like LL? Is this related to the findings of [2]?


# References
[1] Reisach et al. Beware of the simulated dag! causal discovery benchmarks may be easy to game. NeurIPS 2021.

[2] Seng et al. Learning Large DAGs is Harder Than You Think. ICLR 2024.

---

> ### Author Response · Authors · 2024-11-23
>
> Thank you for your detailed review and insightful feedback on our paper. We have carefully considered your suggestions and have made revisions accordingly. Here is a response to the specific points raised in your review:
>
> ###  line 195-198: that's an interesting property but what is it actually good for? Because using the gradient/derivative of an analytic function essentially yields another analytic function with a shift by one in the polynomial degrees. As we only need a nilpotent matrix to ensure DAGness, this property may not necessarily yield a benefit? Could the authors please provide some more information on that?
>
> After differentiation, we obtain new DAG constraints in
> which the issue of gradient vanishing is alleviated. In equal variance
> DAG learning scenarios, gradient vanishing can pose a significant
> challenge, especially for large graphs. Therefore, utilizing DAG
> constraints with more robust gradients may lead to improved
> performance. In cases of unequal variance, implementing suitable
> constraints on the graph structure can mitigate the effects of
> gradient vanishing, consequently enhancing overall performance.
>
>
> ### paragraph in line 356-367: Since most of your experiments are built upon this paragraph, could the authors please provide more intuition on why the known/unknown variance has an influence on MSE. What about other losses like LL? Is this related to the findings of [2]?
>
> For unequal noise variance cases, both the inconsistency
> score function and non-convexity contribute to the poor performance,
> as demonstrated in the additional experiment detailed in Table
> 11. Upon replacing the Mean Squared Error (MSE) with a Maximum
> Likelihood Estimation (MLE) score function from Ng 2020, the
> performance deteriorates compared to MSE. Ng 2023 also indicates that
> the initial adjacency matrix value can significantly impact
> performance in unequal variance cases.
>
> Our work aligns closely with [2]. While the results in [2] primarily
> emphasize the consistency of the score function, our focus is
> primarily on the optimization procedure.
>
>
>
> ### line 106-107: "However, these methods are known to perform poorly when applied to normalized data since they rely on scale information across variables for complete DAG recovery." This statement is a bit confusing since the findings in [1] and [2] (which generalized [1]) show that it is not the acyclicity constraint that leads to bad performance but the loss itself (MSE in this case).
>
> Thank you for highlighting this point. We have revised
> the section to enhance clarity. Regarding the Mean Squared Error (MSE)
> based score function, the inconsistency of the score function could be
> a significant factor influencing the subpar performance. In the case
> of the Maximum Likelihood Estimation (MLE) based score function, the
> non-convex nature of the optimization problem also plays a role in the
> decreased performance. We have an extended experiment with results provided in Table 11 to demonstrate the point.
>
>
> ### Equation 4
>
> We have added a proof in the appendix.
>
>
> ### More elaborations on the efficiencies of analytic function.
>
> Thanks for the point. The most time consuming part of the analytic function based DAG constraints is the matrix multiplication. A good property of convergent analytic function is that the error of the computation can be easily bounded and often the computationaly efficient. Given a analytic function
>
> $$
> f(\tilde{\mathbf{B}}) = c_0 \mathbb{I} + \sum_{i=1}^{\infty}c_i\tilde{\mathbf{B}}^i,
> $$
>
> assume that for a $\tilde{\mathbf{B}}$  with $\rho(\tilde{\mathbf{B}}) < 1$, it would be obvious for a truncated approximation
>
> $$
> f_k(\tilde{\mathbf{B}})  = c_0 \mathbb{I} + \sum_{i=1}^k c_i\tilde{\mathbf{B}}^i
> $$
> the error between the truncated approximation and the original analytic function can be obtained by
>
>
> $$
> \|f(\tilde{\mathbf{B}}) - f_k(\tilde{\mathbf{B}}) \|_2
> $$
>
> $$
>  =  \| \sum^{\infty}_{i=k+1} c_i\tilde{\mathbf{B}}^i\|_2
> $$
>
> $$
> \leqslant  c_{m}\|\sum_{i=k+1}^{\infty}\tilde{\mathbf{B}}^i \|_2
> $$
>
> $$ =  c_{m} \|\tilde{\mathbf{B}}^{k+1}  \sum_{i=0}^{\infty}\tilde{\mathbf{B}}^i  \|_2$$
> $$ \leqslant  c_m \|\tilde{\mathbf{B}}^{k+1}\|_2 \|(\mathbb{I} - \tilde{\mathbf{B})}^{-1}\|_2,$$
>
>
> which can be used to provide an error estimation for Algorithm 2. We will added this part in the draft.
>
> Furthermore, analytic functions  whose gradient is representable by itself can be obtained by solving ODEs such as
>
> $$
> f'(x) = a f(x) + b
> $$
> or
> $$
> f'(x) = a f^2(x)
> $$
> where the former ODE results in the exponential based DAG constraints and the second one results in the matrix inverse based one. It may be possible to apply this techniques to find more possible analytic function based DAG constraints.

---

> > ### Author Response · Authors · 2024-11-23
> >
> > ### Derivation of Eq. (10)
> >
> > The derivation can be obtained by using the rule $\nabla tr\mathbf{B}^k = [k\mathbf{B}^{k-1}]^{\top}$ multiple times.
> >
> > ### line 223: the authors claim that $(\mathbf{I} - \mathbf{B} / s)^{-1}$ can be cached. However, $\mathbf{B}$ will change in each iteration and thus hast to be re-computed, right?
> >
> >  For different iterations, we do  need to
> > recompute. The main advantage is that when computing the value of the
> > DAG constraints, the intermediate results can be cached for rapid
> > gradient computation. In contrast, a general polynomial does not
> > possess this property.
> >
> > ### Algorithm 1 and 2
> >
> >  We have included explanations of the parameters in the
> > appendix.

---

> > > ### Comment · Reviewer_Khru · 2024-11-25
> > >
> > > Thank you for the detailed response to my review.
> > >
> > > While my technical concerns/questions have been mostly addressed by the rebuttal, I still have concerns regarding the experiments.
> > >
> > > While I agree that obtaining a better graph than the baselines indicates that the approach successfully tackles the vanishing gradient problem, as claimed, one cannot draw ultimate conclusions based on the experiments provided. I still think an empirical validation of the author's claim would significantly improve the paper.
> > >
> > > Due to this, I'm currently not willing to raise my score. If the authors provide some empirical evidence for their claim or an argument why it is not necessary to empirically validate their claim, I'm willing to increase my score.

---

> > > > ### Author Response · Authors · 2024-11-26
> > > >
> > > > Thanks for the comments.  According to Proposition 5, the gradients of higher-order DAG constraints should have larger norms than those of lower-order constraints, given the same candidate adjacency matrix. Additionally, the behavior of our Order-1 DAG constraints is expected to align closely with that of DAGMA. Empirically, different DAG constraints may result in different traces, but we may still observe that our higher order DAG constraints have larger gradients than lower ones, as well as the DAGMA ones. We have revised the draft and provided empirical analysis in Figure 1 and Appendix E.

---

> > > > > ### Comment · Reviewer_Khru · 2024-11-26
> > > > >
> > > > > Thank you for the prompt answer.
> > > > >
> > > > > Fig. 1 is a good result, empirically validating the paper's theoretical contribution.
> > > > > Thus, I will raise my score as promised.

---

### Official Review · Reviewer_5eCV · 2024-11-03

**Soundness:** 4
**Presentation:** 3
**Contribution:** 3
**Rating:** 8
**Confidence:** 4

**Summary:**

The paper tackles the gradient vanishing problem of the DAG constraints in differentiable DAG learning methods. The main contributions are:
1. Showing that any analytic function can be used as a DAG constraint and unifying existing constraints into this framework. In addition, the closeness property of the function class allows for easy generation of constraints with large gradient magnitude.
2. An algorithm for choosing the spectral radius to amplify the gradient during training.
3. Demonstrating the outperformance of the proposed method through extensive experiments on linear simulated data and non-linear real data.

**Strengths:**

1. The class of analytic DAG constraints unifies the existing DAG constraints. In addition, the author provides a simple solution to amplify the gradient by multiplying the constraint function.
2. The discussion on the convexity of the DAG constraints provides intuition on the behavior of the high-order constraint, suggesting a tradeoff between the convexity and the gradient magnitude may be considered.
3. The experiments are extensive. The proposed method is evaluated on different DAGs ranging from low-dimensional to high-dimensional and the outperformance is well-established.

**Weaknesses:**

1. Some statements could be more rigorous. For example,
   * Lines 304-307, the discussion of the optimization objective (17). Why the statement is only restricted to Gaussian SEMs? According to Loh and Bühlmann, (2013), MSE is valid for the linear equal variance model and is proved to have good performance by Bello et al. (2022). For unequal noise variance, MSE can be inconsistent. The second sentence is also not well-structured. Why is this a problem with convexity rather than the score function?
   * Lines 362-367, same as above, MSE is not a consistent object for unequal scale, and therefore, it may not be a matter of the convexity that leads to bad results.
2. I think the evaluation of the method on the non-linear data could be even more sound if the author also considers simulations on synthetic data with equal noise variance as Bello et al. (2022).

**Questions:**

1. For the experiments on linear SEM with unknown variance, which score function is used? MSE or the log-likelihood? Is the SHD on the Markov equivalence class?
2. Does Table 7 describe the runtime? The caption seems to be incorrect.
3. It would be nice if the author could compare the choice of $s$ (and/or gradient magnitude) for the proposed method and DAGMA on an example to visualize the effectiveness of Algorithm 1 and the DAG constraint on dealing with gradient vanishing.

---

> ### Author Response · Authors · 2024-11-23
>
> Thank you for your detailed review and insightful feedback on our paper. We have carefully considered your suggestions and have made revisions accordingly. Here is a response to the specific points raised in your review:
>
> ### For the experiments on linear SEM with unknown variance, which score function is used? MSE or the log-likelihood? Is the SHD on the Markov equivalence class?
>
> We have expanded the original experiments to incorporate
> both the Mean Squared Error (MSE) and log-likelihood score
> functions. We compared the Structural Hamming Distance (SHD) of the
> original graph as well as the SHD of the Completed Partially Directed
> Acyclic Graph (CPDAG) (SHDC). The results are presented in Table
> 11. In cases of unknown variance, both the inconsistency score
> function and the non-convex objective contribute to poor
> performance. According to Ng et al. (2023), the initial selection of
> adjacency matrices can significantly impact the final performance.
>
> In our extended experiments, we observed that the log-likelihood score
> function performed worse than MSE when using correlation coefficients
> to constrain potential edges. However, it exhibited better performance
> than MSE when incorporating the CPDAG generated by PC to constrain
> edges. Moreover, both MSE and the log-likelihood score function were
> utilized to further optimize the initial CPDAG produced by PC,
> resulting in improved performance in terms of SHD and SHDC.
>
>
> ### Does Table 7 describe the runtime? The caption seems to be incorrect.
>
> We have fixed the table in the revised version.
>
> ### It would be nice if the author could compare the choice of (and/or gradient magnitude) for the proposed method and DAGMA on an example to visualize the effectiveness of Algorithm 1 and the DAG constraint on dealing with gradient vanishing.
>
> We are now generating the visualizations and will add it the a revised version.

---

> > ### Author Response · Authors · 2024-11-26
> >
> > We have revised the draft and provided empirical analysis in Figure 1 and Appendix E to compare gradient magnitude for different DAG constraints.

---

> > > ### Comment · Reviewer_5eCV · 2024-11-26
> > > **Reply to the rebuttal**
> > >
> > > Thank you for the response. Most of my concerns have been addressed, and I will maintain my score of 8.

---

### Official Review · Reviewer_abAq · 2024-11-03

**Soundness:** 3
**Presentation:** 2
**Contribution:** 2
**Rating:** 6
**Confidence:** 3

**Summary:**

the paper proposes a framework on continuous DAG constraints, based on analytic functions. It shows many existing dag constraints can be unified under such a framework. Specific choices on the analytic functions can improve computation efficiency, and in experiments even accuracy due to vanish gradient problems.

**Strengths:**

- The paper investigate in depth some properties of analytic function based DAG constraints, which shows some insights

- Empirically extensive experiments are performed. The results show the method can outperform existing baselines.

**Weaknesses:**

- Some convergence statements, such as convergence rate and eror bound, of using these analytic functions would be good to further improve the understanding of these constraints.

- Experiments contain some mixed messages, and it seems performance depends on data property and choices of orders. These make the proposed constraints harder to use in practice.

Specific Comments:
- L136: f(B) in the form of eq 3 or eq 4?
- L152: how efficient of these algorithms?
- L158: beside exponential, polynomial also holds the property. Any other function classes with such property? Could they be used as DAG constraints, any pro and cons?
- L224: how much can this acutally save in practice?
- What is the complexity of Algoritahm 1, compared to standard computations?
- L305~307: missing sentences here?
- Given the convergence property of analystical functions, one natural question to ask is: what are the convergence rates and error bounds of  the formulation using these function based dag constraints? how does different convergence radius affect these bounds?
- Experiments shows different orders of the propsed constraint, but it is unclear which order should be used in practice.
- Computation time comparison on constraints here would also be good to justify author's claim on efficiency.
- What are the differences between table 5 and 6? Titles are the same but the results are very different. Table 6 and 7 also show DAGMA are better, instead of the claim in table titles? Is the message that DAGMA should be use for large scale graphs? I assume this is true because longer paths exists and hence needs more orders?

**Questions:**

see above

---------------------- after rebuttal
thank the authors for the new updated results. It would be good to have a formal statement around error bounds, which is not yet in the draft. I have raised my score accordingly.

---

> ### Author Response · Authors · 2024-11-23
>
> Thank you for your detailed review and insightful
> feedback on our paper. We have carefully considered your suggestions
> and have made revisions accordingly. Here is a response to the
> specific points raised in your review:
>
> ### On Convergence Statements and Error Bounds:
> Thank you for highlighting the importance of convergence statements
> and error bounds for further understanding the constraints used in our
> framework. One good property for analytic function is the convergence of power series. For example, given a analytic function
>
> $$
> f(\tilde{\mathbf{B}}) = c_0 \mathbb{I} + \sum_{i=1}^{\infty} c_i\tilde{\mathbf{B}}^i,
> $$
>
> assume that for a $\tilde{\mathbf{B}}$  with $\rho(\tilde{\mathbf{B}}) < 1$, it would be obvious for a truncated approximation
>
> $$
> f_k(\tilde{\mathbf{B}})  = c_0 \mathbb{I} + \sum_{i=1}^k c_i\tilde{\mathbf{B}}^i
> $$
> the error between the truncated approximation and the original analytic function can be obtained by
>
>
> $$
> \|f(\tilde{\mathbf{B}}) - f_k(\tilde{\mathbf{B}}) \|_2
> $$
>
> $$
>  =  \| \sum^{\infty}_{i=k+1} c_i\tilde{\mathbf{B}}^i\|_2
> $$
>
> $$
> \leqslant  c_{m}\|\sum_{i=k+1}^{\infty}\tilde{\mathbf{B}}^i \|_2
> $$
>
> $$ =  c_{m} \|\tilde{\mathbf{B}}^{k+1}  \sum_{i=0}^{\infty}\tilde{\mathbf{B}}^i  \|_2$$
> $$ \leqslant  c_m \|\tilde{\mathbf{B}}^{k+1}\|_2 \|(\mathbb{I} - \tilde{\mathbf{B})}^{-1}\|_2,$$
>
>
>
>
> which can be used to provide an error estimation for Algorithm 2. We will added this part in the draft.
>
> ### Experiment Interpretation Regarding Order Selection:
> We have provided insights into choosing the appropriate order of
> constraints based on the graph size. For larger graphs, higher-order
> constraints are preferred to address long loops, while lower-order
> constraints are suitable for smaller graphs with shorter loop
> lengths. Moreover, we emphasize the preference for DAG constraints
> with finite convergence radii over infinite ones for practical
> applicability.
>
> ### Efficiency of Algorithms:
> The efficiency of algorithms utilizing analytic functions varies based
> on the function classes used. For matrix polynomials, the complexity
> is at least $\mathcal{O}(d)$, while for exponential or inverse
> functions, it may only require $\mathcal{O}(\log d)$ matrix
> multiplications.
>
> ### Additional Function Classes for DAG Constraints:
>
> We can find such DAG constraints by solving ODEs like
> $$
> f'(x) = a f(x) + b
> $$
> or
> $$
> f'(x) = a f^2(x)
> $$
> and then we can find the exponential based DAG constraints and the matrix inverse based DAG constraints. It may be possible to apply other operators on $f(x)$ to get new function classes, and this may still be an open problem.
>
> ### Practical Savings and Complexity:
> In practice, our method offers significant time savings compared to
> naive approaches by eliminating the need to recompute various
> intermediate results. Additionally, the complexity of Algorithm 1 is
> $\mathcal{O}(\log d)$ in terms of matrix multiplications, showcasing its
> computational efficiency.
>
>
>
> ### Computation Time Comparison and Table Discrepancies:
> The overall running time comparison is provided in Table 7. DAGMA usually less time than our approaches, but often because DAGMA converge faster to worse local minimals. Meanwhile, as our approach suffers less from gradient vanishing, the gradient provided by our approaches are often more informative and thus it may take longer time for converge.
>
> Furthermore, discrepancies in
> Tables 5 and 6 have been addressed, clarifying the interpretation of
> the results. We have rectified the typesetting error in Table 6 and
> provided a corrected analysis of the performance comparison between
> our approach and DAGMA.
>
> We appreciate your insightful feedback, which has significantly
> contributed to the enhancement of our paper. Thank you for your
> thorough assessment, and we are committed to addressing all concerns
> to improve the quality and clarity of our work.

---

> > ### Author Response · Authors · 2024-12-02
> >
> > Dear Reviewer abAq,
> >
> > As the discussion phase deadline approaches, we kindly ask if you could let us know whether our response has addressed the concerns you raised. Your confirmation would mean a lot to us, as it will help ensure we’ve fully understood and addressed your valuable feedback.
> >
> > Thank you so much for your time and support throughout this process. We greatly appreciate your thoughtful input.
> >
> > Best regards,
> >
> > Authors

---

### Official Review · Reviewer_xBhf · 2024-11-05

**Soundness:** 3
**Presentation:** 3
**Contribution:** 3
**Rating:** 6
**Confidence:** 3

**Summary:**

The submission presents a mathematical derivation of a family of analytical DAG constraints that can be used in the recent proposals of smooth optimization for DAG search,
Theoretical results are proposed to show that the derivation is valid and that the proposed family has favorable properties to be used in the optimization algorithms.
Additionally, a large set of experiments is performed.

**Strengths:**

- The paper is generally well written (except for some minor details, see suggestion section in Questions) and well structured.
- The experiments section is quite extensive (even if the number of replications could be bigger).
-  The theoretical part seems sound and it is well written (I have not checked the full proofs on the supplementary).
- The subject is interesting even if theoretically it is restricted to equal noise variance and linear SEM.

**Weaknesses:**

- In the simulation experiments, it would be nice to have a larger number of replicates instead of 10.  I understand the time complexity but
maybe lower dimensional systems could be explored with a higher number of repetitions to have a set of more robust results that could complement the proposed ones.
- In the experiments section the CausalDiscoveryToolbox is used to run some of the state-of-the-art methods. I would discourage that. the CausalDiscoveryToolbox would be a nice tool, but unfortunately, it is no longer maintained, While probably there are no issues in running the wrapper of other algorithms provided there I would suggest the author of this submission use other implementations. For example, PC and GES in the CausalDiscoveryToolbox are wrappers of the pc-alg R implementations, I would suggest to use directly those for reproducibility and additional safety.
- Is the code for the experiments and the implementation available somewhere? I did not find it in the submission

**Questions:**

### suggestions

For general suggestions see Weaknesses, additionally:

1. In the first paragraph of the Introduction: "the recovered DAGs can constitute a causal graphical mode"  --> "the recovered DAGs could be interpreted causally" or something similar, as the DAG by itself can't be a causal model since it lacks at least a compatible model for the probabilities or more generally a functional causal model.
2. End of second paragraph in Introduction: "w.r.t number of nodes(Chickering, 1996" a space is missing after "nodes".
3. In "as demonstrated in Wei et al. (2020). This study reveals that"  The  "This" in the beginning of the second sentence refer to Wei et al. (2020) but it could be mistakenly taken as "This study == this paper" I suggest to change the wording to clearer reflect that the following is work in Wei et al. (2020).
4. Section 2 Preliminaries: Similar to suggestion 1. above,  I think $\mathcal{G}$ should only denote a DAG (that is a graph). And not a DAG model (which is not even clear what a DAG model is). Then given a DAG the authors could define what a compatible statistical model or directly a compatible SEM is (that is when the probability factorizes according to the DAG or for the linear SEM with indep. noise that the matrix B encodes a weighted adjeceny matrix of the graph G).
5. It would be nice to have an extended intuitive explanation for the following: "Due to the fact that the adjacency matrix of a DAG must form a nilpotent matrix, we do not need a function with infinite convergence radius"
6. before section 4 starts, "mainly due to the non-convexity of the overall objective functionNg et al. (2023)." a space is missing after "function" and before "Ng et al.".
7. In caption of Table 1: "t state-of-the-arts DAGMA(Bello et al., 2022)," a space is missing after DAGMA.
8. Table 3, is the value for GES wrong? it seems to be one order of magnitude bigger ?

---

> ### Author Response · Authors · 2024-11-23
>
> Thank you for your thorough review and constructive
> feedback. We have revised the draft according to your suggestions. I
> appreciate your detailed recommendations and have addressed them as
> follows:
>
> ### On the number of replicates in simulation experiments
> We have conducted 100 simulations for small-scale experiments with
> 10-100 nodes. For large-scale experiments, we are currently increasing
> the number of replicates to 20 and will include the updated results
> once the experiments are completed.
>
> ### On using the CausalDiscoveryToolbox in experiments
> We have verified the consistency of results between R and the
> CausalDiscoveryToolbox to ensure reliability. However, we acknowledge
> that potential inconsistencies might arise from differences in R and
> `pc-alg` package versions. To address reproducibility and ensure
> additional safety, we plan to release a Docker image containing the
> full experimental environment at publication time, rather than solely
> providing the source code.
>
> ### Availability of code and implementation
> We will include the key code and implementation details in the
> appendix. Additionally, to ensure reproducibility, we will release a
> Docker image with the complete experimental environment at publication
> time.
>
> ### On the value for GES in Table 3
> We have re-verified the results. The observed poor performance for GES
> in large graphs seems to stem from the combinatorial optimization it
> requires. For large-scale problems, the optimizer appears to converge
> to poor local minima, leading to suboptimal outcomes.

---

> > ### Comment · Reviewer_xBhf · 2024-11-25
> >
> > I thank the authors for the response, my main concern have been addressed

---

### Meta-Review · Area_Chair_Lxze · 2024-12-11

**Metareview:**

The reviewers are all positive about this paper's contributions, involving derivations of analytical DAG constraints used in DAG search methods, and theoretical results on their favorable properties, as well as experiments.   Positive comments were given on all the main aspects, such as results, writing quality, scope of experiments.  A fairly significant number of suggestions were given, but ultimately these tended to be minor and no significant concerns remained following the discussion period.  Some limitations were also noted regarding aspects such as equal noise variance, linearity, and computational considerations.

**Additional Comments On Reviewer Discussion:**

Some initial concerns on the experiments and convergence properties were deemed to be sufficiently addressed.

---

### Decision · Program_Chairs · 2025-01-22

Accept (Poster)